# Multiomic analysis of malignant pleural mesothelioma identifies molecular axes and specialized tumor profiles driving intertumor heterogeneity

Malignant pleural mesothelioma (MPM) is an aggressive cancer with rising incidence and challenging clinical management. Through a large series of whole-genome sequencing data, integrated with transcriptomic and epigenomic data using multiomics factor analysis, we demonstrate that the current World Health Organization classification only accounts for up to 10% of interpatient molecular differences. Instead, the MESOMICS project paves the way for a morphomolecular classification of MPM based on four dimensions: ploidy, tumor cell morphology, adaptive immune response and CpG island methylator profile. We show that these four dimensions are complementary, capture major interpatient molecular differences and are delimited by extreme phenotypes that—in the case of the interdependent tumor cell morphology and adapted immune response—reflect tumor specialization. These findings unearth the interplay between MPM functional biology and its genomic history, and provide insights into the variations observed in the clinical behavior of patients with MPM.

Malignant pleural mesothelioma (MPM) is a rare and aggressive disease associated with asbestos exposure[1]. The World Health Organization (WHO) histological classification distinguishes three major types with prognostic value: epithelioid (MME), biphasic (MMB) and sarcomatoid (MMS)[2]. In the past decade, genomic studies uncovered molecular profiles (clusters) related to MPM's histopathological classification, each enriched for somatic alterations in known cancer genes (for example, *BAP1* in MME and *TP53* in MMS)[3–5]. We and others undertook unsupervised analyses of these data, revealing a molecular continuum of types that explained the prognosis of the disease more accurately than any reported discrete cluster[6,7]. MPM interpatient heterogeneity at the biological and clinical level is therefore expected to be sufficiently explained by the histopathological classification, with phenotypes ranging from MME to MMS[8,9].

Nevertheless, the full extent of MPM phenotypes and the mechanisms by which they evolved are poorly understood. Histopathological features (such as architectural subtypes) and molecular features (such as aneuploidy and immune infiltration) were shown to be independent of histopathological type[8,9], suggesting that there are additional sources of heterogeneity that remain unexplained. In addition, although malignant transformation and cancer development can depend on a wide range of genomic aberrations[10–12], genomic events have not been fully described in MPM as previous efforts have been restricted to profiling only exomes or a reduced representation of genomes[3–5,13]. As a result, biological functions performed by tumor cells, and the role of genomic events in shaping these functions, remain largely unknown, hindering any meaningful progress in the diagnosis, classification and treatment of the disease[8].

We designed the MESOMICS study to uncover the main sources of molecular variation explaining MPM intertumoral heterogeneity, and to identify the underlying biological functions. Using multiomic

✉ e-mail: follm@iarc.who.int; fernandezcuestal@iarc.who.int

analyses combining genomic, transcriptomic and epigenomic data on a novel cohort of 120 MPM tumors (Supplementary Tables 1–3), we show that the current histopathological classification only explains a fraction of the molecular heterogeneity of the disease, while ploidy, adaptive immune response and CpG island methylation are as important. Taking advantage of a large cohort of whole-genome sequencing (WGS) data, we map the molecular landscape of 120 MPMs and elucidate the link between genotype and phenotype.

## Results

### Multiomic analyses uncover four axes of molecular variation

We first found that the current histopathological classification only accounts for up to 10% of the interpatient molecular differences (2–10%, depending on the molecular layer, with an average of 6%), leaving 90% unexplained (Fig. 1a). We then undertook an unsupervised decomposition of the interpatient molecular heterogeneity using Multi-Omics Factor Analysis (MOFA)[14], integrating genomic, transcriptomic and epigenomic data. We identified four independent and reproducible latent factors individually explaining more than 10% of molecular variation in at least one molecular layer, and collectively up to 61% of interpatient differences (19–61%, depending on the molecular layer, with an average of 33%; Fig. 1a, Extended Data Figs. 1–3, Supplementary Fig. 1 and Supplementary Tables 4–7). Only latent factor 2 (LF2) was associated with the histopathological classification, the recent artificial intelligence score based on digital pathology[15] and the previously proposed molecular classifications[3–7] (median q value = 6.94 × 10⁻¹¹; Fig. 1b). Therefore, LF1, LF3 and LF4 capture three prominent sources of biological variation overlooked by previous histopathological and genomic studies.

LF1 (the ploidy factor) is largely explained by tumor ploidy (r = 0.87; Fig. 1c,d). LF2 (the morphology factor) separates the main histopathological types and thus summarizes the morphological and related molecular classifications (Fig. 1a–c). LF3 (the adaptive response factor) summarizes immune infiltration with adaptive response effectors (lymphocytes) (Fig. 1c). For LF2 and LF3, enhancer methylation was the major molecular layer captured (Fig. 1a), partly explained by its implication in the tumor–immune interaction phenotype captured by LF3, and its variability in MPM samples is probably driven by cell-type heterogeneity (Supplementary Fig. 2 and Supplementary Tables 5, 6 and 8). The major feature captured by LF4 (the CpG island methylator phenotype (CIMP) factor) was methylation at gene body and promoter regions, and most of its molecular variation was strongly associated with the CIMP index (r = 0.92; Fig. 1c,e). We then identified proxies to facilitate the interpretation of the latent factors and their implementation in the clinical setting: aneuploidy for LF1; the percentage of sarcomatoid component as reported by pathologists for LF2; an adaptive versus innate immune response score (Methods) for LF3; and a five-gene CIMP index proxy (Methods) for LF4. LF1, LF3, LF4 and their proxies were statistically independent of histopathological type (that is, all histological types can be either high or low ploidy, have high or low adaptive immune responses and have a high or low CIMP index), further confirming that these latent factors represent independent sources of molecular variation (Extended Data Fig. 4a–c).

In line with our previous observations[6], tumor samples did not form clusters in MOFA but rather gradients between extreme molecular profiles (Fig. 1d,e). The ploidy factor ranged between a genomic near-haploidization (GNH) and a whole-genome doubling (WGD) profile, with a gradient of intermediate ploidies due to various levels of chromosome arm and focal amplifications and deletions (Fig. 1d). In contrast with the features found associated with the GNH subtype identified in the The Cancer Genome Atlas (TCGA) cohort[4], the single near-haploid sample, MESO_108, had a ploidy of 1.10, almost no copy-neutral loss of heterozygosity (LOH) (<1%) and no SETDB1/TP53 mutations and did not undergo WGD. Therefore, this sample does not correspond to the GNH subtype as described by Hmeljak and colleagues[4], but to another possible genomic trajectory, where genomic instability is driven by

alternative pathways. Differential gene expression analyses showed that, as reported in other tumor types[12], the most upregulated enriched pathway in WGD-positive (WGD⁺) versus WGD-negative (WGD⁻) cases was E2F targets (q value = 0.048; Supplementary Tables 9 and 10), although we could not replicate this result in the TCGA cohort[4], possibly due to the difficulty of replicating such findings in low-sample-size series (n = 11 WGD⁺ samples). The CIMP factor also ranged between two extreme profiles: CIMP-low and CIMP-high (Fig. 1e). A well-known effect of the CIMP-high phenotype is epigenetic silencing of tumor suppressor genes[16]. In line with this, we identified five Catalogue of Somatic Mutations in Cancer (COSMIC) tumor suppressor genes[17], whose expression was negatively correlated with both the CIMP index and the methylation level of their CpG island(s): CBFA2T3, FBLN2, PRF1, SLC34A2 and WT1 (median q value = 2.6 × 10⁻³; Supplementary Table 11).

We trained latent factor-based survival models and tested their performance over previously proposed prognostic factors to evaluate to what extent each latent factor captured variability predictive of prognosis (Methods). While individually they provided a prediction value similar to each other, when combining the four latent factors there was an increase in their area under the receiver operating characteristic curve value, suggesting that they capture molecular characteristics with independent prognostic value, being informative of MPM progression in a complementary manner (Extended Data Fig. 5, Supplementary Fig. 3 and Supplementary Tables 12–20). In line with evidence from multiple cancer types[12], survival was lowest for the greatest ploidy (Fig. 1f). As expected, samples in the lower extreme of the morphology factor, enriched for sarcomatoid tumors, presented the worst prognosis. The adaptive response factor linked hot tumors (tumors with a high level of immune infiltration) with better survival, whereas CIMP-low tumors had better survival than CIMP-high tumors (Fig. 1f). The previously described proxies also demonstrated prognostic value in the MESOMICS cohort, and allowed for validation of the prognostic value of the latent factors in the validation cohorts (Extended Data Fig. 4d–g). Probably due to the limited power and a potential effect of histology, the prognostic value of the ploidy and CIMP factors was not statistically significant when analyzing MME samples only; however, their respective effect size remained similar to those identified in the entire cohort (Supplementary Fig. 3). We additionally validated the existence of the four dimensions as well as their prognostic values in previously published cohorts (Supplementary Tables 21 and 22).

Finally, combining molecular and drug response data for 59 MPM cell lines from Iorio et al.[18], de Reyniès et al.[5] and Blum et al.[7], we were able to evaluate the therapeutic value of the ploidy, morphology and CIMP factors (the lack of microenvironment in cell culture models did not allow for replication of the adaptive response factor), by assessing the impact that cell line position along each latent factor had on the response to candidate drugs (Extended Data Fig. 6, Supplementary Fig. 4 and Supplementary Tables 23–26). Significant drug responses associated with the different factors were entirely orthogonal (Extended Data Fig. 6a), highlighting the fact that MOFA latent factors capture independent axes of heterogeneity in both tumoral mechanisms and therapeutic responses. Therefore, both survival and cell line analyses showed that these axes of variation are clinically relevant and have the potential for translation into clinical practice.

### Task specialization analyses reveal diverse tumor strategies

Samples along the interdependent morphology and adaptive response factors formed a triangular shape delimited by three extremes (Fig. 2a and Supplementary Fig. 5). The well-established Pareto optimum theory[19] (ParetoTI method) predicted that this pattern results from natural selection for cancer tasks, with specialist tumors close to the vertices of the triangle and generalists in the center (triangle fit P value = 0.001; Fig. 2b). Integrative gene set enrichment analysis (IGSEA) pointed to the following cancer tasks and tumor phenotypes: cell division,

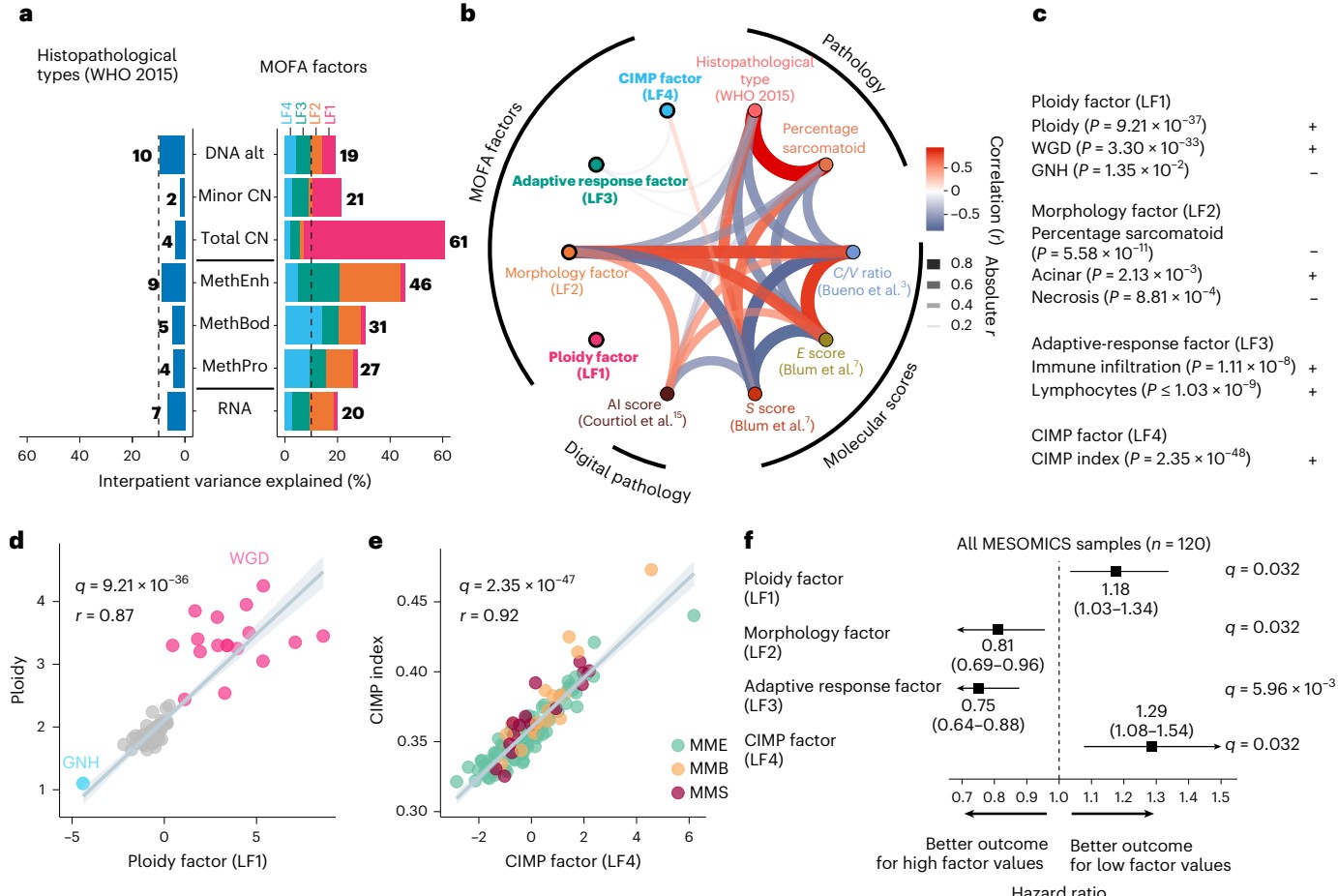

**Fig. 1 | MOFA of whole genomes, transcriptomes and methylomes of the MESOMICS cohort (*n* = 120). a**, Proportion of interpatient variance within a molecular layer explained by WHO-defined histopathological type (left) and MOFA latent factors 1–4 (right). For example, 7% of variation present in RNA expression can be explained by mesothelioma types, in contrast with 20% explained by integrative MOFA. CN, segmental copy number; DNA alt, rearrangements and mutations; MethBod, DNA methylation level at body regions; MethEnh, DNA methylation level at enhancer regions; MethPro, DNA methylation level at promoter regions; RNA, gene expression level. **b**, Network of the correlations between latent factors, tumor histopathology and previously published molecular scores. The arc colors, widths and transparency correspond to Pearson correlation coefficients. Features uncorrelated with any

other features are highlighted in bold. AI, artificial intelligence; C/V ratio, log2 ratio of *CLDN15* to *VIM* gene expression; S score, sarcomatoid gene expression score; E score, epithelioid gene expression score. **c**, Interpretation of MOFA latent factors. Plus signs indicate positive correlations and minus signs indicate negative correlations. **d**, Correlation between the ploidy factor (LF1) and ploidy. **e**, Correlation between the CIMP factor (LF4) and CIMP index. The samples are colored by histological type. **f**, Forest plot of the hazard ratios of MOFA latent factors for overall survival. The squares correspond to estimated hazard ratios and the segments correspond to their 95% confidence intervals. In **b**–**e**, *P* values, *q* values and *r* coefficients were determined by two-sided Pearson correlation tests. In **d** and **e**, the gray bands represent 95% confidence intervals.

tumor–immune interaction and acinar phenotype (Fig. 2c and Supplementary Tables 27–30 for archetypes, IGSEA significant pathways and *q* values).

Tumors specialized in the cell division task displayed upregulation of these pathways, as reported by Hausser et al. in multiple tumor types[20]. This phenotype was enriched for nonepithelioid tumors and presented higher levels of necrosis, higher grade and a greater percentage of infiltrating innate immune response cells (neutrophils) (median *q* value = 0.005). Cell division specialization was supported by high expression levels of the proliferation marker *MKI67* and increased genomic instability (estimated from genomic, transcriptomic and epigenomic data; median *q* value = 1.97 × 10⁻⁴). Tumors specialized in the tumor–immune interaction task carried upregulated immune-related pathways, high expression of immune checkpoint genes and high immune infiltration with an enrichment for adaptive response cells: B lymphocytes, CD8⁺ T cells and regulatory T cells (median *q* value = 2.73 × 10⁻³). The cell division and tumor–immune interaction specialists also showed high expression of hypoxia

response pathways and common enrichment for pathways in the invasion and tissue remodeling universal cancer task. Indeed, we found a higher epithelial-to-mesenchymal transition (EMT) score among tumors in this area of the Pareto triangle, driven by upregulation of mesenchymal genes and hypomethylation of their associated enhancers (median *q* value = 1.61 × 10⁻⁶). In line with in vitro studies showing that asbestos may induce EMT in MPM[21], we found a positive correlation between the expression of mesenchymal genes and asbestos exposure score (*r* = 0.44 and *q* value = 0.01) and a negative correlation between this score and enhancer methylation of mesenchymal genes (*r* = −0.33 and *q* value = 0.02). We also observed overexpression of neoangiogenesis-related genes, corroborating the ability of these tumors to remodel their environment.

The last extreme phenotype was characterized by samples with acinar morphology, presenting a very structured tissue organization with epithelial cells tightly linked into tubular structures, and correlated with the presence of monocytes and natural killer cells (innate immune response cells) (median *q* value = 0.022). This phenotype presented

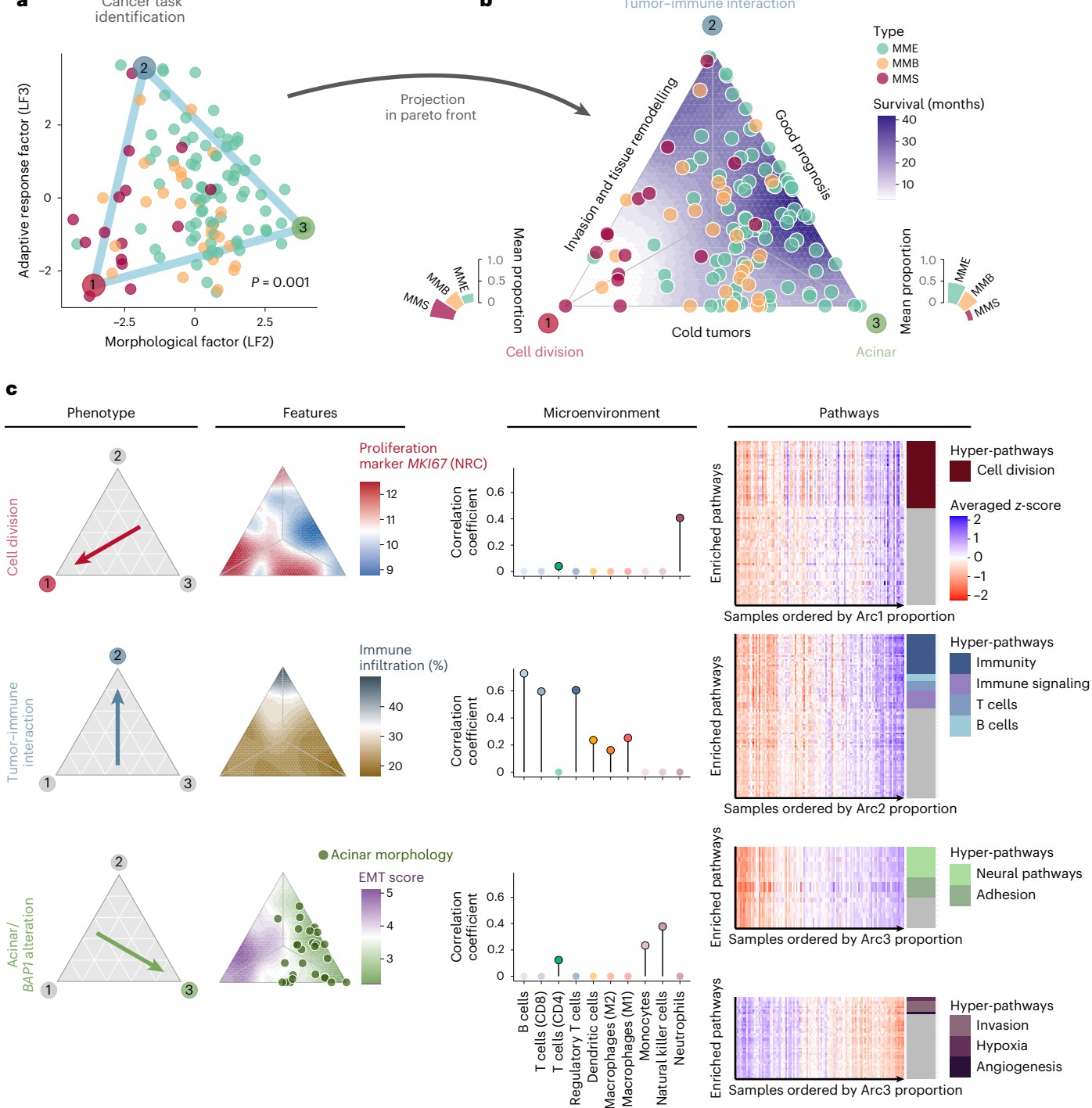

**Fig. 2 | Cancer task inference from the morphology and adaptive response factors ($n$ = 120). a**, Sample positions along the morphological (LF2) and adaptive response factors (LF3) are contained within a triangle formed by three phenotypic archetypes (colored vertices). The $P$ value corresponds to a one-sided test from the Pareto fit. **b**, Ternary plot representing the sample's distance from the three specialized profiles. The bar plots represent the association between archetypes and histopathological types. **c**, Summary table of the main phenotypes, features and overexpressed pathways (columns) identified in each profile (rows). Left, arrows indicate the focal profile of each row. Middle left, ternary plots with a color-filled background representing key features for each profile. NRC, normalized read count. Middle right, lollipop plots presenting the correlation between RNA-seq-estimated immune cell infiltration and the proportion of archetypes. Right, expression heatmaps of cancer tasks inferred from each phenotype. The rows represent enriched pathways and the columns represent the samples, ordered by increasing phenotype proportion. The heatmap color scale corresponds to the averaged $z$ score of each gene set. The colored tiles on the right annotate the gene sets that belong to the hyper-pathways inferred from each phenotype.

the lowest EMT score, with overexpression of epithelial markers such as cell adhesion molecules (median $q$ value = $1.21 \times 10^{-3}$), corroborating the importance of tissue organization in this phenotype, and also low

levels of *MKI67* expression, indicating slow growth. This phenotype showed no particular tumoral specialization in any task based on the few IGSEA upregulated pathways. In line with the better prognosis

reported for this subtype[8], the acinar phenotype is characterized by the highest levels of global methylation[22] ($q$ value = $5.58 \times 10^{-10}$). Altogether, these data provide a biological understanding of the molecular and phenotypic heterogeneity characteristic of MPM tumors.

## WGS uncovers a diverse genomic landscape

We found 97% (111/115) of MPM tumors harboring at least one large genomic event (copy number variant (CNV), amplicon, homologous recombination deficiency (HRD), chromothripsis or aneuploidy; Fig. 3a). As captured by the ploidy factor, MPM samples ranged from haploid to tetraploid (Fig. 1d). The average CNV profile was highly consistent between cohorts (Supplementary Fig. 6), with several recurrent chromosome arm-level CNVs, as well as focal alterations encompassing known cancer genes (Fig. 3b and Supplementary Tables 31–35). As previously reported[23], all of the MTAP alterations co-occurred with CDKN2A/B (Fig. 3a and Supplementary Tables 36 and 37). We also found recurrent deletions of a prominent immune recognition gene, B2M (chr15q14; Fig. 3b).

A comprehensive analysis of mutational signatures, encompassing single-base substitutions, CNVs and structural variants[24,25], allowed us to identify the processes leading to particular somatic alteration patterns (Extended Data Fig. 7). A total of ten active single-base substitution signatures were detected in MPM genomes (Extended Data Fig. 7b); all corresponded to known COSMIC signatures and none was associated with asbestos exposure, as was previously reported[3,4]. Six tumors were found to have extrachromosomal DNA (ecDNA) (Supplementary Fig. 7 and Supplementary Table 38), and in the one sample with transcriptomic data we found increased expression of the genes predicted to be present on the ecDNA, including the known oncogene BRIP1 (Fig. 3c). We observed that the aforementioned ecDNA sample co-occurred with, and may be fueled by, kataegis[26] (Supplementary Fig. 8). Overall, kataegis was rarely seen in our cohort, contributing to only 2% of the MPM clustered mutations (Supplementary Tables 39 and 40). The identified complex mutational processes included a pattern compatible with chromothripsis. This was observed in 20% of the samples (Fig. 3a, Supplementary Fig. 9 and Supplementary Table 39) and also at the transcriptomic level, as fusion transcripts, in half of the positive samples (Supplementary Fig. 10a and Supplementary Tables 41–43). A signature of clustered structural variants was detected and significantly associated with a high structural variant load and chromothripsis (Supplementary Fig. 10b,c and Supplementary Tables 41 and 42). For one sample (MESO_019), the chromothripsis region overlapped with an ecDNA region, suggesting that chromothripsis may have been the source of the circular amplification (Fig. 3c). Finally, 23% of the samples showed a HRD phenotype, identified either by copy number signatures[25] or structural variant pattern-based methods[27] (Supplementary Fig. 11 and Supplementary Table 40). Among these samples, five harbored pathogenic germline mutations (from the ClinVar database) in one of 26 genes known to be involved in homologous recombination[28]—significantly more than the two mutations reported in the 77% of samples without HRD (Fisher's exact test, $P$ value = 0.00587).

We detected an HRD signature in nine out of 21 MPM cell lines from Iorio et al.[18], thus validating the high rate of this pattern in MPM.

In addition, the sensitivity of these cell lines to the clinically approved olaparib showed a tendency toward higher sensitivity in HRD samples compared with non-HRD samples (Supplementary Fig. 12). This may be linked with the results of a clinical trial suggesting a highly complex mechanism between the response to this drug and markers for DNA repair pathway activity[29]. Indeed, in contrast with their original hypothesis, patients with BAP1 mutations had poorer survival when treated with olaparib than wild-type patients. In line with this observation, the olaparib response was positively associated with the prognostic CIMP index factor ($r = 0.65$; Extended Data Fig. 6), meaning that CIMP-low samples were more sensitive to this poly-ADP ribose polymerase inhibitor than CIMP-high samples (which are enriched for BAP1 alterations (Fig. 5a) and associated with poorer survival (Supplementary Fig. 3)).

Despite the low mutational rate (0.98 nonsynonymous small variants per megabase; Supplementary Fig. 13a and Supplementary Tables 44–46), MPM tumors carry a particularly high number of structural variants relative to tumors with similarly low mutational burden (Fig. 4 and Supplementary Fig. 13b). The top genes altered by structural variants (≥5%) were RBFOX1, NF2, BAP1, MTAP and PCDH15 (Supplementary Fig. 14a). For RBFOX1, 13 out of 39 samples have two separate events, with most deleting part of the RNA-binding protein domain (Supplementary Fig. 14b). Many of these genomic rearrangements resulted in fusion transcripts detected at the transcriptomic level (Supplementary Figs. 10a and 15).

Combining the MESOMICS dataset with the two other large datasets from Bueno et al.[3] and the TCGA[4], we reached the sample size ($n \approx 300$) needed to detect rare driver alterations (1%). The IntOGen pipeline[30] discovered 30 MPM driver genes based on small variants (Supplementary Fig. 14c). BAP1, NF2, SETD2, TP53 and LATS2 are all known MPM driver genes. Among the other 25 genes, some were previously reported as recurrently mutated in MPM (PBRM1, KMT2D, DDX3X, PIK3CA, FBXW7, MGA, NF1, SETDB1, MYH9, PTCH1, RHOA and TRAF7)[31–33] or altered by structural variants (PTPRD and LRP1B)[34], two were found overexpressed in MPM cell lines (DNMT3B and EZH2)[35] and, for another two, germline mutations have been discovered, suggesting genetic susceptibility (NCOR1 (ref. [36]) and MYO5A[37]). The remaining seven driver genes have, to our knowledge, not been previously reported in MPM, but they are all known cancer genes, as reported in COSMIC: FAT3, NIN, ARHGAP5, HLA-A, NCOR2, SRGAP3 and WNK2. Of note, NF2 and MYH9 (IntOGen drivers) are located within the significantly deleted chr22q region, along with TTC28—a gene frequently altered by structural variants (Figs. 3a,b and 4). Beyond extending the list of putative MPM drivers, combining point mutations with structural variants allowed for refinement of the frequency of alterations in key MPM genes (Fig. 4 and Supplementary Tables 41–46).

## Genomic alterations tune the molecular profiles of MPM

Genomic events were associated with all MOFA latent factors and the extreme profiles that they encapsulated, as well as with the phenotypic specialists captured by the morphology and adaptive response factors (Fig. 5a and Supplementary Tables 47 and 48). Associated alterations significantly tuned tumor specialization ($P$ value = 0.003; Methods and Extended Data Fig. 8). In addition to ploidy, NCOR2 alterations

---

**Fig. 3 | Genomic characterization of MPM from the MESOMICS cohort.**
**a**, Recurrent large genomic events. Top, clinical, epidemiological, morphological and technical features per sample. T only represents samples with WGS on the tumor sample only. Bottom left, oncoplot describing the genomic events per sample. amp, amplification; del, deletion; ND, HRD type not determined. Bottom middle, barplot of the frequency of each event within the cohort. Bottom right, comparison of the gene expression of cancer-relevant genes belonging to frequent deletions detected by GISTIC, with regards to their copy number (CN) status. Wild-type (WT) cases correspond to samples without copy number, structural variant or single-nucleotide variant events detected. The box plots represent the median and interquartile range and the whiskers the maximum and minimum values, excluding outliers. The $n$ number above represents the number of biologically independent samples for each test. *$0.01 < q$ value ≤ 0.05; **$0.001 < q$ value ≤ 0.01; ***$q$ value ≤ 0.001. NRC, normalized read count. **b**, Cohort-level copy number profile (top), with significantly altered regions identified by GISTIC in focal peaks (middle) and at the chromosome (chr.) arm level (bottom). cnLOH, copy-neutral LOH. **c**, Data from a patient with oncogene amplification due to a chromothripsis event (MESO_019). Left, chromosomes involved in the chromothripsis event (outer circle, shattered regions; intermediate circle, copy number gain and loss; inner circle, structural variants (SVs)). Middle, reconstructed ecDNA structure. Right, gene expression in MESO_019 relative to the expression in other tumors of the cohort (quantile). Oncogenes found within the ecDNA region are represented in red. The $P$ value was determined by two-sided Wilcoxon rank-sum test. kb, kilobases.

and *TERT* amplification were associated with the ploidy factor (*q* values = $4.3 \times 10^{-18}$ and $3.3 \times 10^{-4}$, respectively; Fig. 5a). Thirty-six samples (31%) displayed *TERT* amplification, resulting in a significant increase in *TERT* expression (*P* value = $1.8 \times 10^{-5}$; Supplementary Fig. 16a,b). *TERT* amplification was accompanied by an underlying amplification of

chr5p in 81% of the positive cases. While no association was previously detected between *TERT* promoter mutations and WGD[38], here we found that both *TERT* amplification and its increased expression were associated with WGD events (*P* value = $1.6 \times 10^{-10}$ and 0.009, respectively; Supplementary Fig. 16c).

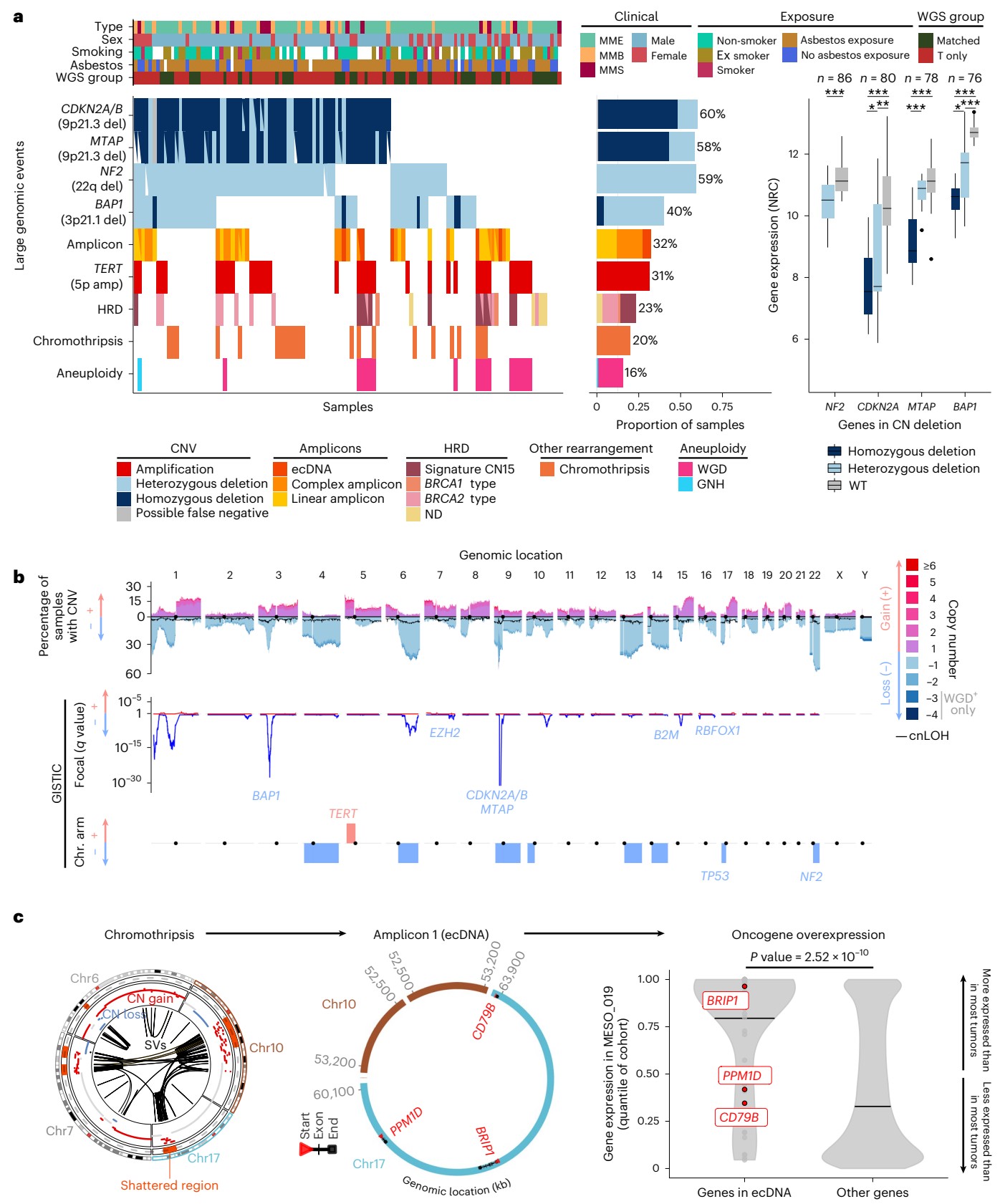

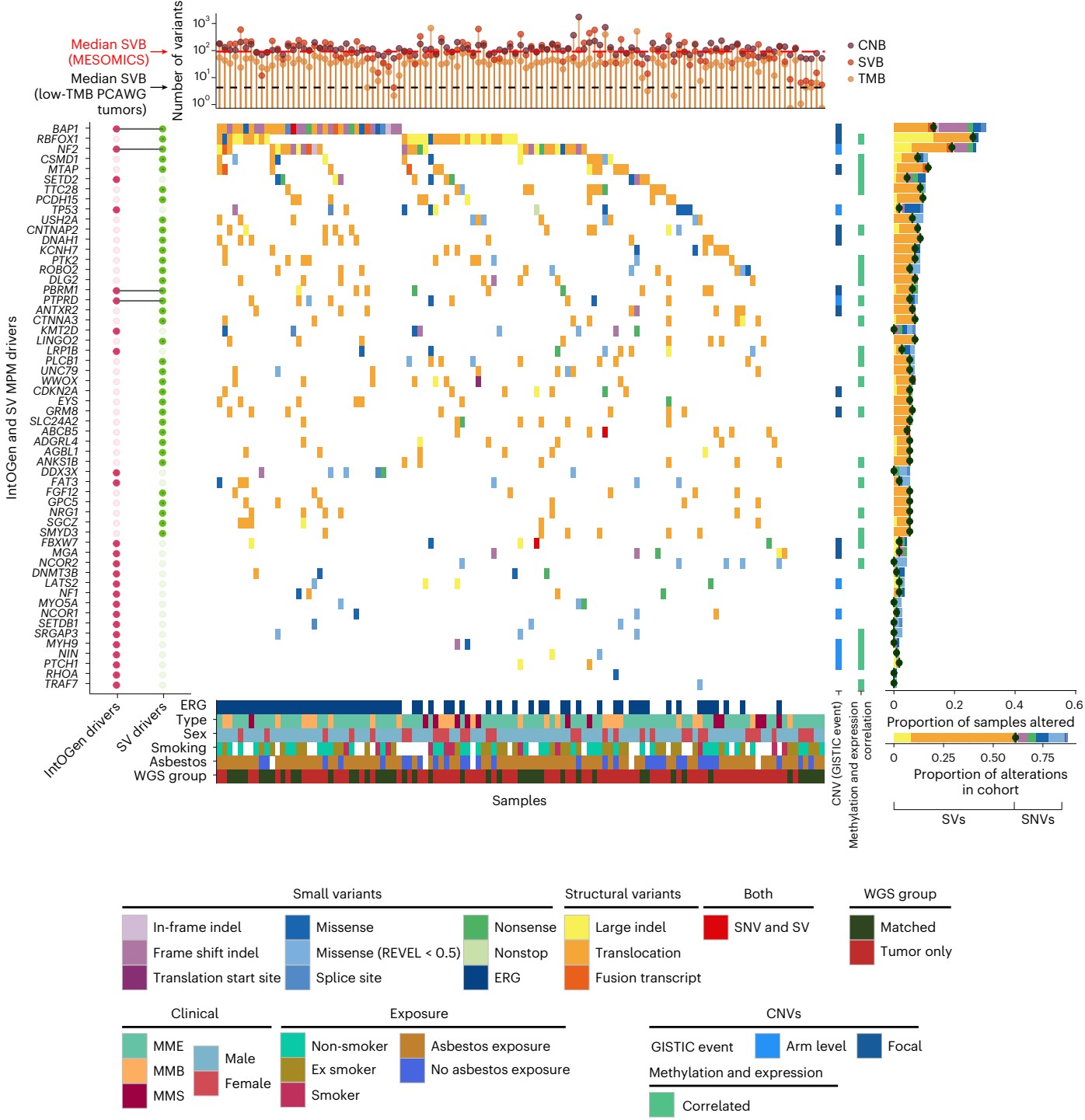

**Fig. 4 | MPM driver genes in the MESOMICS cohort.** Top, tumor mutational burden (TMB), number of segments or copy number burden (CNB) and structural variant burden (SVB) of each sample. Main, oncoplot describing genomic alterations in IntOGen and structural variant MPM driver genes per sample. These genomic events can co-occur with copy number changes. Large indels and translocations refer to structural variant events detected by structural variant callers while fusion transcripts are detected at the transcriptomic level. Each gene is also annotated as belonging to one focal or arm-level GISTIC event, as well as for being regulated by DNA methylation (right bars). Right, frequency of alterations within the cohort. For each gene, the dark green dot represents the frequency of structural variants. In the legend, ERG indicates whether the sample has one or more alteration in an ERG. Key clinical, epidemiological, morphological and technical features are given for each sample. PCAWG, Pan-Cancer Analysis of Whole Genomes; SNV, single-nucleotide variant; SV, structural variant.

Genomic alterations in epigenetic regulatory genes (ERGs) have previously been shown to drive CIMP in cancer[39]. In line with this, we found enrichment for ERGs ($P$ value = $3.4 \times 10^{-3}$; Methods and Supplementary Fig. 17), including the mesothelioma drivers *NCOR2* and *EZH2*, among the genes highly expressed in CIMP-high tumors, and more generally in the list of MPM drivers ($q$ value = $2.1 \times 10^{-5}$). Chr7q36.1del, encompassing *EZH2*, further tuned the position of the samples along the CIMP factor ($q$ value = $5.2 \times 10^{-3}$; Fig. 5a). EZH2 (enhancer of zeste homolog 2) is a histone methyltransferase that functions as part of the Polycomb repressive complex 2 (PRC2) complex to promote gene

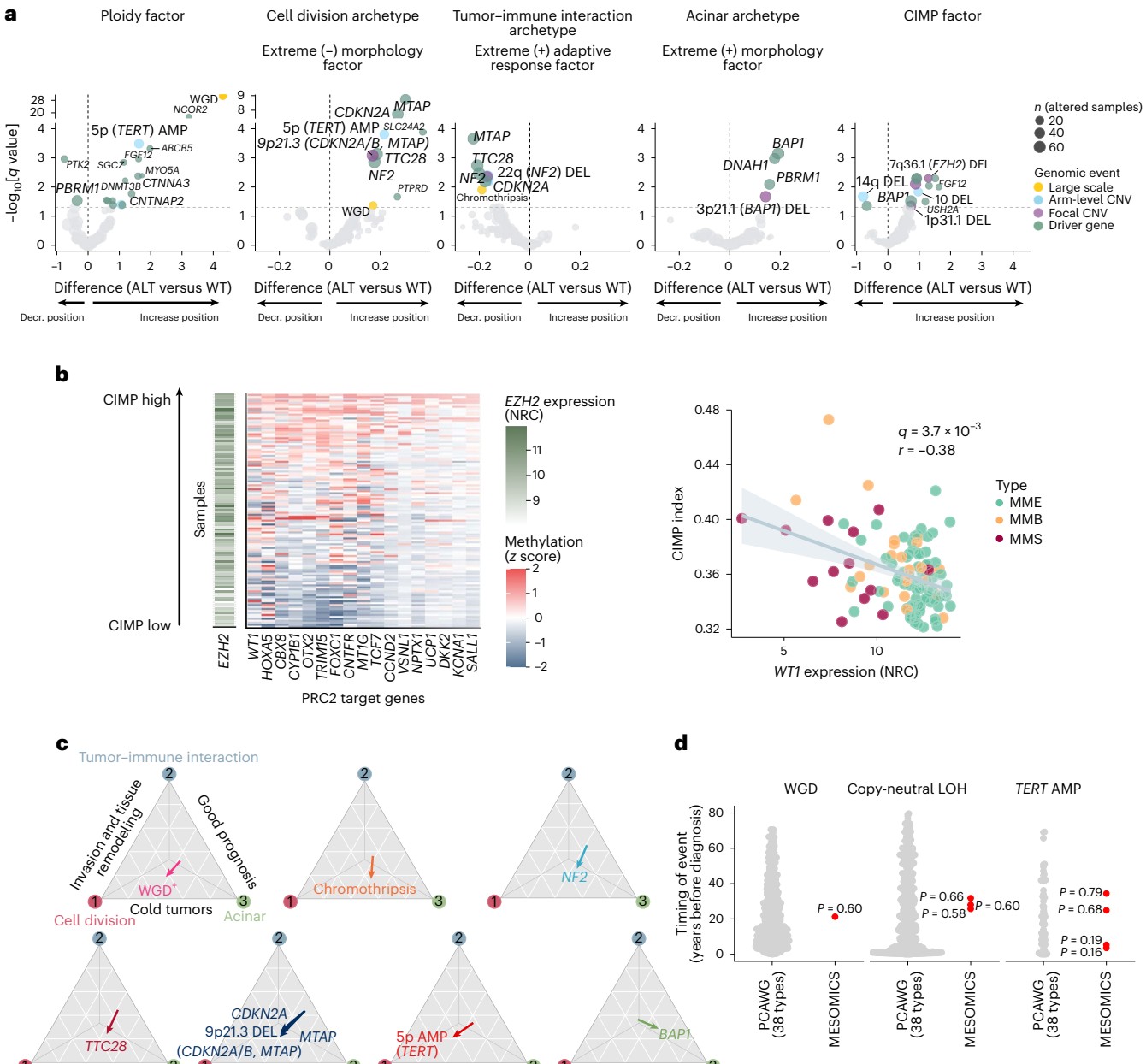

**Fig. 5 | Impact of genomic events on MPM molecular profiles. a**, Association between genomic events and MOFA factors. For each event, the ALT (altered) versus wild-type difference corresponds to the difference between the mean factor value of wild-type samples and that of altered samples. The *q* values correspond to an adjusted analysis of variance *P* value; the dashed horizontal line represents the *q* value threshold of 0.05. AMP, amplification; Decr., decrease; DEL, deletion. **b**, Association between CIMP index, *EZH2* expression (*n* = 109 samples) and PRC2 target gene methylation (*n* = 119 samples). Left, heatmap of *EZH2* gene expression (NRC) and CpG island methylation (*z* score) of PRC2 target genes whose methylation level was significantly positively correlated with CIMP index

(*q* < 0.05), for samples ordered by CIMP index. Right: correlation between *WT1* expression and CIMP index. The *q* value was determined by Pearson correlation test and the gray band corresponds to the 95% confidence interval. **c**, Effect vector of key alterations affecting specialization in the tumor tasks from Fig. 2. The effect vector corresponds to the difference in position on the Pareto front between the centroid of altered samples and the centroid of wild-type samples. **d**, Comparison of the timing of large-scale amplifications in the MESOMICS and PCAWG cohorts. The points represent estimates of the timing of genomic events. The empirical *P* values (red data points) were determined by one-sided outlier tests.

silencing of specific targets[40]. Indeed, genes whose CpG island methylation level was highest in CIMP-high tumors were enriched for PRC2 target genes (*P* value = 0.01; Fig. 5b). *WT1*, which is found downregulated in CIMP-high tumors, is particularly interesting and a vaccine against this PRC2 target is currently being assessed in clinical trials for mesothelioma[41]. Cancers frequently associated with a CIMP-high phenotype include colorectal cancer (CRC) and glioma[42,43], with *BRAF* (CRC) and *IDH1* (glioma) mutations also associated with this phenotype, as well

as with microsatellite instability in CRC[42]. Microsatellite instability and *BRAF/IDH1* mutations were rare or absent events in our series and unrelated to the CIMP phenotype (Supplementary Tables 7, 44 and 49), suggesting that the mutational processes linked with CIMP phenotype in MPM may differ from those of other cancers.

WGD and chromothripsis seemed to push tumors away from the tumor–immune interaction phenotype (*q* values = 0.042 and 0.012, respectively; Fig. 5c); indeed, both cell division and acinar phenotypes

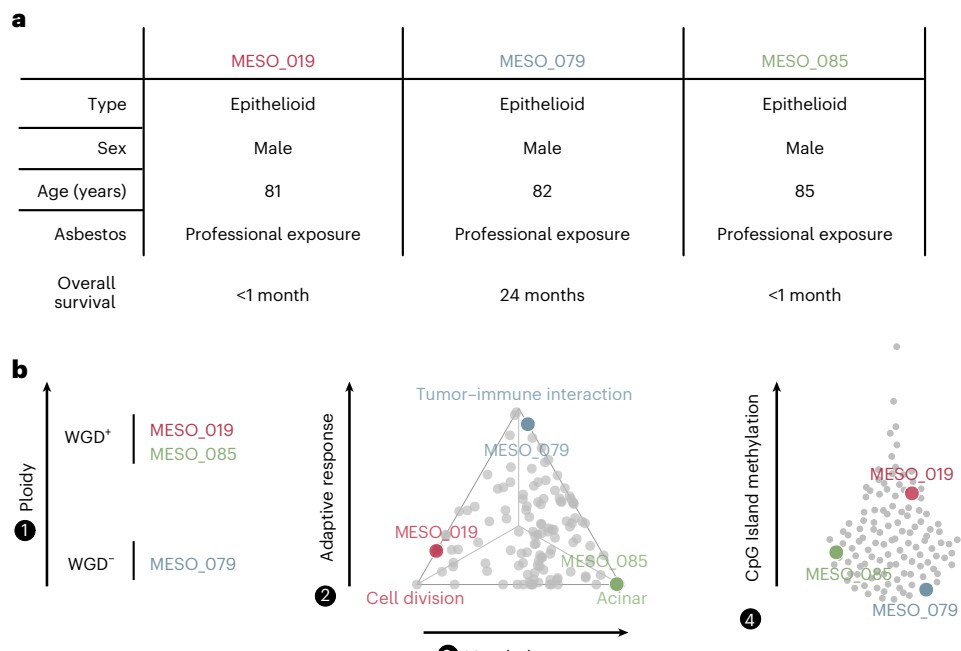

**Fig. 6 | Added value of the four-factor molecular classification in understanding intertumor heterogeneity in three example patients. a**, Patients MESO_019, MESO_079 and MESO_085 had nearly identical clinical characteristics.

**b**, The three patients had vastly different profiles based on our four-factor morphomolecular classification: different WGD status (left), opposite positions on the Pareto front (middle) and variable levels of CpG island methylation (right).

were characterized by low immune cell infiltration (cold tumors), which may be explained by the downregulation of the interferon response pathway and *B2M* expression seen in WGD + MPM tumors ($q$ value = $7.4 \times 10^{-17}$; Supplementary Fig. 18a,b,e and Supplementary Tables 9 and 10). These may represent important mechanisms for WGD$^+$ tumors to avoid the immune response[12,44]. Chromothripsis has also been associated with low immune infiltration as part of the chromosomal chaos that silences immune surveillance[45].

*CDKN2A*, *MTAP* and *NF2* alterations also converged on cold tumors (median $q$ value = 0.003). Within this cold phenotype, *TERT* amplification and alterations in *TTC28*, involved in the mitotic cell cycle, moved tumors towards cell division specialization ($q$ values = $1.6 \times 10^{-4}$ and $7.4 \times 10^{-4}$, respectively; Fig. 5c), whereas chr3p21.1del (*BAP1*, *DNAH1* and *PBRM1*) and *BAP1* mutations moved tumors toward the better-prognosis acinar phenotype ($q$ values = 0.021 and $7.1 \times 10^{-4}$, respectively; Fig. 5c), as expected given the previously reported association between *BAP1* alterations and better survival in MPM[36]. A loss of BAP1 (BRCA1-associated protein-1) expression, measured by immunohistochemistry, was also associated with this phenotype ($r = -0.38$ and $q$ value = $4.61 \times 10^{-5}$; Supplementary Fig. 19). Interestingly, an analysis of splicing variation found that the morphology factor and acinar phenotype were significantly associated with alternative splicing events (Supplementary Fig. 20a–f). Major contributions came from events in cell adhesion genes, and neuronal progenitor BAF, neuron-specific BAF and SWI/SNF complexes, potentially affecting the alternative splicing pattern of genes such as *BCL11A* and *SMARCE1* (Supplementary Fig. 20g,h). The fact that these genes (just like *BAP1*) have important roles in chromatin remodeling suggests that disruption of chromatin remodeling pathways may molecularly define the acinar phenotype.

The specialization of tumors can be influenced by early genomic events. Estimates of the timing of WGD, *TERT* amplification and copy-neutral LOH in the few samples ($n = 6$) with such events where a subclonal deconvolution was possible showed that our samples fall well within the values observed across >2,500 tumors of the Pan-Cancer Analysis of Whole Genomes Consortium[46] (empirical $P$ values = 0.16– 0.79; Fig. 5d and Supplementary Fig. 21). Thus, these genomic events

may indeed have occurred more than 10 years before diagnosis. Three out of the six patients were exposed to asbestos (of the other three patients, two had no known exposure and one had unknown exposure), among whom two had well-documented periods of exposure, from 56 to 21 years before diagnosis for MESO_048 (including the estimated timing of LOH) and from 54 to 50 years before diagnosis for MESO_057, more than 50 years before the estimated timing of *TERT* amplification, suggesting that genomic events can occur both concomitantly with and subsequent to asbestos exposure, although conclusive evidence of the timing of these alterations will need to be investigated in hypothesis-driven studies. Using a multiregional subcohort from 13 patients, we found intratumor heterogeneity in all factors except the ploidy factor, further suggesting that genomic events are mostly early and thus do not vary much across regions (Extended Data Fig. 9, Supplementary Fig. 22 and Supplementary Tables 50–52). Finally, we detected neutral tumor evolution close to the acinar phenotype ($P$ value = 0.0024; Supplementary Fig. 23) at extreme values of the morphology and adaptive response factor, suggesting that tumors with this profile were even less influenced by recent genomic events.

## Discussion

The MESOMICS project represents a substantial advancement toward the comprehensive molecular characterization of MPM, made possible by inclusion of a large WGS dataset[3,4,34] and by the depth of the multiomic integrative analyses undertaken. We demonstrated that ploidy, adaptive immune response and CpG island methylation constitute independent sources of molecular variation with quantitatively similar impacts on interpatient MPM heterogeneity as the histological classification. Despite some individual observations made in previous studies[6,7,13], these three sources of molecular variation have been mostly unexplored or unknown because of the major focus that was put on refining the histological groups, and the lack of comprehensive analysis of a large multiomics dataset. In this sense, the unifying framework aspect of our research approach allowed us to capture the entire molecular landscape of MPM, summarized in four dimensions.

Aneuploidy is one of the morphology-independent features previously reported in MPM[4] but poorly characterized. The ploidy factor identified tumors that underwent WGD, previously described in multiple cancer types as an early transformative event that dramatically destabilizes cell genetics and fuels tumor development[47]. WGD tends to be favored along the evolutionary course of low-mutational-burden tumors like MPM[12] and is suspected to serve as a genetic spare tire in case of lethal alterations[48]. As a consequence, this event shapes the cellular phenotype associated with specific vulnerabilities[12].

The CIMP has been reported in several cancer types, most notably CRC and glioblastoma, with inconsistent associations with survival[49–51]. Here we provide further evidence, to that of Blum et al.[7], of distinct variation in CIMP index within mesothelioma tumors, and have shown that a high CIMP index is independent of morphology and predictive of poorer outcome. While a universal cause for a CIMP-high phenotype has not been established, it has been previously associated with alterations in ERGs[39,52]. Indeed, our data suggest that some mesothelioma tumors may acquire a CIMP-high phenotype through the activity of the ERG *EZH2*, to hypermethylate and silence specific target genes. Such a strategy may be warranted to promote malignant transformation in a lowly mutated tumor such as mesothelioma[35].

Pareto task inference uncovered three specialized tumor profiles in the space delimited by the interdependent morphology and adaptive response factors, presumably resulting from pressures of the microenvironment, each selecting for adaptive alterations and phenotypic traits. Cell division specialists adopted a fast reproduction strategy that was expected to result from unfavorable and unpredictable environments[53], with their genomic instability suggesting adaptation through evolutionary leaps[54,55]. Immune interaction specialists adopted an immune evasion or camouflage strategy. Both phenotypes also presented characteristics of invasion and tissue remodeling specialists[20]. These tumors tended to occur in intensely asbestos-exposed individuals, suggesting that chronic inflammation (promoted by asbestos exposure[56]) may have created the unfavorable environment responsible for selective pressure. Finally, acinar phenotype specialists adopted a structured tissue organization and slow growth strategy. This suggests an equilibrium strategy that is expected to be favorable in stable, resource-rich environments with limited predation[57], in line with the lower level of asbestos exposure and limited inflammation and immune infiltration observed in these tumors. Consistent with limited environmental pressures, acinar tumors were enriched for neutral evolution and *BAP1* alterations—an event that, when combined with weak asbestos exposure in mice, greatly increased mesothelioma occurrence over weak asbestos exposure alone[58].

Overall, the four molecular factors are highly informative and capture specific profiles that are complementary in predicting tumor phenotype and aggressiveness. The fact that they are all independent and mostly unrelated to the morphology factor (histology) means that disregarding them might not only jeopardize the success of any treatment but also miss opportunities to stratify patients based on their molecular profile (Fig. 6). The tightly correlated proxies that we have identified could serve as biomarkers for response to specific therapies (such as immunotherapy for LF3) and could be easily tested in a hypothesis-driven study design. Subsequently, integrating these complementary factors would help to stratify patients for preselected-cohort clinical trials[59], a process that has proven to be beneficial in small-cell lung cancer, another aggressive recalcitrant cancer[60–62]. The results of the MESOMICS project pave the way for the establishment of a more clinically relevant morpho-molecular classification of MPM tumors.

## Online content

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

## Methods

This section briefly describes the main methods (see Supplementary Information for details on the data, processing and analyses).

### Ethics

All of the methods were carried out in accordance with relevant guidelines and regulations. This study is part of a larger study, the MESOMICS project, aiming to perform comprehensive molecular characterization of MPM, and was approved by the International Agency for Research on Cancer (IARC) Ethics Committee (project number 15-17). The samples used in this study belong to the virtual biorepository French MESOBANK. Written, informed consent was obtained from all participants and no participant compensation was provided.

### Clinical data

Age at diagnosis (in years), sex (male or female), smoking status (nonsmoker, ex smoker or smoker), asbestos exposure (exposed or nonexposed), previous treatment with chemotherapy drugs (yes or no), treatment information (surgery, chemotherapy, radiotherapy, immunotherapy or cancer history) and survival data (calculated in months from surgery to the last day of follow-up or death) were collected for all 123 patients. The median age at diagnosis was 67.5 years and 73.3% of patients were male.

### MESOMICS cohort

The MESOMICS cohort includes biological material from 123 patients with MPM (including three nonchemonaive patients who were excluded from all analyses unless explicitly mentioned) kindly provided by the French MESOBANK and annotated with detailed clinical, epidemiological and morphological data. Samples were collected from chemonaive surgically resected tumors, applying local regulations and rules at the collecting site, and included patient consent for molecular analyses, as well as the collection of de-identified data. Samples underwent an independent pathological review by the French MESOPATH reference panel, who determined that of the 120 MPM tumor samples, 79 belonged to the MME type, 26 were MMB and 15 were MMS. Of the 105 samples with an epithelioid component (79 MME and 26 MMB), solid, acinar, trabecular and tubulopapillary architectural patterns were the most frequent in the series ($n$ = 37, 31, 16 and 14, respectively).

### Discovery and intratumoral heterogeneity cohorts

Among the 123 patients with MPM, 13 had two tumor specimens collected for the study of intratumoral heterogeneity (ITH). The one with the highest tumor content, estimated by pathological review, was selected for this descriptive study and is reported in Supplementary Tables 1–3, and the other region is described in Supplementary Tables 50–52. Additionally, three patients have been reported as nonchemonaive and they were excluded from the analyses except if explicitly mentioned otherwise in the Methods.

### Pathological review

For all 136 samples (123 tumors plus 13 additional regions), a hematoxylin and eosin stain from a representative formalin-fixed, paraffin embedded block was collected for pathological review. Our pathologist (F.G.-S.) performed a detailed pathological review and classified all tumors according to the 2015 WHO classification[63,64]. The hematoxylin and eosin stain was also used to assess the quality of the frozen material selected for molecular analyses and to confirm that all frozen samples were at least 70% tumor cells.

### Artificial Intelligence analysis

Whole-slide image-based artificial intelligence prognostic scores were computed using the artificial intelligence MesoNet model based on morphological features, developed by Owkin—an artificial intelligence for medical research company[15].

### Statistical analyses

All analyses were performed in R version 4.1.2. All tests involving multiple comparisons were adjusted using the Benjamini–Hochberg procedure, controling the false discovery rate using the p.adjust R function (stats package version 3.4.4). To limit false discoveries, we took a conservative $q$ value threshold of 0.05. In addition, in line with the American Statistical Association statement on the misuse of $P$ values[65], which intends to 'steer research into a "post $P < 0.05$ era"', we report all $P$ and $q$ values, even those that may be closer to arbitrary thresholds such as the 5% threshold. To improve the reproducibility of our results, we summarize in Supplementary Tables 21 and 22 all $P$ and $q$ values reported in the text and main figures, along with details about the tests performed (hypothesis, model and sample size) and replication performed with additional cohorts.

### Survival analysis

Survival analysis has been performed using Cox's proportional hazard model from which the significance of the hazard ratio between the reference and the other levels has been evaluated using Wald tests. We assessed the global significance of the model using the logrank test statistic (R package survival version 2.41-3) and drew Kaplan–Meier and forest plots using the R package survminer (version 0.4.2).

### DNA extraction

Included samples were extracted using the Gentra Puregene Tissue Kit (4 g) (158667; Qiagen), following the manufacturer's instructions. All DNA samples were quantified using the fluorometric method (Quant-iT PicoGreen dsDNA Assay; Life Technologies) and assessed for purity by NanoDrop (Thermo Scientific) 260/280 and 260/230 ratio measurements. The DNA integrity of the fresh frozen samples was checked with a TapeStation system (Agilent Biotechnologies) using Genomic DNA ScreenTape (Agilent Biotechnologies).

### RNA extraction

Included samples were extracted using the AllPrep DNA/RNA extraction kit (Qiagen) following the manufacturer's instructions. All RNA samples were treated with DNAse I for 15 min at 30 °C. The RNA integrity of the frozen samples was checked with a TapeStation system (Agilent Biotechnologies) using RNA ScreenTape (Agilent Biotechnologies).

Because of unsuccessful extraction (impacting either the quality or the quantity), we obtained different numbers of MPM samples for which WGS, DNA methylation or RNA sequencing (RNA-seq) data are available (Supplementary Tables 1–3).

### DNA sequencing

**Sequencing.** WGS was performed by the Centre National de Recherche en Génomique Humaine (Institut de Biologie François Jacob, CEA) on 130 fresh frozen MPMs, 54 of which with matched normal tissue or blood samples. We used an Illumina TruSeq DNA PCR-Free Library Preparation Kit (20015963; Illumina) according to the manufacturer's instructions and sequenced them on a HiSeq X Five platform (Illumina) as paired-end 150-base pair reads. Samples paired with matched normal tissue or blood had a target sequencing depth of 60× and other samples had a target depth of 30×.

**Data processing.** WGS reads were mapped to the reference genome GRCh38 (with ALT and decoy contigs) using our in-house workflow (https://github.com/IARCbioinfo/alignment-nf; release version 1.0)[66]. In summary, this workflow relies on the Nextflow domain-specific language[67] version 20.10.0.5430 and consists of four steps: read mapping (software BWA[68]; version 0.7.15), duplicate marking (software samblaster[69]; version 0.1.24), read sorting (software sambamba[70]; version 0.6.6) and base quality score recalibration using GATK[71] (version 4.0.12).

**Variant calling and filtering on DNA.** We performed somatic variant calling using the software Mutect2 (ref. [72]) from GATK version 4.1.5.0, as implemented in our Nextflow workflow (https://github.com/IARCbioinfo/mutect-nf; release version 2.2b). Multiregion samples were processed jointly using the multisample calling mode of Mutect2. We called germline variants using Strelka2 (ref. [73]) version 2.9.10-0 using our Nextflow workflow (https://github.com/IARCbioinfo/mutect2-nf; release version 1.2a). Annotation was performed with ANNOVAR[74] (16 April 2018) using the GENCODE version 33 annotation, COSMIC version 90 and REVEL databases. To call somatic variants on tumor-only samples (72/115), a similar procedure was performed (Mutect2 tumor-only mode) but including further germline-filtering steps using a random forest classifier.

**CNV calling.** Somatic CNVs were called using the PURPLE software[75] version 2.52, as implemented in our Nextflow workflow (https://github.com/IARCbioinfo/purple-nf; version 1.0). We used a total of 57 matched WGS samples of MPM (including multiregion samples) for benchmarking the tumor-only mode of PURPLE. We ran PURPLE twice for each matched sample: first using the matched WGS normal/tumor pair as input and second using only the tumor WGS sample as input.

**Structural variant calling.** To identify somatic structural variants, including insertions, deletions, duplications, inversions and translocations, we built a consensus structural variants call set by integrating SvABA[76] version 1.1.0, Manta[77] version 1.6.0 and DELLY[78] version 0.8.3 calls with SURVIVOR[79] version 1.0.7. Somatic structural variants (minimum structural variant size = 50 base pairs) identified by at least two callers and single-caller predictions with a minimum read support of 15 pairs (including paired-end and split-read evidence) were included in the consensus set of each matched sample.

**RNA-seq**
**Sequencing.** RNA-seq was performed on 126 fresh frozen MPM samples in the Cologne Center for Genomics, of which 109 MPM samples belonged to the discovery cohort (Supplementary Tables 1–3). Libraries were prepared using the Illumina TruSeq Stranded mRNA Sample Preparation Kit (20020595; Illumina) and the pool was sequenced using an Illumina NovaSeq 6000 sequencing device and a paired-end 100-nucleotide protocol.

**Data processing.** The 126 raw read files from the MESOMICS cohort and the 21 files from the Iorio and colleagues[18] mesothelioma cohort (downloaded from the European Genome-phenome Archive (EGA) and Sequence Read Archive websites; datasets EGAS00001000828 and PRJNA523380, respectively) were processed in three steps using the RNA-seq processing workflow based on the Nextflow language and accessible at https://github.com/IARCbioinfo/RNAseq-nf (release version 2.3)[66]. Then, reads were realigned locally using ABRA2 (ref. [80]); (workflow https://github.com/IARCbioinfo/abra-nf; release version 3.0) and base quality scores were recalibrated using GATK (workflow https://github.com/IARCbioinfo/BQSR-nf; release version 1.1). Once processed, expression was quantified using StringTie software (version 2.1.2; Nextflow pipeline accessible at https://github.com/IARCbioinfo/RNAseq-transcript-nf; release version 2.2).

The raw read counts of the 59,607 genes in the expression data matrix, from the MESOMICS, TCGA and Bueno cohorts[3,4], from which we removed non-chimionaif samples, were normalized using the variance-stabilizing transform (vst function from R package DESeq2 version 1.14.1); this transformation enables comparisons between samples with different library sizes and different variances in expression across genes.

**DNA methylation**
**EPIC 850K methylation array.** Epigenome analysis was performed on 119 MPMs (Extended Data Fig. 1 and Supplementary Tables 1–3), two

technical replicates and three adjacent normal tissues. Epigenomic studies were performed at the IARC with the Infinium EPIC DNA methylation beadchip platform (Illumina) used for the interrogation of over 850,000 CpG sites (dinucleotides that are the main target for methylation).

**Data processing.** The resulting IDAT raw data files were preprocessed using the R packages minfi (version 1.34.0) and ENmix (version 1.25.1). Raw data were then normalized using functional normalization (function preprocessFunnorm; minfi), to reduce technical variation within the data, and probe removal steps were performed to ensure reliability and accuracy of the final dataset. This resulted in a normalized, filtered dataset of 781,245 probes for 139 samples. Finally, beta and $M$ values were extracted (functions getBeta and getM; minfi). Nine probes recorded $M$ values of $-\infty$ for at least one sample, and these values were replaced with the next lowest $M$ value in the dataset. The three normal tissues and one remaining technical replicate were then removed from the beta and $M$ matrices for the subsequent analyses. This resulted in 135 samples: 122 for discovery and an additional 13 for ITH analyses.

**CIMP index.** A CIMP index value was calculated for all samples as follows. The mean beta value across all probes located within CpG islands was calculated per sample, resulting in beta values for 24,891 and 24,924 CpG islands, MESOMICS (EPIC array), TCGA[4] and Iorio and colleagues[18] cell lines (HM450K array), respectively. The CIMP index was then calculated as the proportion of these 24,891 or 24,924 islands with ≥30% methylation (beta value ≥ 0.3) per sample.

**Integrative unsupervised analyses**
We performed four series of analyses with different subsets of samples: (1) discovery analyses with all of our discovery cohort (MESOMICS cohort; 120 samples), for which WGS, RNA-seq and/or 850K methylation array data were available; (2) and (3) replication analyses with the already published data from Bueno[3] (181 samples after exclusion of nonchemonaive samples) and Hmeljak and colleagues[4] (TCGA cohort; 73 samples in the curated list), respectively; (4) combined analyses integrating the MESOMICS, Bueno and TCGA cohorts[3,4] with a total of 374 samples; and (5) replication combining cell lines from the Iorio study[18] (for which whole-exome sequencing, expression arrays and RNA-seq, 450K methylation arrays and drug responses in the form of half-maximum inhibitory concentration scores are available (21 samples; 265 drugs)) and the de Reyniès[5] and Blum et al.[7] datasets (for which expression arrays and drug responses are available (38 samples; three drugs)). In addition, some single-omic analyses are also described in this section.

**Preprocessing of expression data.** We used normalized read count matrices (see the section 'RNA-seq') for subsets (1)–(4), encompassing 59,607 genes. Among these genes, those having less than one fragment per kilobase of exon per million mapped fragments (FPKM) difference across the samples were excluded from the unsupervised analyses. Also, to mitigate sex influence on the expression profiles, we removed genes from the sex chromosomes. For each analysis, the top 5,000 most variable genes were selected. Similarly, the 5,000 most variable genes from the normalized array expression of cell lines (see the section 'Processing of publicly available expression array processing' in Supplementary Methods) were selected. Whenever several probes were available for the same gene, the one with the highest intensity was selected.

**Preprocessing of methylation data.** DNA methylation was available for both the MESOMICS and TCGA cohorts. First, we extracted the $M$ values of the CpGs from the MESOMICS, TCGA[4], combined MESOMICS/TCGA and Iorio[18] cell line cohorts, respectively[81]. We excluded sex chromosome CpGs, CpGs that did not pass quality control (see the

section 'DNA methylation' in Supplementary Methods) and those having less than 0.1 beta value difference across the (1) 119, (3) 73, (4) 192 and (5) 59 samples. Based on this annotation, the CpG list representing the methylation data was divided according to their association with promoters, enhancers or the gene body using the EPIC 850K array manifest B5 (see the section 'Regional methylation analysis' in Supplementary Methods), resulting in three datasets, respectively named MethPro, MethEnh and MethBod. For each analysis and dataset, the top 5,000 most variable CpGs (calculated from $M$ values) were selected.

**Preprocessing of copy number changes.** Copy number change data were available for the MESOMICS, TCGA and MPM cell line cohorts. We assessed the global (total) and minor (minor) allele copy number states at the gene level using, respectively, the total (total) and minor (minor) copy number estimate given by PURPLE (see the section 'CNV calling') on the hg38 genome for the MESOMICS cohort and SNP array estimates downloaded from the Genomic Data Commons portal for the TCGA–MESO cohort[4] and from the Cell Model Passports portal for the MPM cell lines.

For the three analyses, the resulting value assigned to each gene is an average of the copy number estimate of the tumor by taking into account the tumor purity (purity) estimated by PURPLE. To avoid redundancy, genes with exactly the same resulting copy number value in all samples (because of their genome location proximity) were grouped as one single feature in the dataset. Only the genes or groups of genes altered in at least three samples were selected. To ensure continuity of the data, which is technically necessary for the algorithm, the copy number estimates were centered and scaled before being integrated into the MOFA algorithm. For consistency, somatic CNVs occurring on sex chromosomes were removed and the top 5,000 most variable genes or groups of genes were selected to be integrated.

**Preprocessing of genomic alterations data.** Somatic structural variants data were used only for integrative analyses (1) and (4), while somatic mutations were used in all analyses. Each gene, altered by somatic splicing, structural variants or exonic, damaging mutations (see the section 'Damaging variants and driver detection' in Supplementary Methods) was integrated in a common dataset. Of note, for missense mutations, we used the REVEL annotation included in ANNOVAR for predicting the pathogenicity of these variants and we used a 0.5 cut-off to restrict to the most likely damaging missense events. We also removed genes altered in fewer than three samples. For consistency, we selected genes in non-sex chromosomes, protein-coding or long noncoding RNA genes, and with expression greater than or equal to 0.01 fragment per kilobase of exon per million mapped fragments (FPKM) in at least one sample of the cohort, to be sure to include genes expressed in mesothelioma. We integrated the resulting datasets as a Boolean variable in the following analyses.

**Multiomic integrative analyses.** To provide an integrative low-dimensional summary of the molecular variation across the samples, we performed continuous latent factors identification using the software MOFA (R package MOFA2, version 1.7.0). Indeed, MOFA is able to integrate different molecular datasets (layers) by generating independent continuous variables, named latent factors, that explain most variation from the joint datasets. In total, we performed five analyses: (1) MOFA–MESOMICS ($n = 120$; Fig. 1 and Extended Data Fig. 1a); (2) MOFA–Bueno ($n = 181$; Extended Data Fig. 1c); (3) MOFA–TCGA ($n = 73$; Extended Data Fig. 1b); (4) MOFA–3 cohorts ($n = 374$; Extended Data Fig. 1d) and (5) MOFA–cell lines, as described above ($n = 59$; Supplementary Fig. 4). Additionally, we ran MOFA on our discovery cohort, including the ITH samples (MOFA–ITH; $n = 134$) to evaluate the ITH within MPM samples.

MOFA was performed independently for each analysis, setting the number of latent factors to ten (function runMOFA from the R

package MOFA2). A summary of all of these runs is given in Extended Data Figs. 1 and 2, Fig. 1 and Supplementary Figs. 1 and 4 and coordinates and proportions of variance explained for models (1)–(4) are given in Supplementary Tables 4–8, while those for MOFA–ITH are given in Supplementary Tables 50–52 and those for the cell lines (model (5)) are given in Supplementary Tables 23–26. A comparison with other multiomic methods is provided in Extended Data Fig. 10 (see section 'Multiomic integrative analyses details' in Supplementary Methods).

### Evolutionary tumor trade-off analyses

**Pareto task identification.** The Pareto front model was fitted to different sets of samples using the ParetoTI R package (https://github.com/vitkl/ParetoTI; release version 0.1.13), following the above-mentioned analyses (1)–(4), and additionally on two different kinds of molecular maps: using MOFA (restricting to LF1, LF2, LF3 and LF4) and using expression principal component analysis as technical validation (see the section 'RNA-seq'). In brief, the algorithm tries to find polyhedra by testing successively 1 to $n$ axes, adding them one after another in decreasing order of transcriptomic variance explained. For this technical reason, the MOFA latent factors were ordered as follows by decreasing transcriptomic variance explained: morphology factor (LF2), adaptive response factor (LF3), CIMP factor (LF4) and ploidy factor (LF1). For each number $n$ of axes used, ParetoTI identifies the position of the $n + 1 = k$ vertices (archetypes) in the molecular map defined, and we used 200 bootstraps, each taking 75% of the data to measure the variability in archetype position and infer archetype positions robust to outliers (function fit_pch_bootstrap with the parameters bootstrap = T and bootstrap_N = 200; see our code at https://github.com/IARCbioinfo/MESOMICS_data/blob/main/phenotypic_map/MESOMICS/PhenotypicMap_MESOMICS.md).

**Interpretation of tumor archetypes.** To further characterize the phenotype of each archetype, we used the proportion of each archetype for each sample estimated by ParetoTI. These proportions were used as continuous variables to further test the association between each archetype and clinical, epidemiological and morphological variables, as well as molecular data (Supplementary Tables 27–30).

More specifically, we inferred each archetype phenotype by performing IGSEA on the expression data. To do so, we used the ActivePathways R package (https://github.com/reimandlab/ActivePathways; release version 1.1.0), which is a tool able to integrate different sources of molecular variation to assess the enrichment of Gene Ontology terms by combining $P$ values from different association tests between sources and gene-level data. Here we integrated these proportions as different axes of molecular variation. We restricted the Gene Ontology terms to a minimum size of 20 genes and a maximum size of 1,000 genes as the default parameters of ActivePathways. To infer the pathways specifically altered in each archetype, we integrated the Pearson's $P$ value correlation of each gene from the expression matrix of 59,607 genes with the proportion from each archetype and we selected the pathways for which the enrichment source only corresponded to the tested archetype. We performed two kinds of analyses: one restricted to the genes positively correlated with the proportion (to obtain the upregulated pathways) and the other restricted to the negatively correlated genes (to identify the downregulated pathways).

### Reporting summary

Further information on research design is available in the Nature Portfolio Reporting Summary linked to this article.

### Data availability

The genome sequencing, RNA-seq and methylation data have been deposited in the EGA database, which is hosted at the European Bioinformatics Institute and Centre for Genomic Regulation under accession number EGAS00001004812. Because raw omics datasets

derived from humans are at risk of re-identification when combined with information from other public sources, access must be requested from the MESOMICS data access committee, as detailed at https://ega-archive.org/studies/EGAS00001004812. Minimum datasets of processed somatic alterations for genomic, transcriptomic and epigenomic data, sufficient to reproduce, interpret and extend our main results, are publicly available at https://github.com/IARCbioinfo/MESOMICS_data/tree/main/phenotypic_map/MESOMICS. A data note manuscript detailing all of the quality controls of the dataset is available at https://www.biorxiv.org/content/10.1101/2022.07.06.499003v1 (ref. [82]). TCGA whole-exome sequencing, RNA-seq and methylation array data are available from the Genomic Data Commons portal (TCGA–MESO cohort[4]). Whole-exome sequencing and RNA-seq data from the Bueno and colleagues cohort[3] are available from the EGA under accession number EGAS00001001563. Small variant lists, RNA-seq, expression array and methylation data for the Iorio and colleagues cohort[18] are available from the Gene Expression Omnibus (accession number GSE29354), EGA (accession number EGAS00001000828) and Sequence Read Archive (accession number PRJNA523380). Corresponding drug responses are available from the cancerrxgene.org website (https://www.cancerrxgene.org/downloads/drug_data?tissue=MESO; accessed July 2021). Expression array data for the de Reyniès and colleagues cohort[5] are available from the ArrayExpress platform (E-MTAB-1719) and corresponding drug response data are available from the supplementary material of Blum et al.[7]. All of the other data supporting the findings of this study are available within the article and its Supplementary Information files. Further information and requests for resources should be directed to and will be fulfilled by M.F. (follm@iarc.who.int). Source data are provided with this paper.

## Code availability
All bioinformatics pipelines are available at https://github.com/IARCbioinfo (see Methods for details about which pipelines and versions were used for each analysis). A detailed R notebook allowing reproduction of the MOFA and Pareto tumor task inference results for the MESOMICS cohort is available at https://github.com/IARCbioinfo/MESOMICS_data.

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

## Acknowledgements
The MESOMICS project is part of the Rare Cancers Genomics initiative (www.rarecancersgenomics.com) led by the Rare Cancers Genomics team at the IARC (https://www.iarc.who.int/teams-gem-rcg/). We thank the patients for donating tumor specimens. The human biological samples and associated data were obtained from the French MESOBANK. We also thank R. Argelaguet for advice on using MOFA, H. Begueret, N. Rousseau, D. Bozonnet, E. Wasielewski, G. Clapisson, C. Bonnetaud, K. Washetine, A. Lupo Mansuet, C. Cuenin and E. Clermont for their contribution to the biorepository. We acknowledge the American Association for Cancer Research and its financial and material support in the development of the AACR Project GENIE registry, as well as members of the consortium for their commitment to data sharing. Interpretations are the responsibility of the study authors. The results published here are in part based on data generated by the the TCGA Research Network (https://www.cancer.gov/tcga). We also thank the French National Mesothelioma Surveillance Program and Santé Publique France. This work has been funded by the French National Cancer Institute (PRT-K 2016-039 to L.F.-C. and M.F.) and the Ligue Nationale Contre le Cancer (LNCC 2017 and 2020 to L.F.-C. and M.F.). L.M. has a fellowship from the Ligue Nationale Contre le Cancer. This work also benefited from support from the France Génomique national infrastructure, funded as part of the Investissements d'Avenir program managed by the Agence Nationale de la Recherche (contract ANR-10-INBS-09). Other funding was provided by the Spanish Ministry of Science and Innovation (PID2019-105201RB-I00 to J.P.C.), the Instituto de Salud Carlos III, co-funded by the European Union (ERDF/ESF; Investing in Your Future), a Sara Borrell postdoctoral grant (CD19/00255 to A.I.-C.), the Spanish Ministry of Universities (predoctoral contract FPU18/02275 to R.B.-E.), the Junta de Andalucía (BIO-0139) and the Universidad de Córdoba-FEDER (UCO-202099901918904) (to J.P.C. and A.I.-C.), a GETNE2019 Research grant to J.P.C. and the CIBER Fisiopatología de la

Obesidad y Nutrición (CIBER is an initiative of the Instituto de Salud Carlos III). We finally thank the reviewers and the editor for taking the time to provide very useful and constructive feedback.

## Author contributions

L.F.-C. and M.F. conceived the study idea. L.F.-C., M.F., L.M., N.A., A.D.G. and A.S.-O. developed the study methodology. L.M., N.A., A.D.G., A.S.-O., A.G.-P., A.K., E.N.B. and C.V. developed software. L.M., N.A., A.D.G. and A.S.-O. validated the results. L.M., N.A., A.D.G., A.S.-O., A.G.-P., A.K., E.N.B., C.V., M.A., C.M., P.C., A.G.-P. and F.G.-S. performed the formal analyses. L.M., N.A., A.D.G., A.S.-O., A.G.-P., A.K., E.N.B., J.K., X.L., R.B.-E., A.I.-C., J.P.C., C. Giacobi, M.A., L.S., T.M.D., A.P., C.M. and P.C. performed the investigation. N.L.S., S.B., S.T.-E., F.D., M.B., M.-C.C., S.G.-C., D.D., C. Girard, V.H., P.H., J. Mouroux., C. Cohen, S. Lacomme, J. Mazieres, V.T.d.M., C.P., G.P., N.R., I.R., C.S., A.S., F.T., J.-M.V., A.G.S.I., R.O., V.M., S. Lantuejoul and F.G.-S. provided resources. C. Cuenin performed the methylation experiments. L.M., N.A., A.D.G., A.S.-O. and C.V. curated the data. L.F.-C., M.F., L.M., N.A., A.D.G. and A.S.-O. wrote the original draft of the manuscript. L.F.-C., M.F., L.M., N.A., A.D.G., A.S.-O., A.G.-P., A.K., D.J., H.H.-V., C. Caux, N.G., N.L.-B., L.B.A. and F.G.-S. reviewed and edited the manuscript. L.M., N.A., A.D.G and A.S.-O. created the visualizations of the results. L.F.-C., M.F. and N.A. supervised the project. L.F.-C., M.F., L.M., N.A., M.-C.M., A.B.-A., J.-F.D., J.A., P.N. and A.G. administered the project. L.F.-C., M.F. and N.A. acquired funding. L.F.-C., M.F., L.M., N.A. and A.S.-O. revised the manuscript.

## Competing interests

Where authors are identified as personnel of the IARC/WHO, the authors alone are responsible for the views expressed in this article and they do not necessarily represent the decisions, policy or views of the IARC/WHO. Where authors are identified as personnel of the Centre de Recherche en Cancérologie de Lyon, the authors declare no competing interests. A.S. participated in expert boards and clinical trials with AstraZeneca, Bristol-Myers Squibb, MSD and Roche. N.G. declares consultancy for and research support from Bristol-Myers Squibb, AstraZeneca, Roche and MSD. S Lantuejoul declares research support from AstraZeneca, Sanofi, Bristol-Myers Squibb, Janssen and Eli Lilly and has participated in expert boards for MSD and Bristol-Myers Squibb. D.D. declares research support from AstraZeneca. J. Mazieres declares consultancy for and research support from Roche, AstraZeneca, Bristol-Myers Squibb, MSD and Pierre Fabre. M.B. declares consultancy for and research support from AstraZeneca, Bristol-Myers Squibb and Amgen. I.R. participated in expert boards for AstraZeneca, MSD and Bristol-Myers Squibb. L.B.A. is a compensated consultant and has equity interest in io9. His spouse is an employee of Biotheranostics. L.B.A. is also an inventor on a US patent (10,776,718) relating to source identification by non-negative matrix factorization. E.N.B. and L.B.A. declare US provisional patent applications with the serial numbers 63/289,601 and 63/269,033. L.B.A. declares US provisional patent applications with the serial numbers 63/366,392, 63/367,846 and 63/412,835. C.M. is employed by and has equity interest in Owkin. C.M., P.C. and F.G.-S. are inventors on the US patent 17185924 'Systems and methods for mesothelioma feature detection and enhanced prognosis or response to treatment'. All other authors declare no competing interests.

## Additional information

**Extended data** is available for this paper at https://doi.org/10.1038/s41588-023-01321-1.

**Correspondence and requests for materials** should be addressed to Matthieu Foll or Lynnette Fernandez-Cuesta.

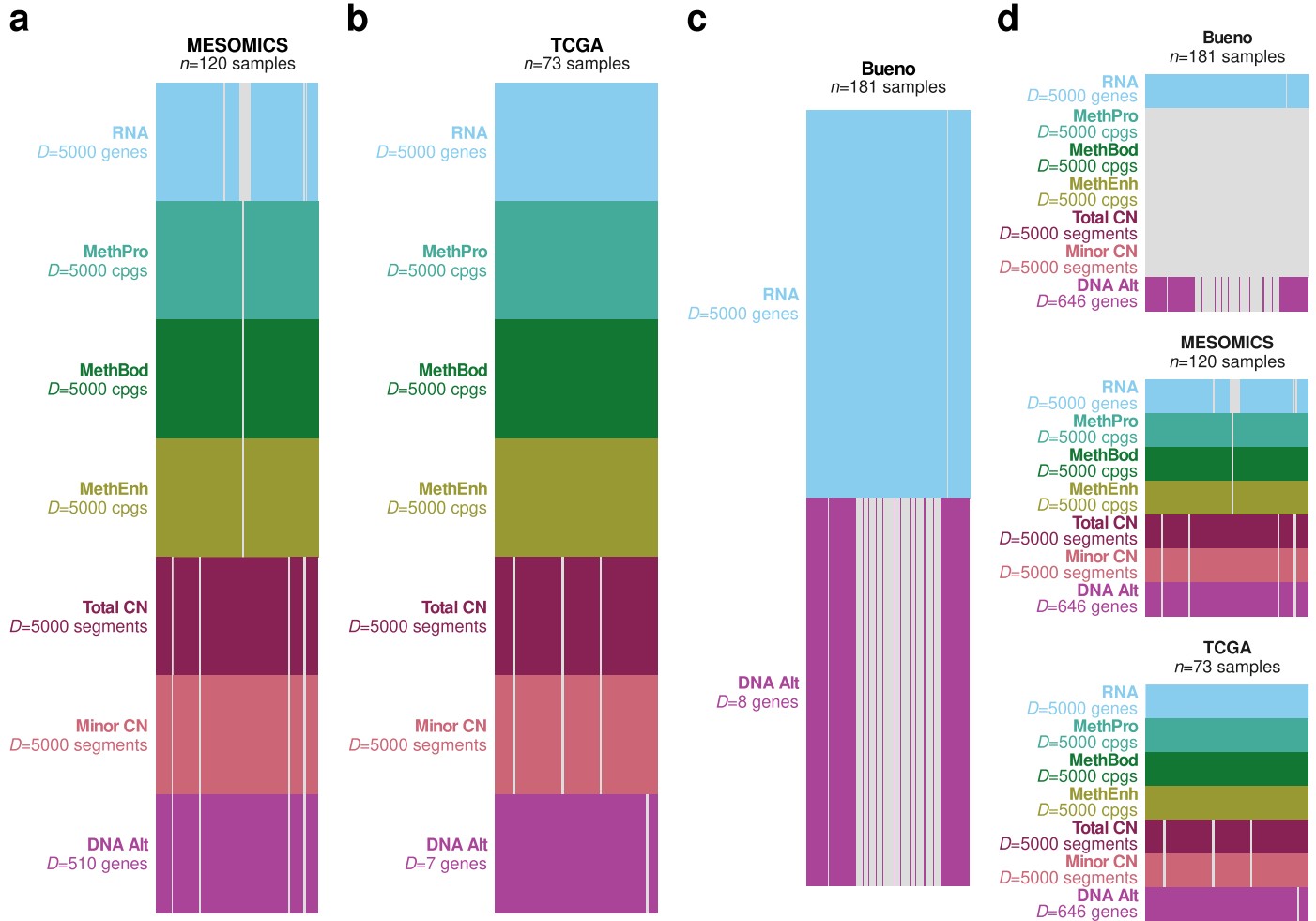

**Extended Data Fig. 1 | Overview of the MPM datasets for multiomic integration with MOFA.** Overview of the omic data sets integrated into multiomics factor analyses (MOFAs), for (**a**) a MOFA of the MESOMICS cohort ($n = 120$), (**b**) MOFA of the TCGA cohort ($n = 73$), (**c**) MOFA of the Bueno cohort ($n = 181$), and (**d**) MOFA of the 3 cohorts ($n = 120 + 73 + 181$). $D$ is the number of integrated omic features from genomic (rearrangements and mutations within *DNA Alt*; allele-specific copy number (CN) in *Total CN* and *Minor CN*), transcriptomic (*RNA*), and epigenomic data at promoter (*MethPro*), gene body (*MethBod*), and enhancer regions (*MethEnh*).

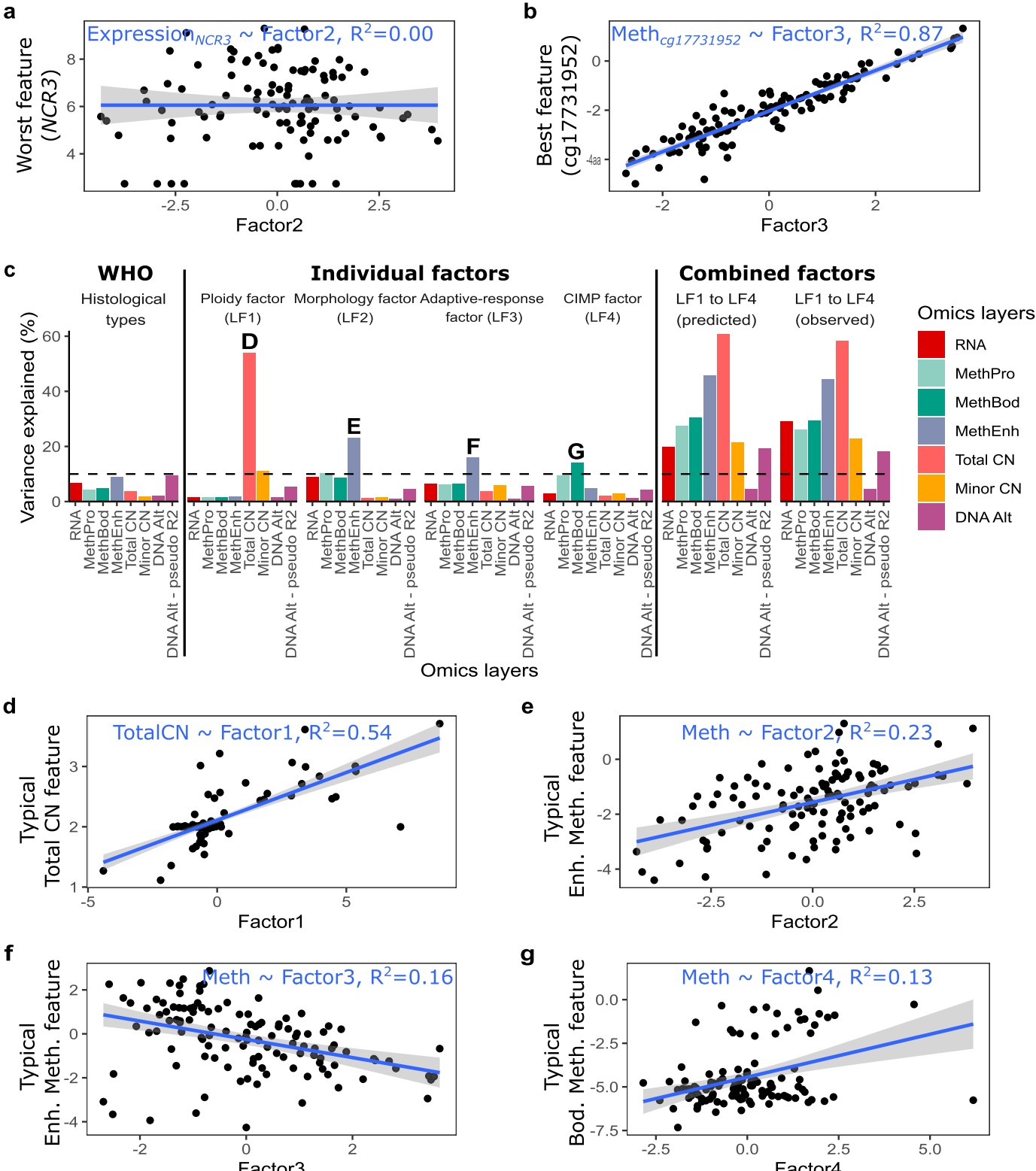

**Extended Data Fig. 2 | Proportion of interpatient variance explained by MOFA latent factors. a**) Example feature for which a latent factor explains 0% of interpatient variance (here factor 2 explains no variance at all in the expression of gene *NCR3*−$R^2$ = 0). **b**) Example feature for which a latent factor explains most of the variance (here factor 3 explains 87% of the variance of methylation site *cg17731952*−$R^2$ = 0.87). **c**) Variance explained by the three histopathological types, each latent factor (LF) independently, predicted total variance explained by all latent factors together if they were completely independent (LF1 to LF4

predicted), and actual variance explained by a model including the four latent factors as covariables (LF1 to LF4 observed). CIMP: CpG island methylator phenotype. **d**) Typical Total copy number (CN) feature associated with Factor 1. **e**) Typical Enhancer Methylation feature associated with Factor 2. **f**) Typical Enhancer Methylation feature associated with Factor 3. **g**) Typical Gene Body Methylation feature associated with Factor 4. In (**a**)-(**b**) and (**d**)-(**g**), the gray band corresponds to 95% confidence intervals.

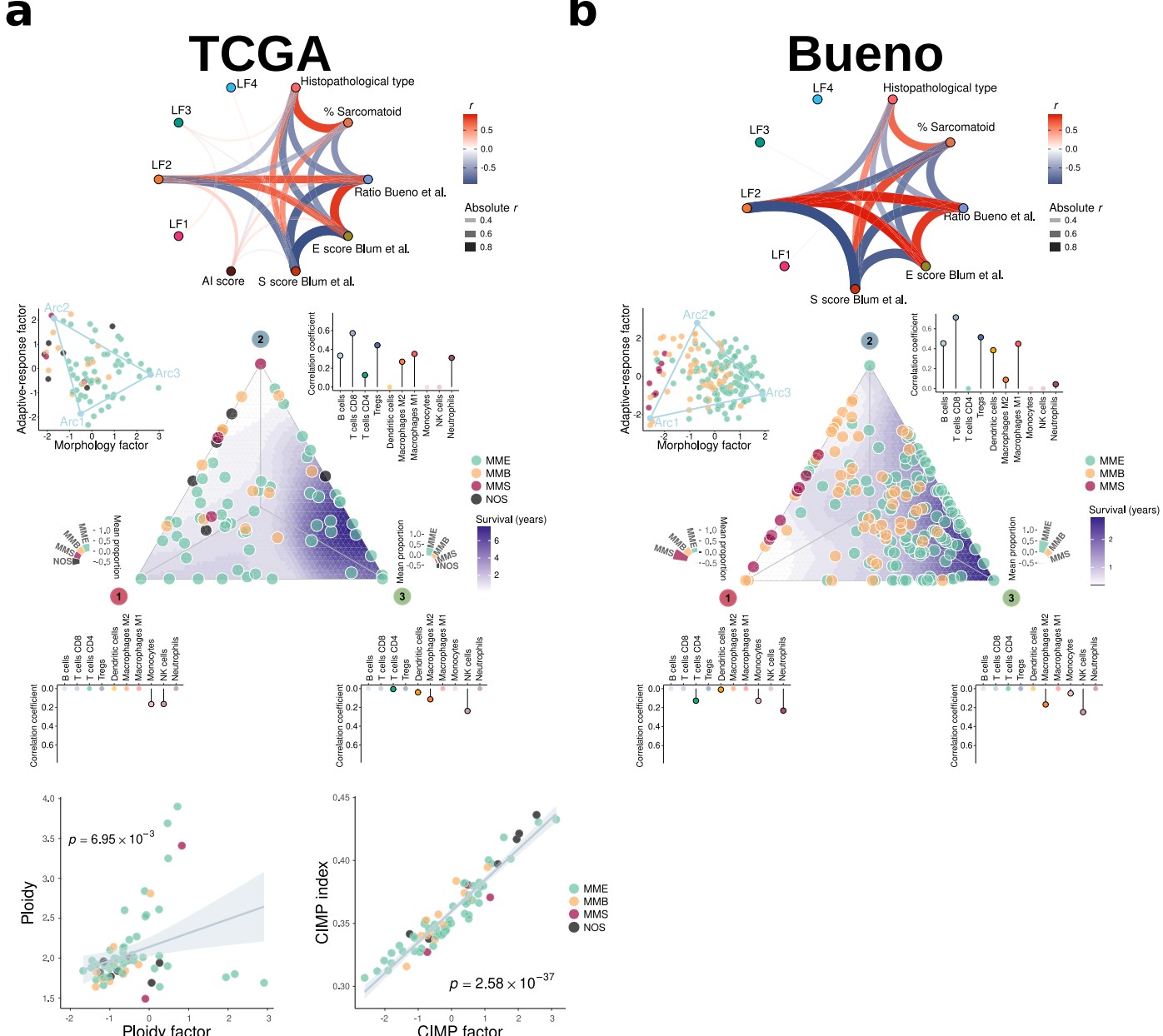

**Extended Data Fig. 3 | Replication of MOFAs latent factors and tumor tasks in major MPM cohorts.** MESOMICS MOFA latent factors and tumor task replicated in the TCGA (**a**) and Bueno (**b**) cohorts. The gray band corresponds to 95% confidence intervals. In (**a**), *P* values correspond to Pearson correlation tests (*n* = 73). MME: epithelioid; MMB: biphasic; MMS: sarcomatoid; NOS: malignant pleural mesothelioma not otherwise specified.

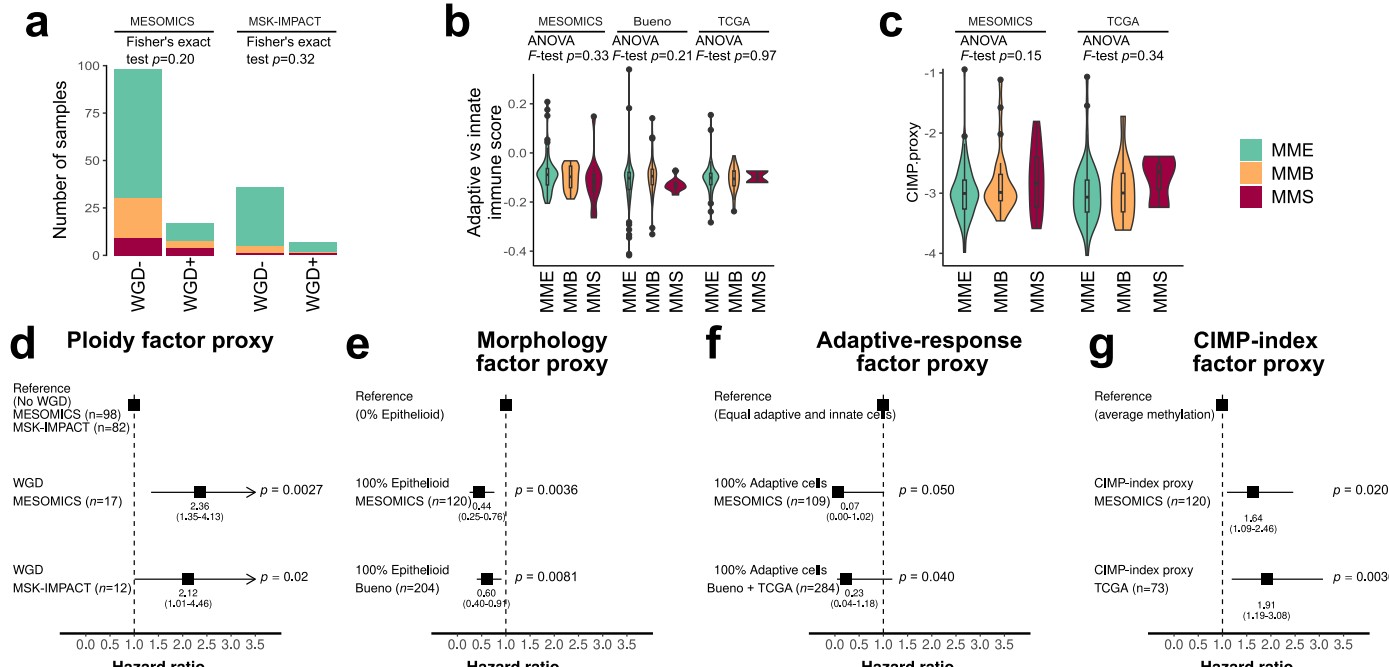

**Extended Data Fig. 4 | Replication of the prognostic value of MOFAs latent factors in other MPM cohorts. a)-(c)** Association between histological types and proxies for the MOFA latent factors. **a)** Association between whole genome-doubling (WGD) status and histological types in the MESOMICS and MSK-IMPACT cohorts, as determined by Fisher's exact tests. **b)** Association between the Adaptive versus innate response score and histological types, in the MESOMICS ($n = 120$) and Bueno ($n = 211$) and TCGA ($n = 73$) cohorts, as determined by ANOVA tests. **c)** Association between the CIMP-index proxy computed on a five-gene panel and histological types in the MESOMICS and TCGA cohorts, as determined by ANOVA tests. **d)-(g)** Forest plots of hazard ratios for overall survival showing the replication of latent factors' prognostic value, using a Cox proportional hazards model. In **(b)-(c)**, boxplots represent the median and interquantile range and whiskers the maximum and minimum values, excluding outliers. **d)** WGD status (proxy for the ploidy factor) in the MESOMICS and MSK-IMPACT

cohorts. **e)** Percentage of epithelioid estimated by pathologists from H&E slides (proxy for the morphology factor) in the MESOMICS and Bueno cohorts. **f)** Adaptive versus innate response score (proxy for the adaptive-response factor), in the MESOMICS and Bueno and TCGA cohorts, computed as the difference between the proportion of lymphocyte B and T-cells minus the proportion of macrophages, monocytes, and neutrophils, estimated from gene expression data (quanTIseq software). **g)** CIMP-index proxy computed on a five-gene panel (proxy for the CIMP-index factor), in the MESOMICS and TCGA cohorts. In all panels, $P$ values indicate the significance of tests. In **(d)-(g)**, squares correspond to estimated hazard ratios and segments to their 95% confidence intervals; tests in the MESOMICS cohort (discovery) are two-sided while tests in validation cohorts (MSK-IMPACT, TCGA, or Bueno cohorts) are one-sided, in the direction found in the discovery cohort.

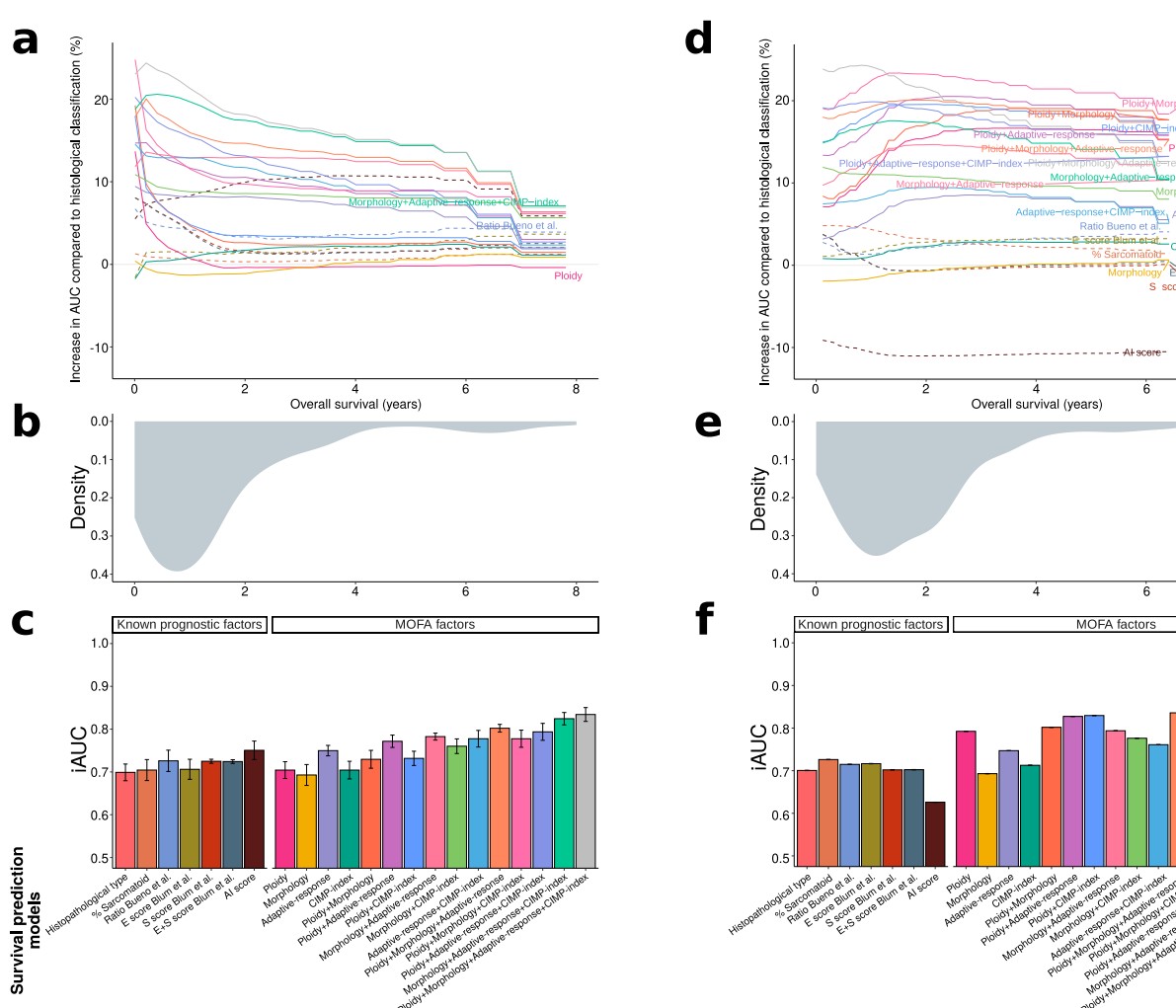

**Cross-validation on MESOMICS cohort**

**Bootstrapping on TCGA cohort**

**Extended Data Fig. 5 | Performance of MOFA factors to predict survival. a)** Increase in area under the curve (AUC) as a function of percentage of change compared to histological classification. **b)** Density of survival time within the MESOMICS cohort. **c)** Integral AUC (iAUC) of twenty-two Cox proportional hazards survival models based on: (i) the three histopathological types (MME, MMB, and MMS); (ii) the proportion of sarcomatoid content; (iii) the log2 ratio of CLDN15/VIM (C/V) expression proposed by Bueno and colleagues; (iv), (v) and (vi) the E score, S score, and combining both scores from Blum and colleagues, respectively; (vii) an Artificial Intelligence (AI) prognostic score; (viii-xi) the one-dimensional summary of molecular data using LFs as a continuous variable;

(xii-xvii), the two-dimensional summary of molecular data using either each combination of 2 LFs as continuous variables, respectively; (xviii-xxi), the three-dimensional summary of molecular data using each combination of 3 LFs as continuous variables; and (xxii), the four-dimensional summary of molecular data using all 4 LFs. Bars represent the mean values and error bars their standard error. Panels (**a-c**) present the out-of-sample accuracy within the MESOMICS cohort (4-fold cross-validation on *n* = 120 individuals), while (**d-f**) present the out-of-sample accuracy within the TCGA cohort (2000 bootstraps on *n* = 73 individuals). The model fit accuracy (no split between training and test sets) on MESOMICS and TCGA cohort are presented in Supplementary Table 17.

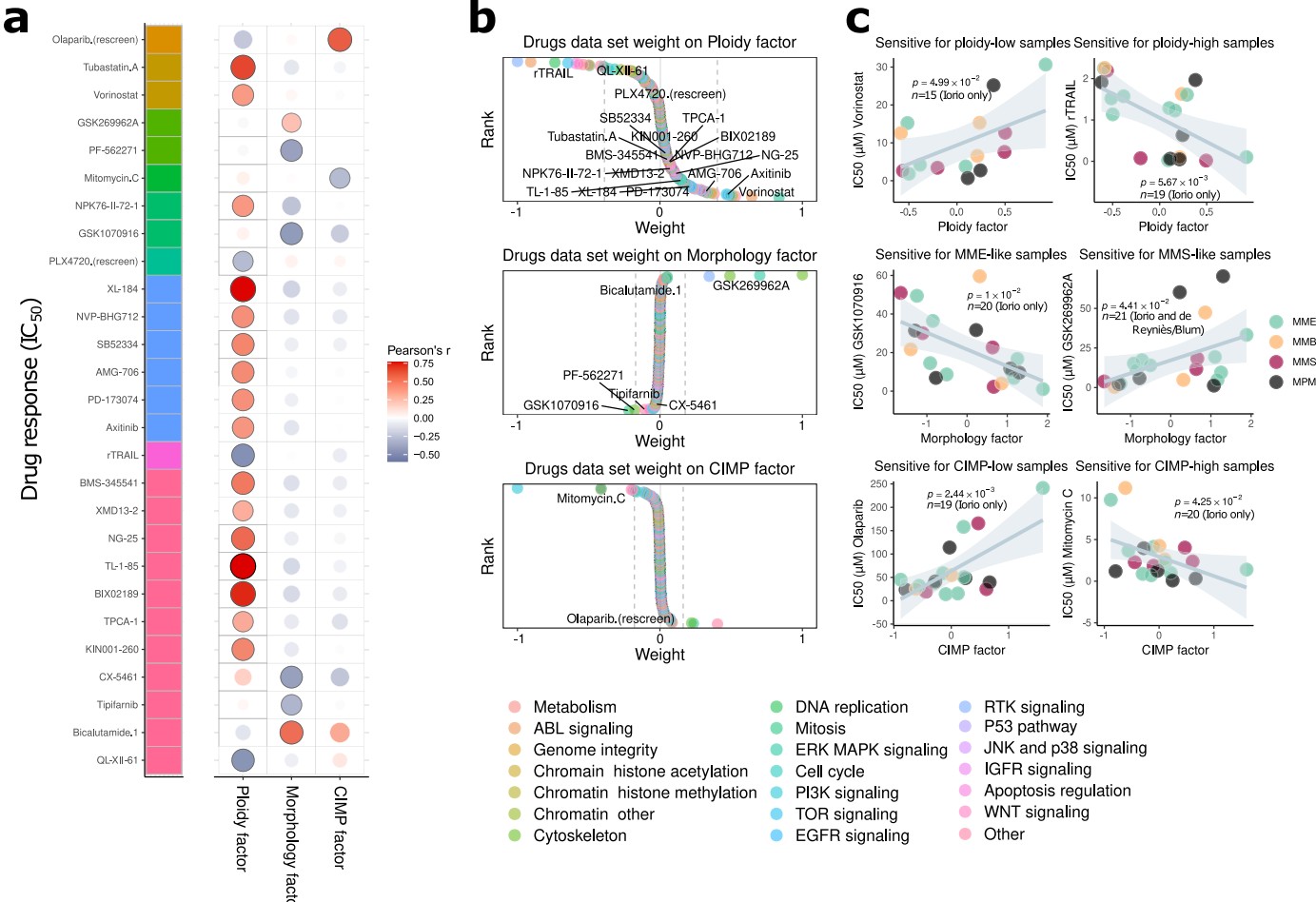

**Extended Data Fig. 6 | MOFA LFs of MPM cell lines and drug response. a)** Correlations between drug responses (measured by half maximal inhibitory concentration, IC50 in μM) and MOFA LFs of cell lines. Significant associations are annotated by black point border. **b)** Distribution of drug response weights from the Drugs data set, with drugs for which the response is significantly correlated with the given LF annotated in black. Targeted pathways are represented in (**a**) by a color bar (left), and in (**b**) by point colors. **c)** Correlations between representative drug responses significantly correlated with MOFA LFs from cell lines (left: negative correlations, right: positive associations). MPM: malignant pleural mesothelioma not otherwise specified. Gray bands correspond to 95% confidence intervals. Pearson correlation coefficients and the associated two-sided $P$ values are displayed in (**a**) and (**c**).

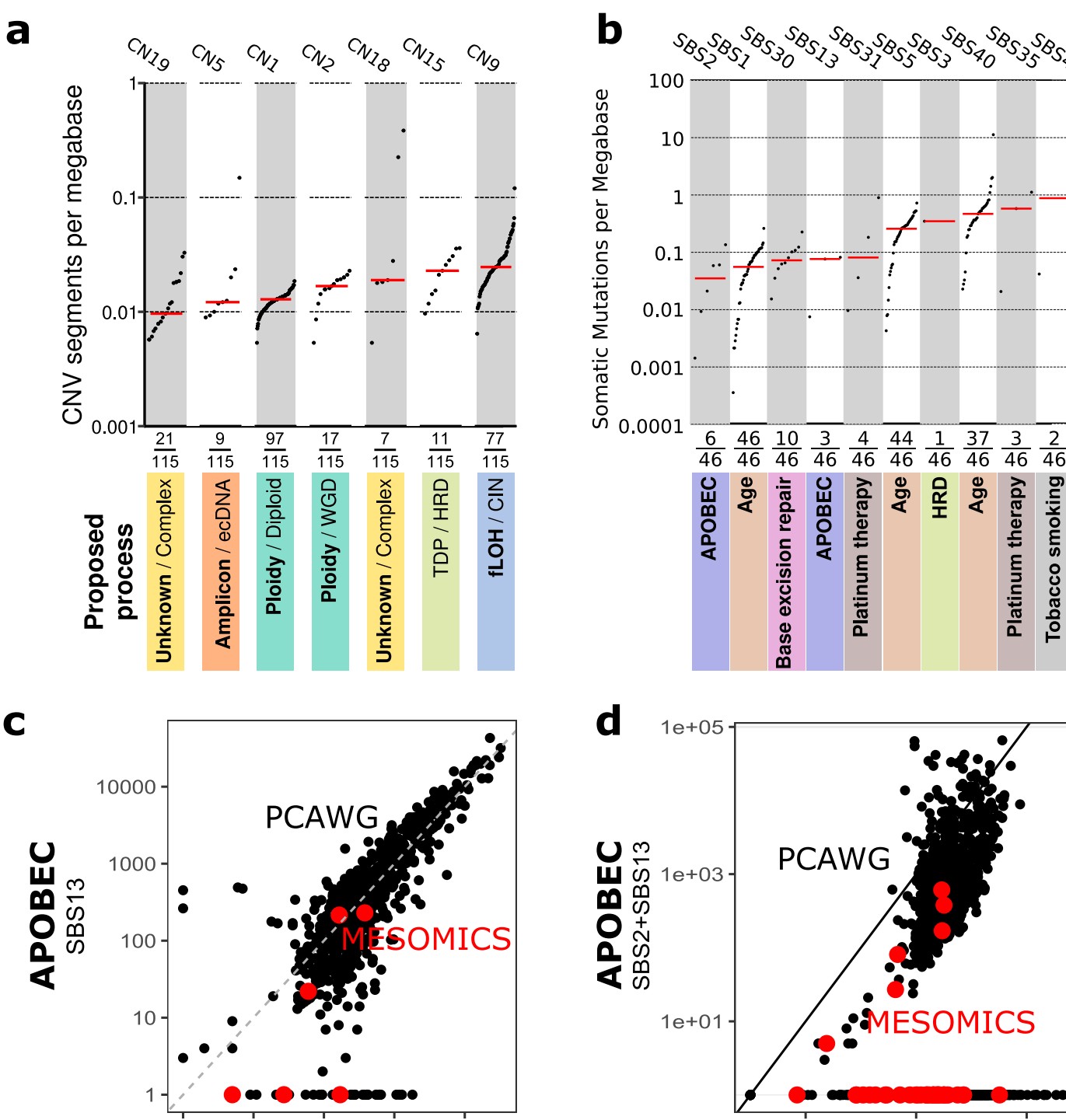

**Extended Data Fig. 7 | Tumor burden of mutational signatures.** Tumor Mutational Burden of **a**) 7 copy number signatures (*n* = 115 biologically independent samples) and **b**) 10 Single Nucleotide Variant Signatures detected in the MESOMICS cohort (*n* = 46 biologically independent samples). Note that although SBS40 is associated with age in many cancers, its etiology is still unknown. TDP: tandem duplicator phenotype; HRD: homologous recombination deficiency; fLOH: focal loss of heterozygosity; CIN: chromosomal instability.

**c**) Comparison of the tumor mutation burden (TMB, in number of mutations) of APOBEC signatures SBS2 and 13 in the MESOMICS cohort and in more than 2000 tumors from the Pan-Cancer Analysis of Whole Genomes (PCAWG) cohort. **d**) Comparison of the Relative TMB in number of somatic mutations of age-related signatures SBS1, SBS5, and SBS40, with that of APOBEC signatures SBS2 and 13 in the MESOMICS (red) and PCAWG cohorts (black).

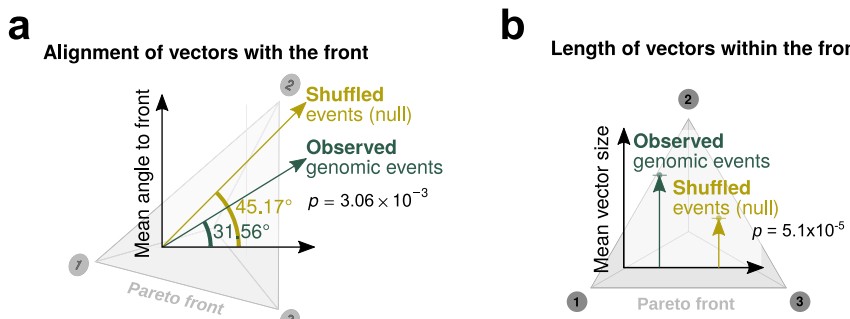

**Extended Data Fig. 8 | Test of the impact of genomic events on cancer task specialization. a**) Alignment of vectors with the Pareto front in degree (0°: perfectly aligned, 90°: completely orthogonal) and (**b**) length of the vector. *P* values correspond to two-sided Wilcoxon tests between observed and shuffled vector distributions.

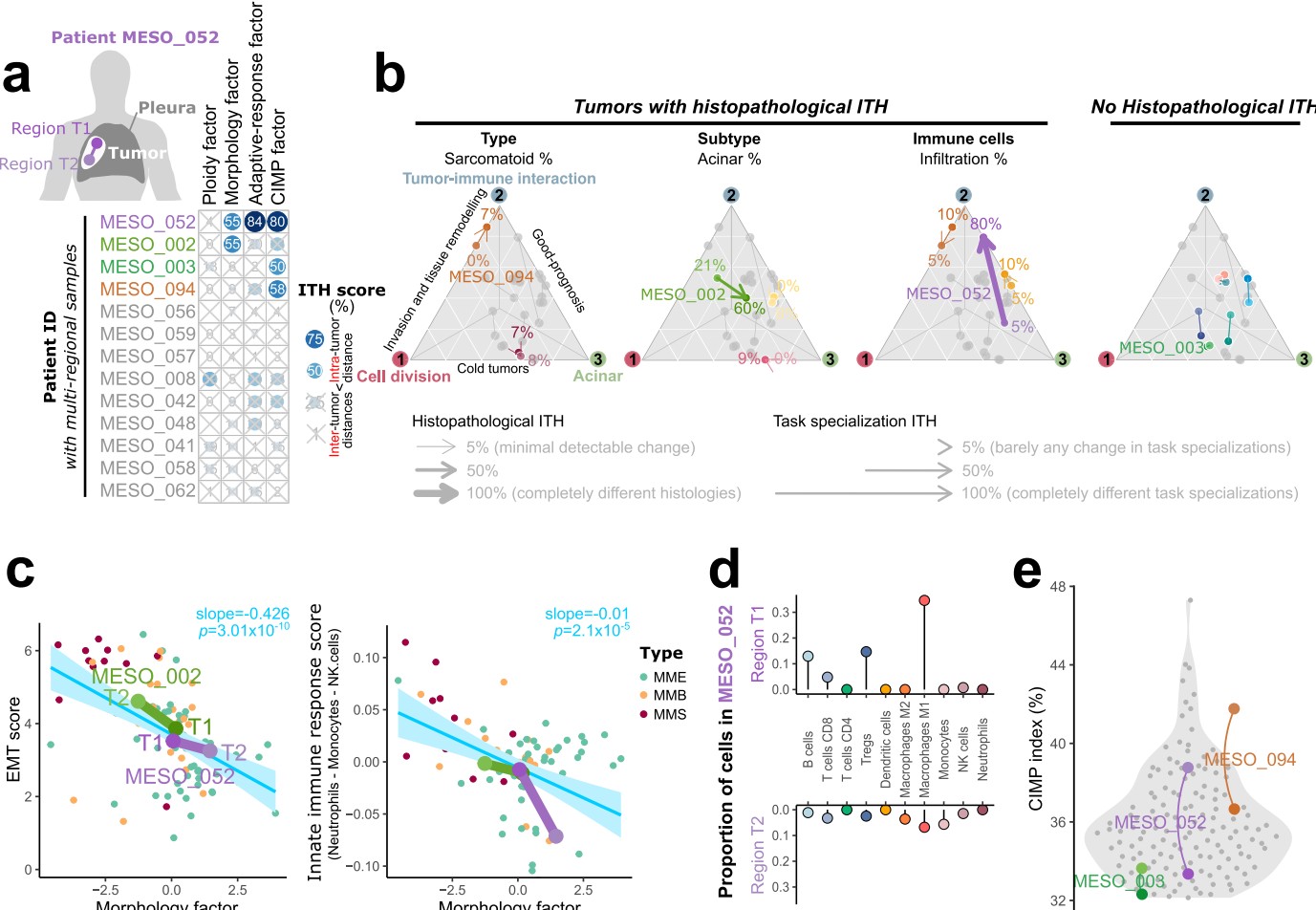

**Extended Data Fig. 9 | Multiomics intra-tumor heterogeneity (ITH) of 13 multiregion samples. a)** Intratumoral heterogeneity (ITH) score, ranging from 0% (no ITH) to 100% (ITH greater than the maximum observed intertumor heterogeneity in the cohort), for each sample (row) and each MOFA latent factor (column). The score is computed as the percentage of inter-tumor distances in a MOFA factor that are lower than the observed intratumor distance between regions. The four samples with ITH score greater than 50% are highlighted in color. **b)** Relationship between histopathological heterogeneity and cancer task specialization. Ternary plots depicting task specialization in three cancer tasks (see Fig. 2). For each histopathological feature, a colored arrow connects regions from tumors with differences in this feature. Numbers correspond to the percentage of this feature in the tumor as estimated by our pathologist. The right

ternary plot represents all samples with no histopathological ITH. **c)** Epithelial to mesenchymal transition (EMT) score and innate immune composition score as a function of MOFA's Morphology factor. Small points correspond to all samples from the MESOMICS cohort, and large points connected by segments to regions from the 3 patients with CIMP factor ITH highlighted in (**a**). Blue bands correspond to 95% confidence intervals, and *P* values to two-sided *t*-tests. **d)** Lollipop plot of the estimated proportion of immune cells in two regions of a sample with ITH in the adaptive-response factor highlighted in (**a**). **e)** CIMP index in regions of two tumors with substantial ITH in the CIMP factor highlighted in (**a**) (colored points connected by an arc), compared to that of the rest of the cohort (grey points).

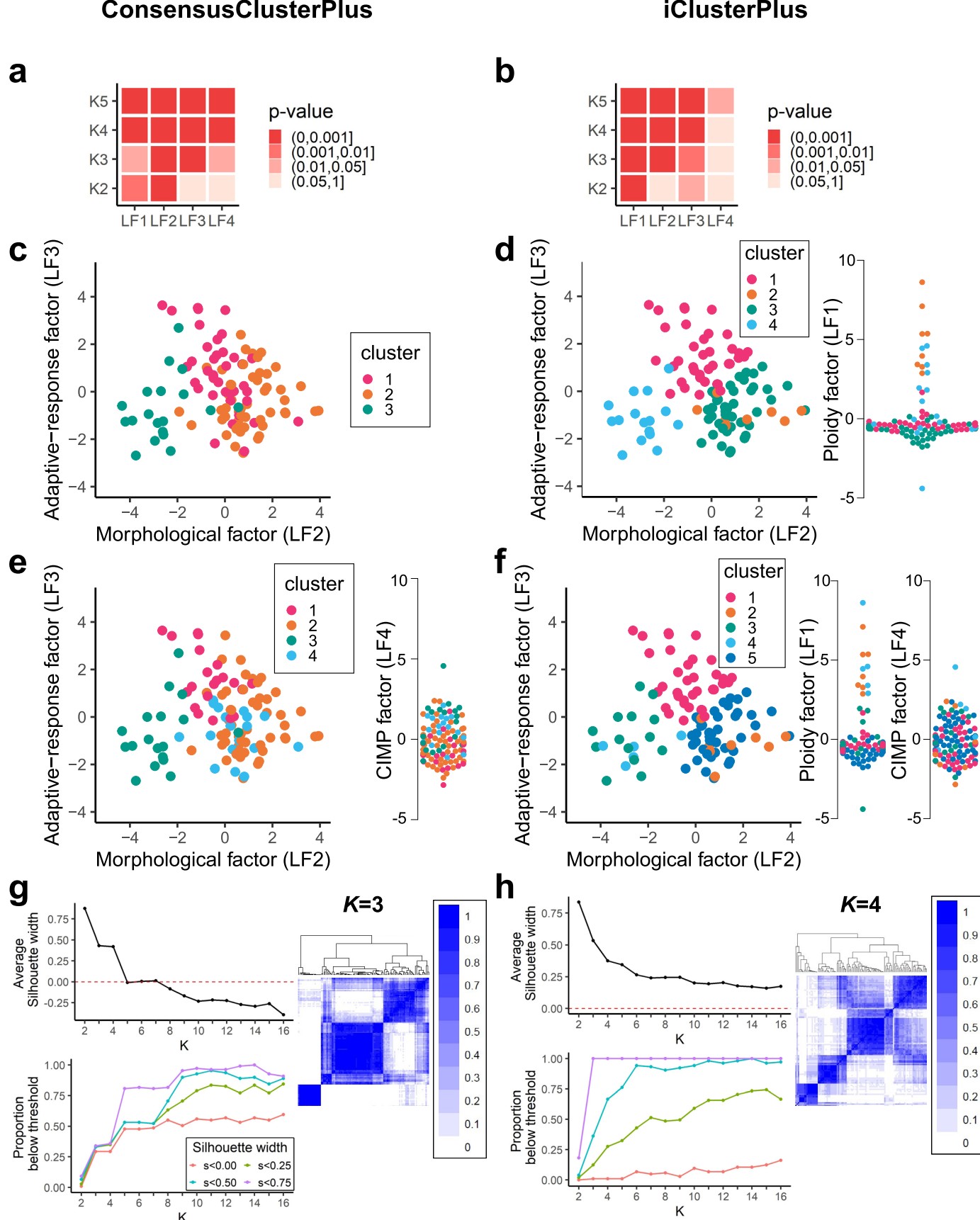

**Extended Data Fig. 10 | See next page for caption.**

**Extended Data Fig. 10 | Association between MOFA latent factors and the clusters identified by consensus clustering (a-d) and integrative clustering (e-h). a**) Kruskal-Wallis rank sum test significance (*P* value) between each *K* (row) and the LFs (column), for *K* from 2 to 5 from consensus clustering results and the first four LFs. **b**) Kruskal-Wallis rank sum test significance (*P* value) between each *K* (row) and the LFs (column), for *K* from 2 to 5 from integrative clustering results and the first four LFs. **c**) Consensus clustering results for *K* = 3. Samples are visualized in MOFA latent factor space of LF2 vs. LF3 and colored by the consensus clustering results. **d**) Integrative clustering results for *K* = 4. Samples are visualized in MOFA latent factor space of LF2 vs. LF3 and colored by the integrative clustering results. On the right, we show the samples in one-dimensional space of LF1 using beeswarm plot. **e**) Consensus clustering results for *K* = 4. **f**) Integrative clustering results for *K* = 5. Samples are visualized in MOFA

latent factor space of LF2 *vs*. LF3 and colored by the integrative clustering results. On the right, we show the samples in one-dimensional space of LF1 and LF4 using beeswarm plot. **g**) Top-left: average silhouette width for consensus clustering with different *K*. Bottom-left: proportion of samples below the selected silhouette width threshold for consensus clustering with different *K*. Right: consensus matrix heatmap for *K* = 3. Color gradient represents consensus values from 0–1. **h**) Top-left: average silhouette width for integrative clustering with different *K*. Bottom-left: proportion of samples below the selected silhouette width threshold for integrative clustering with different *K*. Right: heatmap of the frequencies of samples being clustered together among all clustering results using the set of iClusterPlus lambda values for *K* = 4. Color gradient represents consensus values from 0–1.

# Reporting Summary

## Statistics

For all statistical analyses, confirm that the following items are present in the figure legend, table legend, main text, or Methods section.

| n/a | Confirmed | |
|---|---|---|
| ☐ | ☒ | The exact sample size (*n*) for each experimental group/condition, given as a discrete number and unit of measurement |
| ☐ | ☒ | A statement on whether measurements were taken from distinct samples or whether the same sample was measured repeatedly |
| ☐ | ☒ | The statistical test(s) used AND whether they are one- or two-sided *Only common tests should be described solely by name; describe more complex techniques in the Methods section.* |
| ☐ | ☒ | A description of all covariates tested |
| ☐ | ☒ | A description of any assumptions or corrections, such as tests of normality and adjustment for multiple comparisons |
| ☐ | ☒ | A full description of the statistical parameters including central tendency (e.g. means) or other basic estimates (e.g. regression coefficient) AND variation (e.g. standard deviation) or associated estimates of uncertainty (e.g. confidence intervals) |
| ☐ | ☒ | For null hypothesis testing, the test statistic (e.g. *F*, *t*, *r*) with confidence intervals, effect sizes, degrees of freedom and *P* value noted *Give P values as exact values whenever suitable.* |
| ☐ | ☒ | For Bayesian analysis, information on the choice of priors and Markov chain Monte Carlo settings |
| ☒ | ☐ | For hierarchical and complex designs, identification of the appropriate level for tests and full reporting of outcomes |
| ☐ | ☒ | Estimates of effect sizes (e.g. Cohen's *d*, Pearson's *r*), indicating how they were calculated |

*Our web collection on statistics for biologists contains articles on many of the points above.*

## Software and code

Policy information about availability of computer code

| Data collection | No software was used for data collection. |
|---|---|
| Data analysis | WGS reads mapping: in-house workflow (github.com/IARCbioinfo/alignment-nf v1.0) that used bwa version 0.7.15, GATK version 4.0.12, samblaster version 0.1.24, sambamba version 0.6.6.<br>Variant calling: Mutect2 from GATK version 4.1.5, annotation with ANNOVAR version of April 16th 2018.<br>Copy number variant calling: PURPLE version 2.52 and Gistic2 version 2.20.23.<br>Structural variant calling: SVaba version 1.1.0, Delly version 0.8.3, Manta version 1.6.0, and SURVIVOR version 1.0.7.<br>Amplicon pattern calling: AmpliconArchitect version 1.2 and CNVkit version 0.9.7.<br>RNA-seq data processing: in-house workflow (github.com/IARCbioinfo/RNAseq-nf v2.3) that used Trim Galore (version 0.6.5 for expression quantification, and version 0.4.2 for alternative splicing analyses), STAR version 2.7.3a and StringTie version 2.1.2.<br>Quality control with FastQC version 0.11.9 and RSeQC version 3.0.1.<br>Fusion transcript discovery: realignment with STAR version 2.7.6a followed by Arriba version 2.1.0.<br>Immune contexture quantification: quanTIseq version of July 2020.<br>Statistical analyses with R version 4.0.3 |

For manuscripts utilizing custom algorithms or software that are central to the research but not yet described in published literature, software must be made available to editors and reviewers. We strongly encourage code deposition in a community repository (e.g. GitHub). See the Nature Portfolio guidelines for submitting code & software for further information.

# Data

Policy information about availability of data
All manuscripts must include a data availability statement. This statement should provide the following information, where applicable:

- Accession codes, unique identifiers, or web links for publicly available datasets
- A description of any restrictions on data availability
- For clinical datasets or third party data, please ensure that the statement adheres to our policy

The genome sequencing data, RNA-seq data, and methylation data have been deposited in the European Genome-phenome Archive (EGA) database, which is hosted at the EBI and the CRG, under accession number EGAS00001004812. Because raw -omics datasets derived from humans are at risk of re-identification when combined with information from other public sources, access must be requested to the MESOMICS data access committee (DAC) as detailed at https://ega-archive.org/studies/EGAS00001004812. Minimum datasets of processed somatic alterations for genomic, transcriptomic, and epigenomic data, sufficient to reproduce, interpret and extend our main results, are publicly available at https://github.com/IARCbioinfo/MESOMICS_data/tree/main/phenotypic_map/ MESOMICS. A data note manuscript detailing all quality controls of the dataset is available at https://www.biorxiv.org/content/10.1101/2022.07.06.499003v16. TCGA whole-exome sequencing, RNA-seq, and methylation array data are available from the GDC portal (TCGA-MESO cohort), the whole-exome sequencing and RNA-seq data from the Bueno and colleagues cohort are available from the European Genome-phenome Archive, EGA:EGAS00001001563. Small variants lists, RNA-seq, expression array, and methylation data for the Iorio and colleagues cohort are available from the GEO (GSE29354), EGA (EGAS00001000828), and SRA (PRJNA523380) websites, and corresponding drug responses are available from the cancerrxgene.org website (https://www.cancerrxgene.org/downloads/ drug_data?tissue=MESO; accessed July 2021). Expression array data for the de Reyniès and colleagues cohorts are available from the ArrayExpress platform (E-MTAB-1719), and corresponding drug response data from the supplementary material of Blum et al. All the other data supporting the findings of this study are available within the article and its supplementary information files.

# Field-specific reporting

Please select the one below that is the best fit for your research. If you are not sure, read the appropriate sections before making your selection.

☒ Life sciences        ☐ Behavioural & social sciences        ☐ Ecological, evolutionary & environmental sciences

For a reference copy of the document with all sections, see nature.com/documents/nr-reporting-summary-flat.pdf

# Life sciences study design

All studies must disclose on these points even when the disclosure is negative.

| | |
|---|---|
| Sample size | This study first includes an unsupervised analysis of genomes, RNA-seq, and methylation data using group factor analysis (software MOFA). For such multivariate analyses, no simple power calculations are available, but recommendations for stable latent factors and weights such that the results from the sample can accurately be generalized to the population suggest n>100 (Saccenti and Timmerman J.Proteom. Res. 2016), which is in line with the sample size in our study (total of n=120). This also ensured that the Pareto task inference relying on these factors had a sufficient sample size.<br>All omic and clinical data were available for at least n=100 samples, ensuring that correlation tests have a power of 80% to detect non-zero coefficients r>0.277 given a type I error rate of 5% (Hulley et al. Lippincott Williams & Wilkins 2013), and thus ensuring that downstream analyses such as gene set enrichment analysis relying on correlation test p-values are also properly powered.<br>Combining the MESOMICS, Bueno, and TCGA cohorts allows to reach n=300 samples with exonic sequencing data, allowing to detect recurrent alterations with a prevalence of 1% in malignant pleural mesothelioma in at least 2 samples with a power of 80% (based on the binomial distribution). |
| Data exclusions | Pre-established exclusion criteria were as follows: samples were excluded if they were derived from metastasis or a participant who had undergone chemotherapy for the treatment of malignant pleural mesothelioma. These samples were excluded as we aimed to identify whether particular molecular patterns or profiles in primary tumours were associated with disease progression and patient prognosis, therefore we examined only the molecular profile of primary untreated mesothelioma samples obtained at diagnosis. |
| Replication | We replicated our main findings independently on three external datasets: two of similar malignant pleural mesothelioma cohorts of sizes n= 181 from Bueno et al. Nat. Genet. 2016 and n=73 from Hmeljak et al. 2019, and one of n=59 cancer cell lines from de Reynies et al. Clin. Cancer Res. 2014 and Iorio et al. Cell 2016. |
| Randomization | Samples were assigned a histopathological class through central pathological review. We assessed the importance of covariables like age, sex, smoking status, and asbestos exposure in multivariate analyses using regression analyses. We controlled for these variables in statistical analyses by adding them as covariables in the regressions when appropriate. |
| Blinding | The investigators were not blinded to the histopathological class during the unsupervised analyses, but this was unnecessary because these analyses do not take into account any sample information except the genomic alterations, gene expression, and methylation levels. The investigators could not be blinded to the subsequent differential analyses, which required knowledge of the group membership for each sample. |

# Reporting for specific materials, systems and methods

We require information from authors about some types of materials, experimental systems and methods used in many studies. Here, indicate whether each material, system or method listed is relevant to your study. If you are not sure if a list item applies to your research, read the appropriate section before selecting a response.

## Materials & experimental systems

| n/a | Involved in the study |
|---|---|
| ☐ | ☒ Antibodies |
| ☒ | ☐ Eukaryotic cell lines |
| ☒ | ☐ Palaeontology and archaeology |
| ☒ | ☐ Animals and other organisms |
| ☐ | ☒ Human research participants |
| ☒ | ☐ Clinical data |
| ☒ | ☐ Dual use research of concern |

## Methods

| n/a | Involved in the study |
|---|---|
| ☒ | ☐ ChIP-seq |
| ☒ | ☐ Flow cytometry |
| ☒ | ☐ MRI-based neuroimaging |

## Antibodies

| Antibodies used | Santa Cruz BAP1 antibody (cloneC-4) catalog number sc-28383 |
|---|---|
| Validation | BAP1 antibody was purchased from Santa Cruz (catalog number sc-28383). Immunohistochemistry was performed on FFPE tumor sections 3μm tissue sections after pretreatment using a Ventana's Immunostainer (BenchMark Ultra). After pre-treatment (CC1 solution), primary antibody was applied (dilution 1:50) during 30 min. The detection kit ultra-view (Universal DAB Detection kit, Ventana) was used following manufacturer's recommendations. BAP1 (clone C4) (Santa Cruz: dilution one to 50) nuclear staining was considered positive (when nuclear expression was retained) or negative (complete loss of staining of all tumor cells with a positive internal control on the slides [fibroblast, lymphocytes, etc.]). |

## Human research participants

Policy information about studies involving human research participants

| Population characteristics | Age: median of 67.5<br>Sex: 32 females and 88 males<br>n = 120 chemonaive at the moment of sample collection and three no chemonaive removed from the analyses.<br>The samples used in this study belong to the virtual biorepository French MESOBANK, whose guidelines include obtaining the informed consent from all subjects and no participant compensation. |
|---|---|
| Recruitment | There has not been a prospective recruitment of patients specifically done for this study. The biological specimens were previously collected for clinical routine and stored in the biobanks of the collaborative hospitals, which made the de-identified samples available for research. |
| Ethics oversight | International Agency for Research on Cancer ethics committee |

Note that full information on the approval of the study protocol must also be provided in the manuscript.

