## [Peer Review File · Nature Genetics]

Peer Review Information

Manuscript Title: Multi-omic analysis of malignant pleural mesothelioma identifies molecular axes and specialized tumor profiles driving inter-tumor heterogeneity

Corresponding author name(s): Dr Lynnette Fernandez-Cuesta

Reviewer Comments & Decisions:

Decision Letter, initial version:
--

Dear Dr Fernandez-Cuesta,

Your Article, "Whole-genome sequencing and multi-omic integrative analyses reveal novel axes of molecular variation and specialized tumor profiles in Malignant Pleural Mesothelioma" has now been seen by 3 referees. You will see from their comments copied below that while they find your work of considerable potential interest, they have raised quite substantial concerns that must be addressed. In light of these comments, we cannot accept the manuscript for publication, but would be very interested in considering a revised version that addresses these serious concerns.

We hope you will find the referees' comments useful as you decide how to proceed. If you wish to submit a substantially revised manuscript, please bear in mind that we will be reluctant to approach the referees again in the absence of major revisions.

You'll see that the reviewers are broadly supportive of the work and find the analysis to be of interest. However, they lay out a series of concerns about the analysis and the interpretation of the work which will all need to be addressed. You'll see that the lack of normal control samples for the majority of the WGS set has been flagged as a limitation. We appreciate that you are likely unable to provide said samples at this late stage, but you will need to convince your reviewers that their exclusion does not irreparably undermine the strength of your conclusions, and the overall value of the dataset. Reviewer #1 has asked for functional work to confirm some of your findings. We agree that these data would strengthen the paper, and we would encourage you to add these data if possible. However, the absence of these data will not preclude our interest in the paper. Their other comments must be addressed.

Reviewer #2 has raised a number of technical issues. They've also made requests for clarifications and the provision of missing details. These should all be addressed in full. Reviewer #3 has expressed doubt that your classification system represents a true advance over the current (histology-based) system. This is an important point as 'buy-in' from the mesothelioma community will underpin the paper's eventual success. As such, please address these concerns textually. They have also

highlighted the lack of a validation cohort. Again, we appreciate that you are unlikely to have access to a new, suitably powered cohort at this point in the process. However, this comment will need to be addressed either textually, or through splitting your cohorts to create a discovery and validation grouping (if possible). Their other points should be addressed.

If you choose to revise your manuscript taking into account all reviewer and editor comments, please highlight all changes in the manuscript text file. At this stage we will need you to upload a copy of the manuscript in MS Word .docx or similar editable format.

*2) If you have not done so already please begin to revise your manuscript so that it conforms to our Article format instructions, available [here](http://www.nature.com/ng/authors/article_types/index.html). Refer also to any guidelines provided in this letter.

[redacted]

If you wish to submit a suitably revised manuscript we would hope to receive it within 6 months. If you cannot send it within this time, please let us know. We will be happy to consider your revision so long as nothing similar has been accepted for publication at Nature Genetics or published elsewhere. Should your manuscript be substantially delayed without notifying us in advance and your article is eventually published, the received date would be that of the revised, not the original, version.

Please do not hesitate to contact me if you have any questions or would like to discuss the required

revisions further.

Nature Genetics is committed to improving transparency in authorship. As part of our efforts in this direction, we are now requesting that all authors identified as 'corresponding author' on published papers create and link their Open Researcher and Contributor Identifier (ORCID) with their account on the Manuscript Tracking System (MTS), prior to acceptance. ORCID helps the scientific community achieve unambiguous attribution of all scholarly contributions. You can create and link your ORCID from the home page of the MTS by clicking on 'Modify my Springer Nature account'. For more information please visit please visit www.springernature.com/orcid.

Thank you for the opportunity to review your work.

Sincerely,

Safia Danovi
Editor
Nature Genetics

Referee expertise:

Referee #1: mesothelioma genomics

Referee #2: lung cancer genomics

Referee #3: lung cancer genomics

Reviewers' Comments:

Reviewer #1:

Remarks to the Author:

This is an impressive WGS study of MPM with very sophisticated bioinformatic analyses. While the dataset represents a invaluable resource, there are few actionable findings and the few interesting observations are not followed with functional data.

1. many of the q values are not impressive, i.e. barely below 0.05
2. the authors describe CIMP in a substantial minority of MPM. While methylation of the mesothelin promoter was noted in the TCGA study, a CIMP phenotype was not called in that dataset. More detail is needed on this observation. In CRC, CIMP overlaps partially with MSI. A comment is needed on MSI and TMB in the CIMP subset of MPM. That would make this observation potentially actionable.
3. the authors describe HRD in 23% of MPM. This is an observation that begs for functional validation in terms of PARPi sensitivity.
4. p.8: LATS2, not LAST2
5. the authors state that " the higher rate of chromothripsis in the Acinar phenotype could be a result of BAP1 inactivation known to impair DNA repair" but Fig 3A shows a fairly obvious trend to mutual exclusivity between BAP1 alterations and chromothripsis.

6. p.9: the statement that "The specialization of tumors was influenced by early genomic events. WGD, TERT amp and copy neutral LOH may have occurred more than ten years before diagnosis, both concomitantly with and subsequent to asbestos exposure" is based on data that are either not convincing or poorly presented (Fig 5D).
7. Fig 3A: it would be very unusual for MTAP to undergo homozygous deletion without concurrent CDKN2A homozygous deletion. The authors should review these cases for undervalued CDKN2A deletion.
8. Fig 4A shows 3 SETDB1 mutated samples but the authors report only one GNH case. Please discuss.
9. Fig 4A reveals that a concerning proportion of the WGS samples had no matched normal.

Reviewer #2:

Remarks to the Author:

This is the largest WGS analysis of MPM to date and the authors used unusual approaches for the analyses. In my opinion, the analyses require major revisions and validation (or at least comparisons of results) using alternative approaches. However, there are some important limitations that cannot be fully addressed even with major revisions (first of all, the lack of germline material as reference for most tumors).

Reviewer #3:

Remarks to the Author:

The authors report findings from whole genome sequencing and multi-omic integrative analyses of tumor tissues from patients with malignant pleural mesothelioma. The authors state that ploidy, adaptive immune response, CpG island methylation along with histological subtypes explain the variation in the phenotypic behavior in this disease with nearly uniform poor outcomes. State of art tools from sequencing to analyses are used from the French Mesobank source (predominantly). Potentially, this approach could improve the current solely histology-centric classification. The paper would be of great interest to the community of mesothelioma researchers and clinicians given that this body of work represents the largest data to date from whole-genome sequencing. I commend the authors for this remarkable work.

1. It is not clear what cohort was used for various analyses. There are multiple cohorts- MESOMICS, Discovery (with and without ITH), and other published cohorts, TCGA, Bueno, in addition to cell lines. Most of the confusion centers around MESOMICS and Discovery cohorts. What is the Discovery cohort and what constituted this cohort (demographics, histology, etc)
2. There should be a clear description of the cohorts (in the written part of the manuscript) outlining the treatment received (surgery, systemic therapy) and follow up for the Meso cohort and as well as the Discovery Cohort
- 3 The first section titled integrative multi-omics analyses uncover novel axes of molecular variation is hard to follow. It is hard to understand how much the latent factors (LF) 1,3, 4 add to the histological factors? Is one histology more heterogeneous in terms of LF than the other histology? Despite its shortcomings, histological classification is the current and only standard in the clinic for treatment decision-making (surgery and the choice of initial systemic therapy, for example). I have a hard time

understanding what these three (excluding the morphology factor) contribute to the "molecular variation"

4. Results shown in Figure 1 e regarding the prognostic value of LF1-4 would have been stronger if there is validation in an independent cohort. Those findings did not hold up in the larger subtype of the epithelioid type where only LF3 appeared to be significant (Fig S5B). This should be highlighted. Unfortunately, it is hard to assess the sarcomatoid histological subtype without adequate numbers. Once again, one cannot draw definite conclusions in the absence of a validation cohort

Author Rebuttal to Initial comments

RESPONSE TO THE EDITOR (SD)

SD: You'll see that the reviewers are broadly supportive of the work and find the analysis to be of interest. However, they lay out a series of concerns about the analysis and the interpretation of the work which will all need to be addressed.

LFC: We have addressed all their comments below, as largely as possible, to cover any possible concerns.

SD: You'll see that the lack of normal control samples for the majority of the WGS set has been flagged as a limitation. We appreciate that you are likely unable to provide said samples at this late stage, but you will need to convince your reviewers that their exclusion does not irreparably undermine the strength of your conclusions, and the overall value of the dataset.

LFC: We completely understand the concerns of the reviewers and editor regarding the lack of matched normal for many of the samples in our series; however we were very much aware of this limitation from the very beginning of the project and we have taken the necessary steps to limit the impact of this lack of normals on achieving the objectives of our study. We explain all of this in the response to *Reviewer #1 - Comment 9* and *Reviewer #2* below.

SD: Reviewer #1 has asked for functional work to confirm some of your findings. We agree that these data would strengthen the paper, and we would encourage you to add

these data if possible. However, the absence of these data will not preclude our interest in the paper. Their other comments must be addressed.

LFC: Due to the specific expertise of our team, and the lack of infrastructure, we have not been able to run such functional analyses in-house. However, we have used publically-available MPM cell line data (Iorio *et al.* 2016 and de Reyniès *et al.* 2014) to address as much as possible the reviewer's point. We have also addressed all his/her other comments.

SD: Reviewer #2 has raised a number of technical issues. They've also made requests for clarifications and the provision of missing details. These should all be addressed in full.

LFC: We have addressed all the technical concerns of *Reviewer #2*, as well as the requests for clarifications, that were related to (i) the relationship between the methods used here and those used in previous genomic characterization of mesothelioma and other tumors, and (ii) the lack of matched normal tissue/blood for some samples.

SD: Reviewer #3 has expressed doubt that your classification system represents a true advance over the current (histology-based) system. This is an important point as 'buy-in' from the mesothelioma community will underpin the paper's eventual success. As such, please address these concerns textually. They have also highlighted the lack of a validation cohort. Again, we appreciate that you are unlikely to have access to a new, suitably powered cohort at this point in the process. However, this comment will need to be addressed either textually, or through splitting your cohorts to create a discovery and validation grouping (if possible). Their other points should be addressed.

LFC: We have addressed textually all the concerns of the reviewer regarding the importance of our discoveries, and added analyses to validate our results in publicly available cohorts. We have also improved our description of the methods and results and now mention explicitly which results could be validated using external cohorts, and if none was available, what steps we took to validate our results.

Finally, we would like to thank the editor and reviewers for highlighting the following features:

(i) that our WGS MPM tumor dataset ($n=115$) represents the largest to date (in comparison to WES only in *Hmeljak et al. Cancer Discov. 2018*, $n=20$ WGS in *Bueno et al. Nat Genet. 2016*, and $n=15$ new WGS in the recent work by *Creaney et al. Genome Med. 2022*);

(ii) that our study also represents the largest MPM cohort profiled at three omic layers (genome, transcriptome, and epigenome), which facilitated the use of a unified framework (*i.e.* Multi-Omics Factor Analysis) research approach, this framework is necessary given the previous individual observations made by us and others (*Alcala et al. EBioMedicine 2019*, *Blum et al. Nat Commun. 2019*, and *Creaney et al. Genome Med. 2022*) that capturing and summarizing the entire molecular landscape would be required to really progress the field of MPM research; and lastly,

(iii) that we applied insightful theories and methods from other fields, such as the Pareto task allocation theory from cancer ecology and evolution, in order to not only get the most out of our dataset but to be able to uncover aspects of MPM biology not otherwise identifiable.

RESPONSE TO THE REFEREES

Reviewer #1: (mesothelioma genomics)

Remarks to the Author: This is an impressive WGS study of MPM with very sophisticated bioinformatic analyses. While the dataset represents an invaluable resource, there are few actionable findings and the few interesting observations are not followed with functional data.

Answer: We thank the reviewer for noting the value of our work and the quality of the paper, and express our thanks for the many valuable suggestions on how to improve the manuscript. Below we fully address the specific comments provided, but would like also to address the important general comments on actionability and functional data. Given the

design of our study, and the nature of the material available, we didn't anticipate our study would have clinical impact in the very short term. But rather, through our unsupervised analyses we aimed to examine the inter-patient molecular differences that may uncover the possible reasons for the limited impact novel therapeutics have had on mesothelioma survival. Therefore in order to better fulfill this aim, where applicable, we have now included discussions on the actionability of our findings, which are really focussed on the way they may be used for patient stratification in clinical trials. With regard to functional data, due to material and resources available we were not able to conduct such wet-laboratory validation ourselves, but that does not preclude us from investigating our findings in a biological model. That is why we have now moved some of our original analyses conducted in mesothelioma cell lines from supplementary material into the main text, but also included new analyses on this valuable resource, as suggested by the reviewer. We hope that these inclusions are of interest to the reviewer and improve the relevance of the manuscript to the mesothelioma community.

1. Many of the q values are not impressive, i.e. barely below 0.05

Answer: We fully agree with the reviewer about the importance of reproducibility of results, and thus now provide more information along with the manuscript corroborating our results and interpretations, in particular highlighting in a new **Table S6** (see simplified version below and **Table S6** extended version for review only) all replications that we performed, including for the less impressive q -values. Regarding the q -values themselves, it might be that we have not chosen the best way of disclosing them. Indeed, all but 2 p -values reported in the main text and Figures (now 19/21, previously 23/24), and most q -values (now 44/59, previously 21/37) are actually below a more stringent cutoff of 0.01. The remaining (now 15 out of 59, previously 16 out of 37) q -values reported are between 1% and 5%.

Many of the remaining q -values (6 out of 16 previously) between 1% and 5% were actually the *largest* among a set of q -values: this is the case for all q -values referring to **Figure 2c** and **Table S8** (for a total of more than 200 q -values, the maximal is >0.01 , but the median is well below 0.01 in 5/6 instances), and **Table S4** (for a total of 14 q -values, the maximal is 0.028, but the median is below 10^{-3}). Therefore, we have decided to refer directly to the supplementary table gathering all the q -values supporting our conclusions or to report in the main text the median q -value instead of the maximal q -value to give a better sense of their scale to the reader. In addition, we now avoid graphical representation of q -values

ranges and therefore we have replaced the panels illustrating the IGSEA results by more conventional graphics, heatmaps of the enriched pathways (**Figure 2c**).

Five out of the remaining q -values between 1% and 5% were computed from clinical variables (the asbestos exposure score and overall survival), and tended to be larger (0.01 and 0.02), which is expected given the larger rate of missing data in these precious clinical variables ($n=47$ for the asbestos analysis instead of $n=120$ for genomic analyses). We now report a successful replication of the associations with asbestos in the Bueno cohort with a cruder exposition (yes/no instead of our score taking into account duration, frequency and strength of exposure), and a replication of the association between the biological interpretation of each latent factor and survival in **Table S6**.

The last q -values between 1% and 5% correspond to analyses with lower power. Firstly, the overexpression of the E2F pathway in WGD samples, which might also suffer from limited power due to the low number of samples in the WGD+ group ($n=17$). We now mention (p. 4) that this analysis is itself a replication of the results from the PCAWG consortium (Quinton et al. *Nature* 2021), which found this pathway consistently upregulated across most common cancers, although we could not replicate this result in the TCGA MESO cohort, possibly due to a lack of power (11 WGD samples in this cohort). Secondly, q -values related to the relationship between specific alterations and archetypes (**Figure 5c**); these analyses are based on ANOVAs between archetype proportions and the mutational status (altered or wild-type) and thus have a power that depends on the number of altered tumors (ranging from 3 to 71), and thus lower than other molecular analyses such as those based on continuous variables like gene expression. We have now systematically performed all statistical analyses, where possible, independently in the Bueno and TCGA cohorts, and report these, the majority of which were able to be replicated, although the lack of WGS prevented the calling of structural variant and thus strongly limited the possibility to replicate findings involving large genomic events or alterations in drivers mostly affected by structural variants, as is often the case in MPM (see **Table S6**).

Summary of Table S6. List of p - and q -values from MESOMICS cohort and replication

Initial submission	n	Category	n	Action taken in R1	Result	Comment (see above)
		Largest among set of values	6	Report median & replicate	5/6 medians < 0.01; all	

0.01 < p or q < 0.05	17				replicated	
		Low power clinical associations	5	Replicate results	4/5 replicated	No detailed asbestos exposure limited replication
		Low power molecular tests	5	Replicate results	3/5 replicated; 1/5 is replication of other studies	Hypotheses including large genomic events could not be properly tested
		Other	1	Replicate results	1/1 replicated	

Finally, we now stress in the **methods (p. 16)** our philosophy regarding the reporting of p - and q -values and maximizing results reproducibility. We first note that we took a conservative q -value threshold of 0.05, instead of some q -value thresholds of 0.1 or 0.2 commonly observed, which means that among 59 q -values reported we only expect two or three to be false positives. In addition, in line with the American Statistical Association statement on the misuse of p -values (**Wasserstein and Lazar, *Am Stat.* 2016**), which intends to “steer research into a ‘post $p < 0.05$ era’,” we decided to report all p - and q -values, even those that may be closer to arbitrary thresholds such as the 5% threshold.

- 2. The authors describe CIMP in a substantial minority of MPM. While methylation of the mesothelin promoter was noted in the TCGA study, a CIMP phenotype was not called in that dataset. More detail is needed on this observation. In CRC, CIMP overlaps partially with MSI. A comment is needed on MSI and TMB in the CIMP subset of MPM. That would make this observation potentially actionable.**

Answer: In order to clarify our analyses with regard to this phenotype we have provided additional details about the CIMP index, and its replication in the TCGA cohort, below, and in the study **methods (section: CpG island methylator phenotype index, p. 29)**. As suggested we have now examined microsatellite instability (MSI) in our cohort (see **methods p. 19** and **Table S2**), and provide a discussion of its link with CIMP index and clinical impact below and in the **results (p. 9)**.

To validate the finding of variation in CIMP in our cohort, we examined whether this was also the case in the TCGA dataset. We apologize that this validation was not clearly explained in our initial submission and have improved the relevant methods section. Using the identical method employed in the MESOMICS cohort, of calculating the proportion of all CpG islands with mean beta value ≥ 0.3 , we found the CIMP index in TCGA samples to range from 0.31 to 0.44 (**Table S2**, and **methods** section: *CpG island methylator phenotype index*). In the MESOMICS cohort, the CIMP index ranged from 0.32 to 0.47 (**Table S2**), and there was no significant difference in the distribution of CIMP index values between TCGA and MESOMICS (p -value=0.98, student's t-test). As with the CIMP index in the MESOMICS cohort, CIMP index values in the TCGA cohort were also correlated with one MOFA latent factor (TCGA-LF4, see **Figure S2** and **S3**), and were associated with a significantly higher hazard ratio in survival analysis (see **Figure S5G**).

We have been deliberate in not describing this finding in MESOMICS and TCGA as a CIMP+ phenotype as the method we have used to investigate CpG island methylation level, based on DNA methylation array data, differs from the classical gene panel model assessed through methylation-specific PCR (*Weisenberger et al. Nat Genet. 2006; Hughes et al. Cancer Res. 2013*). Instead we refer to our measurement as a CIMP *index*, with a continuous rather than categorical interpretation. Nevertheless, we did compute a CIMP proxy based on the mean methylation level (beta value) of promoter CpG islands for the five genes in the *Weisenberger et al.* panel (see **methods** section: *CpG island methylator phenotype index*), and found this proxy to be significantly correlated with CIMP index for both the MESOMICS (p -value= 1.36×10^{-36} , $r=0.86$) and TCGA (p -value= 3.08×10^{-24} , $r=0.84$) cohorts (**Table S2**). Therefore we are confident in our finding of variation in CIMP within MPM, and that a subset of samples display a high level of CIMP.

With regard to CIMP in colorectal cancer, it is well established that CIMP positive status often co-occurs with MSI (*Phipps et al. Gastroenterology 2015*). We thank the reviewer for their suggestion to investigate this further and have therefore performed additional analysis to call MSI in our cohort using the MSIsensor-pro software (*Jia et al. Genomics, Proteomics & Bioinformatics 2020*, see **methods** section **p. 19**). We detected one sample (MESO_084_T) to display MSI, which also had a high tumor mutational burden compared with other mesothelioma samples (13.3 mutations per mb, while the median TMB in our cohort was 0.98 per mb, **Table S13**). However this sample had a CIMP index value of 0.34, compared with the median of 0.36 in our cohort, therefore we conclude that MSI is very rare in our series of MPM, and co-occurrence with high CIMP index could not be established. This is now mentioned in the **results** (**p. 9**).

There has been limited studies of potential therapies to target CIMP+ tumors, which may be due to its variability in prognostic value (for example, CIMP+ status is associated with better survival in gliomas but poorer survival in colorectal cancer) and its uncertain etiology (Malta *et al. Neuro-Oncology* 2017; Juo *et al. Ann of Oncol.* 2014). In this study we have investigated potential drivers of high CIMP in mesothelioma, which given the association between higher CIMP index and poorer survival, could be considered targets for future therapies. As discussed in the manuscript, we identified *EZH2* overexpression as one potential mechanism which could be exploited, but upon the reviewer's suggestion we have also examined additional mutational drivers found in other tumors known to display a CIMP+ phenotype. CIMP positive status has been associated with mutations in *BRAF* in CRC, and *IDH1* in glioma, however in our cohort only one tumor was positive for *BRAF* mutation (MESO_061_T, CIMP index 0.35) and no *IDH1* mutations were detected, therefore unfortunately an analysis of CIMP index in association with these mutations was not possible. This is now mentioned in the **results (p. 9)**.

3. The authors describe HRD in 23% of MPM. This is an observation that begs for functional validation in terms of PARPi sensitivity.

Answer: We thank the reviewer for this useful comment. While we are unable to create new functional data, we further examined the cell line data from Iorio *et al.* included in the manuscript in order to provide functional validation of our HRD results, and validated the high rate of this pattern within MPM. This dataset includes 1,001 cancer cell lines (21 MPM cell lines) annotated for somatic mutations, copy number alterations, and hypermethylation, and screened with 265 anti-cancer compounds, including five PARPi (Iorio *et al. Cell* 2016). Indeed, the data from MPM cell lines that we present in **Figures S8-S9** actually contains copy number alterations that we have now used to call copy number signatures using SigProfiler in order to detect HRD signatures in these cell lines. We detected an HRD signature in nine out of 21 lines and thus validated the high rate of this pattern in MPM. Then we tested the association between the response to five PARPi drugs (two Olaparib -one approved and one in clinical development-, ABT-888, AG-014699, and BMN-673) and the HRD pattern. The sensitivity of these cell lines to the clinically approved Olaparib, shows a tendency towards higher sensitivity in HRD samples compared to non-HRD samples (see **Figure S17**) probably due to the presence of non-HRD samples with the same level of sensitivity as the HRD samples.

Interestingly, the first clinical trial with published results (Ghafoor *et al. J Thorac Oncol. 2021*) also demonstrated unexpected results in the association between Olaparib sensitivity and alterations in DNA repair genes, whereby, contrary to their original hypothesis, patients with *BAP1* mutations had poorer survival when treated with Olaparib than wild-type patients. This suggests a highly complex mechanism between the response to this drug, and markers for DNA repair pathway activity. This complexity might explain the reason why we only observe a trend between Olaparib and the HRD pattern. Concerning *BAP1* alterations, our observations in these cell lines go in the same direction as those in the Ghafoor trial. According to this paper and against their first hypothesis, *BAP1* alterations are not a good marker of PARPi sensitivity and are associated with decreased PFS and OS. In our data, Olaparib response is positively associated with the prognostic CIMP index factor ($r=0.65$, **Figure S9**), meaning that CIMP-low samples are more sensitive to this PARPi than the CIMP-high (which are enriched for *BAP1* alterations (**Figure 5a**), and associated with poorer survival (**Figure S5**)). Overall, as suggested by this recent clinical trial and by our data, the interplay between PARP inhibitor sensitivity and markers of DNA repair mechanisms appears more complex than expected, and might explain the weak signal we observed between HRD and PARPi sensitivity. This data is now included in the **results (p. 7-8)**.

4. p.8: LATS2, not LAST2

Answer: We have corrected the typo.

5. The authors state that " the higher rate of chromothripsis in the Acinar phenotype could be a result of BAP1 inactivation known to impair DNA repair" but Fig 3A shows a fairly obvious trend to mutual exclusivity between BAP1 alterations and chromothripsis.

Answer: We agree that our statement was not clear. What we meant is that among other possible mechanisms, for those samples carrying *BAP1* mutations this could be a mechanism for the chromothripsis pattern. Additionally we tested the hypothesis of mutual exclusivity between *BAP1* alterations and chromothripsis using Fisher's exact test and found no significant association between the two events (p -value=0.23), therefore we acknowledge

that our statement might have been too speculative and decided to completely remove it from the text.

In addition, regarding the Acinar phenotype, we would like to note that additional analyses on the splicing machinery have been performed during the time of this review, which support the key role of chromatin-remodeling pathways disruption in defining this phenotype. This is now mentioned in the **results (p. 9-10)** and **methods (p. 25)** and in a new **Fig. S26**.

- 6. p.9: the statement that "The specialization of tumors was influenced by early genomic events. WGD, TERT amp and copy neutral LOH may have occurred more than ten years before diagnosis, both concomitantly with and subsequent to asbestos exposure" is based on data that are either not convincing or poorly presented (Fig 5D).**

Answer: We apologize and acknowledge that **Figure 5, panel d** was not very clear. We now provide a novel design for this panel and rephrased its description in the main text, to make these results comparable with similar results obtained for other cancer types, explicitly state our hypothesis and methods, and better convey the scope and interpretation of our results. In particular, we now explicitly state the limitations of this analysis, which due to the challenges of working with highly immune-infiltrated tumors, is based on the few samples for which a subclonal deconvolution was possible (**methods, p. 42**).

Our hypothesis was that the timing of large-scale genomic alterations in mesothelioma does not differ from that observed in other cancer types, and more specifically, that whole-genome duplication (WGD), large-scale losses of heterozygosity, and amplification of the chromosomal region including *TERT*, typically predate diagnosis by a decade. Due to the limited number of samples ($n=6$), we tested this hypothesis using outlier tests, designed to test whether values observed in mesothelioma were outliers in the empirical distribution of timings of such events in other cancers, but that cannot say whether there is a more subtle trend of slightly higher or lower average timing in mesothelioma. We now report the results of the test of this hypothesis in **Figure 5d** and associated text (see a reproduction of the figure below), showing that estimates of the timing of WGD, *TERT* amp, and copy neutral LOH in our mesothelioma samples fall well within the values observed across >2500 tumors by the Pan-cancer Whole-Genome Consortium (**Gerstung *et al. Nature* 2020**) (empirical p -values ranging from 0.16 to 0.79). We conclude in the main text that

“thus WGD, *TERT* amp and copy neutral LOH may indeed have occurred more than ten years before diagnosis, both concomitantly with and subsequent to asbestos exposure, although conclusive evidence of the timing of these alterations will need to be investigated in subsequent studies, with a hypothesis-driven study design.” (p. 10)

New Figure 5d. Comparison between estimated timings of large-scale genomic events across >2500 tumors from 38 types by the pan cancer whole genome consortium (PCAWG, in gray) and estimated timings from our MESOMICS cohort of MPM (red). p-values correspond to empirical p-values (quantile of the distribution of timings from the PCAWG cohort). PCAWG: Pan-Cancer Whole-genome cohort (Gerstung et al. 2020), MESOMICS: our MPM cohort.

- 7. Fig 3A: it would be very unusual for *MTAP* to undergo homozygous deletion without concurrent *CDKN2A* homozygous deletion. The authors should review these cases for undervalued *CDKN2A* deletion.**

Answer: We thank the reviewer for this comment, that made us realize that this figure was misleading. Indeed, the white color in the figure represented either wild-type or uncertain copy number. We have now added a special light blue/dark blue striped pattern in cases when the gene is cut and therefore the copy number changes within the focal genomic region, and a gray coloring to specify samples for which the alteration is in a segment that did not pass our quality thresholds. The five cases with *MTAP* deletions for which we did not report a *CDKN2A* co-deletion that the reviewer is alluding to actually fell into these two categories: three samples had a cut within the *CDKN2A* region and thus the copy number of the gene was partially 1 and partially 0, and two samples were sitting on a low-quality copy number segment call and thus excluded. We now correctly mention in the **results (p. 7)** that *MTAP* deletion happened exclusively with concurrent *CDKN2A* deletion in our cohort as the reviewer mentioned. In addition, we have corrected the copy number matrix that was used in MOFA analysis that also combined wild-type copy number and ambiguous copy number into the same category; this change only concerned 0.5% of segments and thus had a very marginal impact on the results and did not impact any of our conclusions (see **updated Figures**).

- 8. Fig 4A shows 3 *SETDB1* mutated samples but the authors report only one GNH case. Please discuss.**

Answer: We have added more details about this near-haploid sample and its relatedness with the GNH subtype presented in the TCGA study (Hmeljak et al. *Cancer Discov.* 2018), in the **results (p. 4)**, showing that our results do not contradict what was found in the TCGA

data, but suggest the existence of different types of haploidization and interplays with *SETDB1* alterations.

In particular, both our *SETDB1* mutated samples and our near-haploid sample differ from what was described in the TCGA. Indeed, the three *SETDB1* mutations we found are two missense mutations with uncertain significance ($0.25 < \text{REVEL score} < 0.5$) and a splicing mutation, contrary to the five cases reported in the TCGA that were all more damaging (readthrough mutation, indels, and homozygous deletions). Therefore, our results suggest that *SETDB1* mutations overall are not exclusive to the GNH subtype, but does not contradict the observation that the most damaging *SETDB1* mutations could be enriched in or exclusive to GNH samples. Regarding our near-haploid sample, we now mention that this sample (MESO_108) has a ploidy of 1.10, almost no copy-neutral LOH ($< 1\%$), no *SETDB1* nor *TP53* mutations, and did not undergo whole-genome doubling, contrary to the GNH subtype discovered in the TCGA cohort. Therefore, we conclude that this sample actually does not correspond to the GNH subtype in the sense of Hmeljak and colleagues, but corresponds to another possible genomic trajectory, where genomic instability is driven by alternative pathways than *TP53*, not followed by a genome doubling event and *SETDB1* alterations. The absence of GNH samples akin to those of the TCGA in our cohort of 115 genomes is not surprising given the rarity of this subtype (5/154 TCGA and ICGC samples; Fisher test p -value=0.073).

9. Fig 4A reveals that a concerning proportion of the WGS samples had no matched normal.

Answer: One of the biggest limitations when studying rare cancers is obtaining high quality biological material for a relatively large number of patients to run meaningful genomic studies. This is further complicated by the limited numbers of matched-normal tissue/blood that are collected in biobanks due to cost and space constraints and the fact that normal tissue is not required for routine diagnosis. This limitation is further exacerbated in the case of diffuse malignant pleural mesothelioma, because the diffuse nature of the disease plays against obtaining pure normal adjacent tissue, and also does not help when trying to acquire distant normal tissue.

For our study we took advantage of the French MESOBANK, a multi-centric virtual and exhaustive repository of clinical data, biological samples, and standardized operational procedures for mesothelioma (Galateau-Salle *et al. Ann Pathol.* 2014). Despite containing

more than 10,000 specimens with unique depth of etiological and clinical annotations, most of the biological specimens are formalin-fixed paraffin-embedded material, and for the limited cases with frozen tissue available, matched normal was rare and matched blood almost nonexistent, which are recurrent general limitations of most biobanks. An initial assessment of the tumor-normal (T-N) pairs available with frozen material resulted in more than 100 T-N pairs available. Unfortunately, we experienced a high failure rate of almost 50%. Two major sources of failure were identified: sample retrieval and tumoral content.

With regard to sample retrieval, although virtual biobanks are a very much encouraged option for rare cancers, whereby samples are stored in a decentralized manner, one drawback of virtual biobanks is that the availability of tissue within each center is not necessarily updated regularly, leading to a discrepancy between *reportedly* available and *physically* available samples. Following identification of the suitable matched T-N pairs, only 46 tumors with normal tissue were actually physically available. We faced an additional challenge related to the high infiltration nature of these tumors, which significantly reduced the tumor content, thereby again decreasing the number of suitable samples available for sequencing to 46 T-N pairs. Because funding was available and frozen tumor was also available for a significant number of samples, we decided to proceed with a series of tumor only samples because while we agree that this is an additional challenge that requires tailored computational work to ensure the validity of the results in terms of molecular genomic alterations, **the lack of matched normal is not an issue for transcriptomic and methylation analyses, for which the normal reference is not required.**

Given that the lack of germline for a part of our MESOMICS samples (72/115) was an early fact in our project, **we designed and implemented a battery of bioinformatic tools to computationally solve this issue with the aim of performing accurate and confident calling of all kinds of MPM somatic alterations.** Our main idea was to implement a strategy that took advantage of our matched series (46 WGS) throughout the whole process. In particular, the matched series was used for training advanced machine learning methods, building a custom panel of germline variants (Panel of Normals, PoN), and finally, but most importantly, for performing extensive benchmarking that led to the improvement of all the computational tools employed for calling somatic alterations in tumor-only samples. Through extensive benchmarking, we demonstrate that our implemented computational strategy, coupled with a subset of tumor/normal WGS samples, achieved excellent results in all somatic alteration calls (CNV, Point mutations, and SVs). Therefore, our matched series of 46 WGS was the distinct resource that allowed us to computationally learn the specific pattern of somatic alterations found in MPM WGS. Indeed, using our computational strategy we were able to call with confidence somatic structural variants at a high precision (~90%),

recall (~87%), and accuracy (~89%), as well as somatic copy number variants. When combined, this permitted us to call and study for the first time in MPM complex mutational processes such as chromothripsis and whole genome doubling, with an increased power (115 instead of 46 WGS).

In summary, **adding the tumor-only series allowed us to increase the statistical power for expression and methylation analyses, while the computational strategy that we developed for calling somatic alterations helped us to reduce to a minimum the impact of the lack of normal material. Beyond our study, the proposed strategy for calling somatic alterations in the case of missing normal tissue is key for genomics studies of rare cancers, in which each sample counts. This is the reason why we describe our methodology in a separate data note manuscript submitted to GigaScience and available in biorxiv (<https://www.biorxiv.org/content/10.1101/2022.07.06.499003v1>), including very detailed information on the strategy and additional quality control for all the datasets presented. For simplicity, we provide additional methodological details on the strategy when addressing the **Reviewer #2** suggestions.**

Finally, we would like to note that many of our results were derived using integrative multi-omic analyses that are robust to missing matched-normal tissue and that were validated using exome data from published cohorts. Indeed, the three novel sources of variation (**Figure 1**) and three cancer tasks (**Figure 2**) were replicated in the TCGA and Bueno series, and copy number (**Figure 3**) and small variant profiles (**Figure 4**) as well as several of their associations with sources of variation and cancer tasks (**Figure 5**) were replicated in the TCGA series. However, replicating our results regarding large genomic events (**Figures 3 and 4**) and their association with sources of variation and cancer tasks (**Figure 5**) would require an additional series of WGS data and could thus not be replicated with an existing external cohort. Nevertheless, as we show in the data note, **we rigorously evaluated the performance of our method by splitting the dataset into training and testing sets and found that we could predict these structural variants with high accuracy.**

Reviewer #2: (lung cancer genomics)

Remarks to the Author:

This is the largest WGS analysis of MPM to date and the authors used unusual approaches for the analyses. In my opinion, the analyses require major revisions and validation (or at least comparisons of results) using alternative approaches. However, there are some important limitations that cannot be fully addressed even with major revisions (first of all, the lack of germline material as reference for most tumors).

Answer: We would like to thank the reviewer for noting the importance of our data. Regarding their comment on the “*unusual approaches*”, we acknowledge that we favored recent methods for data integration (MOFA, Argelaguet *et al. Mol Syst Biol.* 2018; Argelaguet *et al. Genome Biol.* 2020), and interpretation (Pareto task analysis, Hausser and Alon, *Nat Rev Cancer* 2020), rather than those used, for example, in the classical TCGA papers (consensus clustering of expression data, from Wilkerson and Hayes, *Bioinformatics* 2010, or integrative clustering, from Mo *et al. PNAS* 2013), and previous mesothelioma genomics papers (Bueno *et al. Nat Genet.* 2016, Hmeljak *et al. Cancer Discov.* 2018). MOFA and Pareto have multiple advantages when capturing molecular variation compared to the previous approaches, such as allowing us to integrate arbitrarily many ‘omic datasets (*vs* a single dataset in ConsensusCluster+ and four in icluster+), capturing both continuous and discrete independent sources of biological variation (*vs* discrete clusters only), and quantifying the importance of each ‘omic layer in separating samples and clusters (*vs* unknown relationships between clusters). However we understand the reader may want to see more conventional representation of the related results and compare the findings from these exciting methods with those previously used in mesothelioma publications.

We now provide further justifications for our choice of methods (p. 34-36) and provide in **Figure S32** (see below) a comparison of our results with those obtained using the exact methods used in previous large-scale genomic studies (A-D, ConsensusCluster+ as in Bueno *et al. Nat Genet.* 2016, and E-H, icluster+ as in Hmeljak *et al. Cancer Discov.* 2018). We show that these methods capture some of the variation in the dataset: the three archetypes (B and F, to be compared with **Figure 2a**), higher-ploidy samples (G, middle panel), and the CIMP index (C, G, right panels), but in a much cruder way than with MOFA and the Pareto task analysis, mostly because the discreteness assumption of clustering methods misses the inherent continuity in the data (noted in mesothelioma by Blum *et al. Nat Commun.* 2019, and ourselves, Alcalá *et al. EBioMedicine* 2019). Indeed we observe small and fragile clusters, with negative or low (<0.25) silhouette widths for a large number

of samples, indicating uncertain or wrong cluster assignment (D and H), and preventing downstream analyses. We think that these new analyses both validate our results, and at the same time highlight the need to use more modern techniques that were not yet available in previous mesothelioma studies.

Note that MOFA is increasingly becoming a new standard in multi-omic analyses, with the two methodological papers now having surpassed in only two years the number of citations that the iCluster+ paper accumulated in almost a decade (427 vs 344 according to google scholar), and recent high-profile reviews in *Nature Biotechnology* (Argelaguet *et al. Nat Biotechnol.* 2021) and *Nature Methods* (Efremova and Teichmann, *Nat Methods* 2020), and high-profile cancer applications (e.g., Lu *et al. Nat Cancer* 2021), making it appealing and familiar to the *Nature* journals readership.

New Figure S32. Association between MOFA latent factors and the clusters identified by consensus clustering (A-D) and integrative clustering (E-H). (A) Kruskal-Wallis rank sum test p-values between each clustering (row) and MOFA LFs (column). (B) Consensus clustering results for K=3 clusters (colors) in MOFA latent factor space of LF2 vs. LF3. (C) Same as (B) but for K=4 clusters, with an additional plot of the samples in the one-dimensional space of LF4. (D) Top-left: average silhouette width for consensus clustering with different values of K. Bottom-left: proportion of samples below the selected silhouette width threshold for consensus clustering with different K. Right: consensus matrix heatmap for K=3. (E-F) Follow the same design as (A-D) but using iCluster+ instead.

Additionally, we have taken it upon ourselves to alter the visual representation of the IGSEA results obtained from Pareto task analysis. These results, used to infer cancer tasks assigned to each archetype, are now represented as heatmaps (Figure 2c), a more classical and reader-friendly display.

Regarding the reviewer's comment on the "lack of germline material as reference for most tumors", as previously addressed in **Reviewer #1, comment 9**, we acknowledge that missing germline material for 72 out of 115 tumors, due to the difficulty to obtain adequate samples for such rare tumors, is a challenge that requires much additional computational work to ensure the validity of the results. The lack of germline for a part of our MESOMICS samples was an early fact in our project and as such we designed and implemented a battery of bioinformatic tools to computationally solve this issue with the aim of performing accurate calling of all kinds of MPM somatic mutations. To implement our strategy, we took advantage of our matched series (46 WGS) throughout the whole process. In our strategy, the matched series was used for training advanced machine learning methods, building a custom panel of germline variants (Panel of Normals, PoN), and finally, but most importantly, for performing extensive benchmarking that led to the improvement of all the computational tools employed for calling somatic mutations in tumor-only samples (see **Figure below**). Therefore, our matched series of 46 WGS was the distinct resource that allowed us to computationally learn the specific pattern of somatic mutations found in MPM WGS.

Our strategy generated almost perfect results for calling somatic copy number variants (see **Figure below**). Briefly, we ran the PURPLE copy-number caller twice for each matched sample (**Figure below, panel A**): first using the matched WGS samples as input, and second using only the tumor WGS as input. Subsequently, we performed a direct comparison of the PURPLE tumor-only calls with their corresponding matched-pair calls for several features including ploidy, the proportion of the deleted genome, and copy number states at the gene level, among others. The benchmark revealed a high concordance across all the evaluated metrics between tumor-only and matched PURPLE calls. Indeed, the agreement for ploidy ($R=1$) and percentage of genome deleted ($R=0.999$) exceeded a 0.99 correlation (**Figure below, panel B and C**). Moreover, a high concordance was also observed at the gene level with minor copy number alleles reaching $R>0.977$ (**Figure below, panel D**). The benchmarking demonstrated that calling copy number variants in MESOMICS tumor-only samples was highly accurate and indeed almost matched the calling of tumor-normal pairs, and therefore we confidently applied the tumor-only mode of PURPLE to call somatic CNV in our MESOMICS tumor-only WGS series.

For calling single nucleotide variants and indels, we trained and evaluated the performance of a supervised machine learning model based on random forest for distinguishing germline from somatic variants in tumor-only WGS (see **Figure below**). The matched WGS were used as input for training and evaluating the performance of the random forest (RF) model (**Figure below, panel A**). Point mutations were called using Mutect2 as

implemented in our NextFlow pipeline (<https://github.com/IARCbioinfo/mutect-nf>, release v2.2b). We designed a RF model with a total of 20 features divided into three main classes, namely: associated with external databases (gnomAD, COSMIC), genomic location/impact, and features obtained directly from the Mutect2 variant caller. For training the RF model, the 46 tumors with matched normal available, called with both the tumor-only and matched modes of Mutect2 were used. The matched somatic calls (ground-truth) were used to annotate the variants of the tumor-only WGS into germline and somatic classes. A total of 407,984 SNVs in a 1:1 proportion of germline:somatic were used for training (75%) and evaluating (25%) the performance of the RF model. The same strategy was employed for training and evaluating the RF model for classifying indels. The performance of the optimal RF-models for SNVs and indels reached a precision of 0.92 and 0.92, recall of 0.94 and 0.93, and accuracy of 0.93 and 0.94, respectively (**Figure below, panel B**). Finally, the trained RF models were used to classify a total of 1,454,942 point mutations of which 217,436 were classified as somatic. The benchmarking result demonstrates that we developed a highly accurate and robust methodology to call point mutations in tumor-only WGS datasets for which a series of matched tumor-normal samples are available.

Similarly to point mutations, we implemented custom RF models to distinguish at high accuracy somatic from germline structural variants (SVs) in tumor-only samples. The RF-models were composed of a total of 19 features based on external databases (PCAWG, gnomAD), a custom panel of normal (46 matched WGS), genomic regions, and SV features obtained directly from state-of-the-art SV callers (Delly, Manta, and SVaba). The training (75%) and evaluation (25%) of the RF model for each SV caller were performed using a total of 12,454, 16,720, and 12,264 SVs at 1:1 somatic:germline proportions for Delly, Manta, and SvABA, respectively. The performance in terms of precision, recall, and accuracy achieved by each RF model were on average 0.90 ± 0.009 , 0.87 ± 0.016 , and 0.89 ± 0.010 , respectively (**Figure below, panel C**).

Calling and benchmarking of copy number variants of tumor-only samples

Calling and benchmarking of point Mutations and structural variants in tumor-only samples

Overview and benchmarking results of the computational strategy implemented for calling somatic alterations in tumor-only MPM samples. Panels are reproduced from Data Note manuscript, figures 4A-C and 5A, C, and E.

In summary, we have demonstrated that we were able to implement a computational strategy that, coupled with a subset of tumor/normal samples, achieved excellent benchmark results in all kinds of somatic mutations (CNV, point mutations, and SVs). Furthermore, we are providing with this revision an additional manuscript submitted to the *GigaScience* journal and currently available in biorxiv (<https://www.biorxiv.org/content/10.1101/2022.07.06.499003v1>) describing the full details of the computational strategy and implementation developed to call with confidence genomic alterations in T-only MPM samples, as well as additional quality control of the novel multi-omic data generated for this manuscript.

Reviewer #3: (lung cancer genomics)

Remarks to the Author: The authors report findings from whole genome sequencing and multi-omic integrative analyses of tumor tissues from patients with malignant pleural mesothelioma. The authors state that ploidy, adaptive immune response, CpG island methylation along with histological subtypes explain the variation in the phenotypic behavior in this disease with nearly uniform poor outcomes. State of art tools from sequencing to analyses are used from the French Mesobank source (predominantly). Potentially, this approach could improve the current solely histology-centric classification. The paper would be of great interest to the community of mesothelioma researchers and clinicians given that this body of work represents the largest data from whole-genome sequencing. I commend the authors for this remarkable work.

Answer: We thank the reviewer for the positive assessment of our manuscript and the encouraging note on how our work will be of great interest for the mesothelioma community.

- 1. It is not clear what cohort was used for various analyses. There are multiple cohorts- MESOMICS, Discovery (with and without ITH), and other published cohorts, TCGA, Bueno, in addition to cell lines. Most of the confusion centers around MESOMICS and Discovery cohorts. What is the Discovery cohort and what constituted this cohort (demographics, histology, etc)**

Answer: We thank the reviewer for this comment and we apologize for the lack of clarity when describing these cohorts in the initial submission. We now clearly distinguish each of these cohorts by indicating the cohort and sample size in the text and legends for all analyses. In brief, the MESOMICS cohort has two parts, the 'discovery cohort' and the 'ITH cohort', the MESOMICS discovery cohort consists of one tumor sample for each of the 120 patients, and is always referred to as MESOMICS (**Table S1**). The MESOMICS ITH cohort is made up of patients from the MESOMICS discovery cohort for which we have more than one tumor

sample available, and consists of 13 samples from the discovery cohort (12 of them from the discovery cohort) and an additional 13 samples from the same patients to study intra-tumoral heterogeneity (**Table S15**). We have clarified this in the **methods (p. 15)**, and include a dedicated cohort description in the main text (see response to your **Comment 2** below). TCGA and Bueno refer to two independent replication cohorts from previously published studies: **Hmeljak et al. Cancer Discov. 2018**, called “TCGA” and **Bueno et al. Nat Genet. 2016**, called “Bueno”. The cell line cohort is used for replication and clinical translation, and was obtained from **Iorio et al. Cell 2016**, **de Reynies et al. Clin Cancer Res. 2014**, and **Blum et al. Nat. Commun. 2019**. The MSK-IMPACT cohort is used for replication and was obtained from **Bielski et al. Nat Genet. 2018**.

2. **There should be a clear description of the cohorts (in the written part of the manuscript) outlining the treatment received (surgery, systemic therapy) and follow up for the Meso cohort and as well as the Discovery Cohort**

Answer: We now provide an additional tab in **Table S1** with a detailed description of the three primary MPM cohorts (MESOMICS, TCGA, and Bueno) including when available sex, race, age, asbestos exposure history and source, histological type, surgery, therapy, stage, survival time, and OS per histological type. This description is expanded in the **methods (p. 15)** as described in the answer above (**Comment 1**), where we also added its main differences with TCGA and Bueno cohorts.

3. **The first section titled integrative multi-omics analyses uncover novel axes of molecular variation is hard to follow. It is hard to understand how much the latent factors (LF) 1,3, 4 add to the histological factors? Is one histology more heterogeneous in terms of LF than the other histology? Despite its shortcomings, histological classification is the current and only standard in the clinic for treatment decision-making (surgery and the choice of initial systemic therapy, for example). I have a hard time understanding what these three (excluding the morphology factor) contribute to the "molecular variation"**

Answer: We apologize because obviously we have not been able to properly explain what these novel latent factors are. We now first show that the current histological classification

only explains 9% of the inter-patient variation in terms of gene expression, DNA methylation, copy number variation, structural variants, SNVs and indels, leaving more than 90% of inter-patient molecular differences unexplained (**Figure 1a**). We then better state in the **results (p. 3-4)** and in the **discussion (p. 10)** that LF1-4 are all independent sources of inter-patient heterogeneity that together explain more than 60% of the molecular differences between patients, and that LF1, 3, and 4 are previously unexplored/unknown because of the major focus that previous studies have put on refining the histological groups.

The fact that they are all independent and mostly unrelated to the Morphology factor (histology, LF2) is precisely why they are interesting. Indeed, the identification of factors unrelated to morphology means that if we focus only on LF2 (morphology) we are disregarding very important additional and independent sources of molecular variation with prognostic value. To illustrate this, we have identified tightly correlated proxies for these LF (see below, answer to **Comment 4**), which allowed us to explicitly state in the results and show in a novel **Figure S5 (panels A-C, see below)** that all histological types can be either WGD+ or WGD-, have high or low adaptive immune responses, and have a high or low CIMP index, and that there is no significant association between these features and any histological type (**results, p. 4**). In addition, we emphasized the complementarity of these sources of variation in predicting patient prognosis in **Fig. S5**, where we compare multiple predictive models using prognostic factors in MESOMICS and TCGA cohorts. These analyses show that, together, these four axes better predict survival than the previous factors strongly associated with the histological classification (**results, p. 5**). Thus these three novel axes (LF1, LF3, and LF4), further explored in this study, provide three additional opportunities to stratify patients based on their molecular profile, and this could be the missing piece to the puzzle in understanding why many novel therapeutic approaches applied to unselected populations of malignant pleural mesotheliomas are failing to produce clinical benefit.

New Fig. S5. Association between histological types and proxies for the MOFA latent factors. (A) Association between Whole-genome doubling (WGD) status (proxy for the ploidy factor) and histological types in the MESOMICS and MSK-IMPACT cohorts. (B) Association between the Adaptive versus innate response score (proxy for the adaptive-response factor) and histological types, in the MESOMICS and Bueno and TCGA cohorts. (C) Association between the CIMP-index proxy computed on a five-gene panel (proxy for the CIMP-index factor) and histological types in the MESOMICS and TCGA cohorts.

If we take as example another recalcitrant cancer such as small-cell lung cancer (SCLC), clinical trials for SCLC focusing on unselected populations have yielded disappointing results and this is likely because SCLC has been regarded as a homogeneous disease. But the recent classification of SCLC based on their transcriptomic profiles by Rudin and colleagues (Rudin *et al. Nat Rev Cancer* 2019) has unveiled therapeutic vulnerabilities to treatments previously considered ineffective in SCLC (Gay *et al. Cancer Cell* 2021). For example, considering the high tumor mutation burden, immunotherapies have been recently added as front-line therapy in the treatment of SCLC, but they only made modest improvements in overall survival (Plaja *et al. Cancers* 2020; Esposito *et al. Cancers* 2020). In a retrospective analysis of the IMpower133 trial, a trend was observed that the greatest benefit was provided to one of these expression groups that they named SCLC-inflamed. This corresponds to a minority of SCLC patients who have a relatively activated and immune “hot” microenvironment (Gay *et al. Cancer Cell* 2021; Dora *et al. Mol Oncol.* 2020; Owonikoko *et al. J Thorac Oncol.* 2021).

The same kind of outcome is what we anticipate from our study on MPM. By unveiling the so far unknown unknowns in this disease we hope to provide the clinical community with the tools to design better suited clinical trials and better interpret the results of previous or ongoing trials for a disease for which the prognosis and clinical management have not significantly changed over the past decades. We feel that continuing to refine the histological classification will not provide any meaningful improvement to the current unmet needs, and therefore, new ground-breaking approaches are needed. Indeed, while the nature of our data makes our findings difficult to be implemented in the clinic, the tightly correlated proxies that we have identified, could serve as biomarkers for response to specific

therapies (such as immunotherapy for LF3) and could be easily tested in a hypothesis-driven study design.

In summary, we hope this work would reset the course of clinical testing, by identifying potential additions to diagnostic work up and removing the sole emphasis on histology, which, as the reviewer correctly said, is the current and only standard feature used in the clinic for treatment decision-making despite, as shown in our study, it only representing one out of the four sources of molecular variation in MPM. We have tried to convey all of this in the **discussion** of the manuscript (p. 11-12), and also illustrate how our novel classification explains inter-patient differences in three patients with similar clinical characteristics but vastly different molecular profiles (novel **Fig. S31** reproduced below).

New Fig. S31. Added value of the four-factor molecular classification in understanding inter-tumor heterogeneity in three example patients. The three patients have similar clinical characteristics (top table) but different overall survival, and different molecular profiles captured by our four-dimensional molecular classification.

4. **Results shown in Figure 1e regarding the prognostic value of LF1-4 would have been stronger if there is validation in an independent cohort. Those findings did not hold up in the larger subtype of the epithelioid type where only LF3 appeared to be significant (Fig S5B). This should be highlighted. Unfortunately, it is hard to assess the sarcomatoid histological subtype without adequate numbers. Once again, one cannot draw definite conclusions in the absence of a validation cohort**

Answer: We share the same opinion as the reviewer on the importance of reproducibility, and now better highlight which results have been independently validated in **Table S6** (see answer for **Reviewer #1, Comment 1**), and in particular cover the validation of the prognostic value of the latent factors in a new **Figure S5 (see below)**.

We acknowledge the influence of the prognostic histological type on the prognostic value of these latent factors. This is the reason why we provided in **Figure S6** (former **Fig. S5**) and **Table S5** the results for survival analyses in MME only. We found the Adaptive-response factor (LF3) to be significantly associated with survival in both the mixed and MME-only cohorts from MESOMICS, while Morphology factor (LF2) unsurprisingly lost its prognostic value in this subset of samples, due to its association with the histopathological types. Whilst position along the Ploidy (LF1) and CIMP-index (LF4) factors was not associated with tumor type, when excluding MMS samples their prognostic value was not significant following correction for multiple testing, however their respective effect size remains similar to those identified in the entire cohort including non-epithelioid samples. Specifically, for LF1 HR=1.18 in the all MESOMICS cohort $n=120$ vs 1.22 in MESOMICS MME-only subset $n=79$. For LF4, HR=1.29 in the all MESOMICS cohort $n=120$ vs 1.14 in MESOMICS MME-only subset $n=79$. It is possible that the loss of statistical significance is due to a decrease in power. We now point this out in the **results (p. 5)**.

As the reviewer highlighted in **Comment #3**, the interpretation of the latent factors and their use in the clinic is not straightforward, which is why we chose to represent and validate simpler proxies for each factor instead. For each proxy we display the prognostic value in the MESOMICS cohort, and in a validation cohort obtained from a previously published study, which data allowed for such validations (see **Methods, p. 17**). For the ploidy factor, we used the Whole-Genome Doubling (WGD) status from the mesothelioma samples of the MSK-IMPACT cohort (**Zehir et al. Nat Med. 2017; Bielski et al. Nat Genet. 2018**). As initially reported by the authors across more than 30 tumor types, we find that

WGD is associated with poorer survival in the MSK-IMPACT MPM cohort (**Fig. S5D**). For the Morphology factor, we used the percentage of sarcomatoid component as reported in the Bueno cohort (**Bueno *et al. Nat Genet. 2016***) by pathologists from microscopic examination of H&E slides, which is as expected associated with a worse prognosis (**Fig. S5E**). For the Adaptive-response factor, we used an Adaptive vs Innate immune response score, defined as the difference between the proportions of adaptive cells (B and T cells) minus the proportion of innate response cells (macrophages, monocytes, neutrophils), that we could obtain from the estimated immune cell proportions computed from the gene expression data in all RNA-seq cohorts (MESOMICS, Bueno, and TCGA) This is based on the fact that B and T cells are associated with good prognosis while macrophages are associated with bad prognosis, and this has been recently proven also true in mesothelioma (**Ollila *et al. Front Oncol. 2022***). We found that this score was associated with a better prognosis (**Fig. S5F**). Finally, for the CIMP-index factor, we used a simpler CIMP-index proxy computed from only five genes, and replicated our results in the only cohort containing methylation data, the TCGA cohort (**Fig. S5G**).

New Fig. S5. Forest plots of hazard ratios for overall survival showing the replication of latent factors' prognostic value. (D) WGD status (proxy for the ploidy factor) in the MESOMICS and MSK-IMPACT cohorts. (E) Percentage of epithelioid estimated by pathologists from H&E slides (proxy for the morphology factor) in the MESOMICS and Bueno cohorts. (F) Adaptive versus innate response score (proxy for the adaptive-response factor), in the MESOMICS and Bueno and TCGA cohorts. (G) CIMP-index proxy computed on a five-gene panel (proxy for the CIMP-index factor), in the MESOMICS and TCGA cohorts.

Even with validations on these additional cohorts, MPM numbers are still very low (e.g., the MSK-IMPACT is the only other cohort available with a satisfactory number of primary tumor specimens assessed for WGD, yet still contains only 12 WGD+ samples). We acknowledge that this limits the interpretation of the prognostic value of the four factors we report in the same way as it has limited the interpretation of the prognostic value of markers proposed in previous molecular studies, but this demonstrates the great value of the MESOMICS cohort, as it will provide to researchers a needed multi-omic resource to test their own hypotheses, in particular providing the largest set of sarcomatoid tumors to date ($n=15$), more than doubling the number of such samples available from the Bueno and TCGA cohorts.

Other minor additional changes

Note that in addition to the changes mentioned above, we have made a series of small improvements and corrections:

- we have improved the computation of the variance explained by each latent factor, using correlation coefficients as in a classical PCA instead of using MOFA's intricate model residuals with a more complex interpretation (see **Fig. 1a**, and **methods p. 36**)
- we have corrected the representation of CN in sex chromosomes in **Fig. 3b**, that wrongly represented males as having lost one copy
- we have improved the assessment of damaging structural variants (previously selected through too stringent filters based on the position of breakpoints and the location of the genes), leading to the detection of additional driver SVs (see **Fig. 4**, **Fig. S19**)
- we have improved our method for finding the correspondence between latent factors in multiple embeddings, applying a theorem specially derived for this purpose (see **Fig. S2** and **methods p. 36-37**)
- we now represent the median % of amplifications and deletions across all samples of each PCAWG cohort in **Fig. S18**, instead of the median across samples with CNVs only
- we have improved the differential expression analysis model comparing WGD+ and - samples, using a log-transformation of expression values as suggested in Quinton et al. *Nature* 2021 (see **Fig. S24**)

Decision Letter, first revision:

21st Sep 2022

Dear Dr Fernandez-Cuesta,

Your Article, "Whole-genome sequencing and multi-omic integrative analyses reveal novel axes of molecular variation and specialized tumor profiles in Malignant Pleural Mesothelioma" has now been seen by 3 referees. You will see from their comments below that while they find your work of interest, some important points are raised, particularly by Reviewer #2 who considers your response to their feedback to be incomplete.

We therefore invite you to revise your manuscript taking into account all reviewer and editor comments, and we strongly urge you to fully address all of Reviewer #2's points in full, otherwise we will be reluctant to send the paper back out. Please highlight all changes in the manuscript text file. At this stage we will need you to upload a copy of the manuscript in MS Word .docx or similar editable format.

*2) If you have not done so already please begin to revise your manuscript so that it conforms to our Article format instructions, available [here](http://www.nature.com/ng/authors/article_types/index.html). Refer also to any guidelines provided in this letter.

[redacted]

We hope to receive your revised manuscript within four to eight weeks. If you cannot send it within this time, please let us know.

Sincerely,

Safia Danovi
Editor
Nature Genetics

Reviewers' Comments:

Reviewer #1:

Remarks to the Author:

The authors have provided very thorough responses and have made changes that have improved the paper. I have one general comment and one specific comment.

1. the paper seems to be missing a schematic figure summarizing the novel classification findings. Most of the figures relating to MOFA and latent factors are not visually intuitive and the novelty is hard to grasp. This is also reflected in the abstract which just describes the study in general terms without a clear declarative statement of the novel findings.
2. in Fig 4, the use of the abbreviation ERG for epigenetic regulatory genes (ERGs) in the same list as altered genes will lead to confusion as this is also a gene name. I suggest that the ERG track be moved to the lower section with the type/sex/smoking/asbestos/WGD tracks.

Reviewer #2:

Remarks to the Author:

I reviewed the authors' response to two of my comments and I am satisfied by their responses. I have also reviewed the manuscript in Biorxiv related to tumor-only analyses and we tested the related codes. It seems to work well particularly for SNVs (less for SVs).

However, there is no response to many of the additional comments I originally made. I am not sure why they have been ignored.

See my comments in quotation "" for each original question.

This is the largest WGS analysis of MPM to date and the authors used unusual approaches for the analyses. In my opinion, the analyses require major revisions and validation (or at least comparisons of results) using alternative approaches. However, there are some important limitations that cannot be fully addressed even with major revisions (first of all, the lack of germline material as reference for most tumors). This study aims to identify factors that can contribute to a new classification of malignant pleural mesotheliomas (MPM) with clinical implications. It has the largest MPM whole genome sequencing data to date, as well as RNA-seq and methylation data. The authors performed many analyses using approaches that are not typically used in cancer genomic analyses and with the potential to identify novel findings. They are laudable for their effort to tease apart MPM genomic, epigenomic and histological features to suggest a possible new classification. However, I have several major concerns related to the analyses and data interpretation. I summarize the comments based on the analysis types:

1. Among the 120 tumor samples, >60% of WGS and 90% of RNA seq and methylation analyses lack matched normal or blood samples for the analyses. Although the authors attempted to address this issue with random forest-based and other approaches and even using additional normal samples from other datasets for some analyses, the lack of matched germline reference can affect the results. For example, TERT amplification in Fig.3a appears to be identified almost exclusively in tumor-only samples; and many analyses are presented based on the TERT amp in the manuscript. Another example is the deletion of RBFOX1. It would be important to systematically verify all major findings in the T/N matched samples (at least those based on WGS, since only a handful of normal tissue samples are available for RNA seq and methylation) and discuss discrepancies between matched and tumor-only results. "As mentioned above, a response was provided, although the authors did not specifically commented on the apparent enrichment of genomic driver events in tumor-only samples vs. T/N."

2. In Fig.1a, methylation data in LF1, LF2, and LF3 explains the largest component of the variance. This could be due to the unbalanced size of the MPM datasets used for the analysis. In Fig. S1, it appears that the methylation data is three times larger than the RNA and DNA Alt data. This could also explain why the DNA Alt (driver genes and other genomic features) appears not to explain any variance component. How much variance is explained by each data type? How much variance could be explained by including a larger number of LFs? Are these 4 LFs statistically independent from each other? How sensitive is MOFA to sample size (although this is the largest WGS study of MPM to date, is still small). And the Bueno cohort used for replication lacks methylation data, so it is not ideal to replicate an analysis mostly driven by methylation data. "No answer was provided. The authors need to address the issue of unbalanced MPM datasets"

3. In Fig.S5, LF1 was associated with RNA-seq batch effects and LF4 (CIMP factor) was associated with sample wells (at least this is my interpretation since there is no legend). Did the authors consider these factors in their analyses? How? "No answer was provided to this comment, although the CIMP factor was discussed in response to Reviewer 1. Also, in the revised manuscript (now Figure S6d), the association is not there anymore."

4. In general, it would be important to validate or at least compare this LF classification using other approaches, e.g., iCluster or Cluster-of-clusters, typically used in genomic analyses of other cancer types similarly composed of mixed histologies and molecular features. "The authors responded to this question"

5. The strongest cancer prognostic factor is tumor stage, but the survival analyses do not appear to have been adjusted by stage. This could strongly bias the results. All survival analyses should include stage in the model. "No answer was provided"

6. In the survival analyses in Fig.1e the strongest effect is provided by LF2, which corresponds to the known morphological classification. The other latent factors do not appear to strongly affect survival beyond this factor. In Fig.S5, in the analyses restricted to the epithelioid group, only LF3 is (marginally) associated with survival. In Figure S6, 22 Cox proportional hazards survival models were tested: how did the authors take multiple testing into consideration? In the cross-validation results, the combination of all 4 factors does not substantially improve the AUC (0.79) over the combination on any of three factors (0.78-0.80) or two factors (0.73-0.78). And this seems to be true (although with different estimates) also for the TCGA data. Also, based on the Introduction, aneuploidy and immune infiltration had been already identified as playing a role in MPM, and yet the morphological classification remained as the major classification variable. Thus, it is not clear whether the addition of the data related to the LF1, 3, and 4 can be useful in clinical settings or could be considered for treatment strategies. "No answer was provided, although the authors tangentially addressed this issue in response to Reviewer 3."

7. There is a large difference in terms of purity estimated from RNA-seq, WGS, and by pathology (no purity estimates are reported based on methylation data) in Supp Table 1. How did the authors reconcile the purity differences and which purity estimates were used for the analyses? How much does tumor purity affect MOFA results? Is any LF correlated with tumor purity? In addition, purity was included as an additive covariate in the regression analyses, but others have shown that a multiplicative model would be more appropriate (see Zheng et al. 2017 Genome Biology for example). "No answer was provided."

8. In the Pareto TI analysis, outlier tumors based on gene expression are known to interfere with the position of the archetypes. Did the author verify whether there were any outlier tumors? "No answer was provided."

In Fig2a, the authors described that only LF2 and LF3 create the "triangular shape delimited by 3 extremes", not ploidy or CIMP factors. Can the results based on ploidy and CIMP be presented so one can verify the results and compare them with LF2 and LF3? "No answer was provided."

9. There is a large discrepancy between CIMP-index (32-56%) and CIMP-normal index (1.3-19%). How do the authors explain this large difference? The range of CIMP in tumors is based on beta values >0.3. This is not the typical way the CIMP phenotype is identified. CIMP is important because exhibits highly recurrent hyper-methylation of TSS genes. Can the authors replicate the CIMP analyses using unsupervised clustering methods and evaluate how the LFs vary by CIMP subtypes? "No answer was

provided, although the authors discussed the CIMP factors in response to Reviewer 1.”

10. The TCGA-MOFA methylation analyses are based on the arrays with ~450K probes and appears to replicate the MESOMICS analyses based on the EPIC array with ~850K probes. Can more features in enhancer probes (enriched in the 850k EPIC array) improve the MOFA analysis? “No answer was provided.”

11. Notes: lines 1077-1078 report 2 different estimates (1.3% or 0.0013?); lines 1125-1127: the number of probes used do not match: $150 + 207 + 2446 = 2803$ not 3764 probes. “No answer was provided.”

12. I am not clear how the tumor evolution was estimated across all samples. Do the different LFs identify tumors with different evolutionary trajectories? Fig5d only describes 6 samples. Was the timing of the lesions only conducted in these 6 samples? Four out of 6 are linked to TERT amp (likely based on tumor-only samples). And was asbestos exposure only ascertained in one sample (grey area)? It would be important to time the asbestos exposure in relation to different genomic changes to try to understand whether the asbestos exposure was involved in the initiation or progression of the tumors or had no detectable effect on tumor evolution. “No answer was provided, although the authors acknowledged the limitations of their approach in response to Reviewer 1.”

13. For the gene expression analyses, did the authors take into account copy number alterations (beyond WGD), which could substantially affect gene expression levels? “No answer was provided.”

14. How did the authors identify 3 specific cancer tasks from IGSEA analyses? Can results from other pathways be shown (Table S7 only reports the 3 archetypes)? “No answer was provided.”

15. IGSEA results are confusing: for many gene sets, there only 10-15% of genes overlap with a given pathway, but they reach very low p-values. For example, for downregulated pathways - myogenesis has 11/200 overlapping genes but has much lower p-value ($3.90E-07$) than the down-regulated pathways in the xenobiotic metabolism with 33/200 genes overlap ($p=0.02$). Can the authors clarify these findings? “No answer was provided.”

Table S3 is confusing (and there are 2 “Table S3”). It’s unclear what is the mean expression between the two groups. For example, how were genes with different expression estimate signs labelled (up/down regulated)? In addition, gene names annotation should be included. Similar issues are in Table S7.

Samples with DNMT3B and EZH2 mutations - are they overly expressed in the tumors? “No answer was provided”

16. The authors calculated a genomic instability score using CNV, gene expression, and methylation data, but I am not clear on how these scores were combined or which score was used for which analyses. It seems that there is no great correlation between the scores calculated from different omics in Supp Table 2. “No answer was provided.”

17. In Fig.3A, MPM driver gene overview, the top genes BAP1 and NF2 are both identified as having large deletion events (mostly heterozygous deletions). What is the relationship between these heterozygous deletions and SV (and SNV) events? The gene expression plot for figure 3a should separate different alteration types for these two genes. “No answer was provided.”

18. 23% of the samples have homologous recombination-deficiency (HRD) phenotype. Given the potential clinical implications of this finding, can the authors verify these results with other approaches (e.g., CHORD, HRDetect)? Are there any germline variants in HRD genes in these samples? “No

answer was provided, although the authors reported previous cell lines results and clinical trial data to support their findings on HRD in response to Reviewer 1."

In Figure 3, only NF2 is reported on chr22q; however, there are multiple cancer driver genes (TSS) located on this region (such as CHEK2 and EP300). Unless there is a need to emphasize NF2 for its link to MPM, these two cancer driver genes should be considered. "No answer was provided."

19. In Fig.4, can the specific structural variants be defined (large indels, translocations and fusion transcripts)? For example, do large indels overlap with heterozygous deletion? What is the difference between translocation and fusion transcripts? How was the "CNB" estimated? By number of segmentations? Copy number breakpoints? "No answer was provided"

Are any specific alteration types in BAP1 and NF2 in the matched T/N WGS data? "No answer was provided."

20. Mutational signatures: the authors mentioned de novo signature extraction for SNVs, but no results are reported. "No answer was provided"

It is interesting that APOBEC mutations have much lower TMB than other signatures, including the age signature (Fig S10). This is different than many cancer types and could be discussed in the Discussion. "No answer was provided". Moreover, APOBEC 2 and 13 are usually associated. In this figure, SBS13 is only presented in half of samples with SBS2. Also, 3 samples have a platinum therapy signature: weren't the samples treatment naïve? If not, this should be mentioned at the beginning, since treatment may strongly affect genomic changes. If yes, why these signatures? The authors may want to check the signature assignment. "No answer was provided."

The authors proposed age as the etiological factor responsible for signatures 1, 5, and 40. However, at least for signature 40 the etiology is still unknown.

The authors showed 7 copy number signatures but it's unclear how the number of copy number signatures were derived/chosen and how they were assigned to be associated with certain processes. "No answer was provided."

21. ecDNA: on Figure 3C, the ecDNA region on chr10 appears to have extremely high WGS coverage compared to the rest of the region, but the estimated copy numbers are not higher than other regions. I suspect copy numbers were wrongly estimated here. Otherwise, how can the authors explain this? "No answer was provided."

In Figure S11, there appears to be a recurrent ecDNA identified on chr13:15-17Mbp. However, this region largely overlaps with the chr13 centromere, which should be excluded from the analyses. "No answer was provided."

In Figure S13, in tumor 019T there is chromothripsis in the same regions where ecDNA was identified. The authors could emphasize this finding, since chromothripsis can be a primary mechanism of genomic DNA rearrangements and amplification into ecDNA. "No answer was provided."

22. In Figure S8, the color legend is missing. "The color legend for the Pearson correlation has been added to what is now Figure S9."

Reviewer #3:

Remarks to the Author:

I appreciate the thoughtful response to the previous critiques.

Author Rebuttal, first revision:

Reviewer #1:

Remarks to the Author:

The authors have provided very thorough responses and have made changes that have improved the paper. I have one general comment and one specific comment.

1. the paper seems to be missing a schematic figure summarizing the novel classification findings. Most of the figures relating to MOFA and latent factors are not visually intuitive and the novelty is hard to grasp. This is also reflected in the abstract which just describes the study in general terms without a clear declarative statement of the novel findings.

Answer: We thank the Reviewer for the positive assessment of our paper. We now better highlight in the abstract that we propose a novel morpho-molecular classification that explains major the inter-patient molecular differences, and promoted our summary **Fig. S31**, which summarizes our findings and our proposed classification using the example of 3 patients with similar clinical characteristics but vastly different survival and molecular profiles, to main **Fig. 6**.

2. in Fig 4, the use of the abbreviation ERG for epigenetic regulatory genes (ERGs) in the same list as altered genes will lead to confusion as this is also a gene name. I suggest that the ERG track be moved to the lower section with the type/sex/smoking/asbestos/WGD tracks.

Answer: Good point, we have made the suggested modification.

Reviewer #2:

Remarks to the Author:

I reviewed the authors' response to two of my comments and I am satisfied by their responses. I have also reviewed the manuscript in Biorxiv related to tumor-only analyses and we tested the related codes. It seems to work well particularly for SNVs (less for SVs).

However, there is no response to many of the additional comments I originally made. I am not sure why they have been ignored.

Answer: We thank the Reviewer for their thorough review of the manuscript and the code presented in the data note paper in Biorxiv, and for their positive assessment of our response to their first comments. We took great care to answer to the best of our ability to each and every comment we received, and would like to highlight that the reason for not having responded to the list of 22 individual comments was that they were not provided to us. We are very sorry for this unfortunate situation, and thank the Reviewer for having gone through the tedious work of parsing all the other Reviewer's answers carefully to find where their original comments were addressed. We also thank the Reviewer for prioritizing together with the Editor, the most urgent concerns that needed to be fully addressed (we have highlighted them below in yellow based on the information that we received from the Editor). While we have tried to address all the 22 comments as much as possible since the Reviewer made an enormous effort in performing such a detailed review that can only help improve our manuscript, we have paid particular attention to the depth by which we have addressed the highlighted comments below. We highlight in red novel text that was added for this second revision, and in blue, text that we already added in the first revision but that is linked with an answer to this revision.

Note that for this second revision, in order to limit the already very large amount of supplementary material in this manuscript, we have tried to find a balance between adding figures to the supplementary information (denoted as **Fig. SX**), when we felt that they carried critical information for the readers, and suggested adding them for this review only (denoted as **Fig. RX**), when we felt that they explored other important aspects of the data but were perhaps not critical for assessing the validity of our conclusions. We want to mention to the Reviewers and Editor that if the paper is accepted, we would like to select the transparent peer-review option so all responses to comments, including the figures therein, will be publicly available. Nevertheless, if the Reviewers or Editor feel that some particular figures should be included in the supplementary instead of only the rebuttal letter, we will be happy to include them.

See my comments in quotation "" for each original question.

This is the largest WGS analysis of MPM to date and the authors used unusual approaches for the analyses. In my opinion, the analyses require major revisions and validation (or at least comparisons of results) using alternative approaches. However, there are some important limitations that cannot be fully addressed even with major revisions (first of all, the lack of germline material as reference for most tumors).

Answer: This part has already been addressed during the first round of revision and we are glad that the Reviewer is satisfied by our answers.

This study aims to identify factors that can contribute to a new classification of malignant pleural mesotheliomas (MPM) with clinical implications. It has the largest MPM whole genome sequencing data to date, as well as RNA-seq and methylation data. The authors performed many analyses using approaches that are not typically used in cancer genomic analyses and with the potential to identify novel findings. They are laudable for their effort to tease apart MPM genomic, epigenomic and histological features to suggest a possible new classification. However, I have several

major concerns related to the analyses and data interpretation. I summarize the comments based on the analysis types:

1. Among the 120 tumor samples, >60% of WGS and 90% of RNA seq and methylation analyses lack matched normal or blood samples for the analyses. Although the authors attempted to address this issue with random forest-based and other approaches and even using additional normal samples from other datasets for some analyses, the lack of matched germline reference can affect the results. For example, TERT amplification in Fig.3a appears to be identified almost exclusively in tumor-only samples; and many analyses are presented based on the TERT amp in the manuscript. Another example is the deletion of RBFOX1. It would be important to systematically verify all major findings in the T/N matched samples (at least those based on WGS, since only a handful of normal tissue samples are available for RNA seq and methylation) and discuss discrepancies between matched and tumor-only results. "As mentioned above, a response was provided, although the authors did not specifically commented on the apparent enrichment of genomic driver events in tumor-only samples vs. T/N."

Answer: We thank the Reviewer for this important comment. As we have described in our previous response to Reviewers, and in our accompanying data note, we took care to implement a robust computational strategy to call genomic variants and alterations in tumor-only samples, however, we also note that some specific somatic genomic alterations appear to have different prevalence rates in the

tumor-only vs. tumor-matched samples. We believe this is primarily due to the proportion of different tumor types within the tumor-only vs. tumor-matched sample sets, whereby the nature of our cohort resulted in the tumor-only samples being enriched for sarcomatoid and biphasic types in comparison to the tumor-normal. This is relevant for specific genomic alterations that are more prevalent in particular types, such as *TERT* amplifications in sarcomatoid mesothelioma, and *BAP1* mutations in epithelioid mesothelioma, hence we would expect to find more *TERT* amplifications and less *BAP1* mutations in the tumor-only compared with tumor-normal samples. In order to reassure readers that there is no influence of missing normal material on genomic alteration rates, we have performed an analysis restricted to epithelioid samples, which are well-balanced between the two data types (38 tumor-only and 39 T/N matched). In **Fig. S33**, we show that recurrent somatic alterations are not enriched in tumor-only samples (Fisher's exact tests for all recurrently altered genes p -values > 0.05 except for one single gene out of more than 100 tests, even without multiple-testing correction, and all q -values > 0.05 after Benjamini-Hochberg adjustment; see new tab of **Table S13**). We now also include these results in the Methods on p.20, p.21, and p.22. In reference to the specific genes mentioned by the Reviewer, we did not find a significant difference in rates of *TERT* amplifications nor *RBFOX1* deletions between tumor-only and T/N epithelioid samples.

Figure S33. Comparison of the number of alterations in gene drivers between T/N matched samples and tumor-only samples of the epithelioid type ($n=77$ samples, 38 T-only and 39 T/N). (A) Small variants. (B) Structural variants. (C) Copy number amplifications. (D) Copy number deletions.

2.a In Fig.1a, methylation data in LF1, LF2, and LF3 explains the largest component of the variance. This could be due to the unbalanced size of the MPM datasets used for the analysis. In Fig. S1, it appears that the methylation data is three times larger than the RNA and DNA Alt data.

Answer: We thank the Reviewer for their comments (2a, b, and c) regarding the proportion of variance explained by MOFA, which has led us to improve our explanation, computation, and display of the proportion of variance explained in each omic type presented in **Fig. 1a**.

Fig. 1a was misleading because it implied, through grouping by latent factor and stacking the omic datasets, that the omics datasets explained variance within a factor. It appeared that with a value of 21%, the omics datasets explained 21% of the variance within LF1. Subsequently the reader could plausibly interpret that methylation would explain more variance within a latent factor, than for example RNA, simply because the methylation dataset was larger. The correct interpretation is in fact the opposite, that the latent factors explain variance within each omic dataset. We now have changed this representation to *barplots stacked by 'omic layer* (see revised **Fig. 1a** below) in order to clarify our message. In this figure, we show much more clearly that for a focal omic layer, such as gene expression (RNA), the value of 20% indicates that the four latent factors collectively explain 20% of the inter-patient variance within the gene expression dataset. Thus, the variance within a dataset that is explained by a given latent factor is independent of how well that latent factor explains the other six datasets, and therefore independent of their size.

Revised Figure 1a. Proportion of inter-patient variance within an omic type explained by histopathological type (left) and MOFA latent factors 1-4 (right). E.g. 7% of variation present in RNA expression can be explained by mesothelioma types, in contrast to 20% explained by integrative multi-omics factor analysis.

To illustrate in detail to the interested reader, we now also provide a detailed **Fig. S3** (see below) with a break down of all variance explained by each latent factor individually and combined, and also provide illustrative examples of molecular features that are badly explained by a latent factor (panel **A**) and well-explained by a latent factor (panel **B**), and typical features associated with each latent factor (panels **D-G**). We believe that these new representations better convey the underlying statistical model, where each molecular feature is the explained variable and factors are the *explanatory* variables (e.g., $\text{Expression_gene1} \sim \text{Factor1}$, $\text{Expression_gene2} \sim \text{Factor1}$), and thus each dataset (including methylation datasets) are treated independently. A detailed explanation has also been added to the Methods p.38-39.

New Figure S3. Proportion of inter-patient variance explained by MOFA latent factors. (A) Example feature for which a latent factor explains 0% of inter-patient variance (here factor 2 explains no variance at all in the expression of gene *NCR3*— $R^2=0$). (B) Example feature for which a latent factor explains most

of the variance (here factor 3 explains 87% of the variance of methylation site cg17731952– $R^2=0.87$). (C) Variance explained by the three histopathological types, each latent factor independently, predicted total variance explained by all latent factors together if they were completely independent (LF1 to LF4 *predicted*), and actual variance explained by a model including the four latent factors (LF1 to LF4 *observed*). (D) Typical Total CN feature associated with Factor 1. (E) Typical Enhancer Methylation feature associated with Factor 2. (F) Typical Enhancer Methylation feature associated with Factor 3. (G) Typical Gene Body Methylation feature associated with Factor 4.

2.b. This could also explain why the DNA Alt (driver genes and other genomic features) appears not to explain any variance component. How much variance is explained by each data type?

Answer: Regarding the small proportion of variance in DNA alterations explained by each latent factor, this is an artifact due to the difficulty to properly account for the proportion of variance explained for an indicator variable, and we agree with the Reviewer that this misleadingly suggested that MOFA did not explain DNA alterations. Indeed, for consistency with the other omic types, we originally displayed the R^2 of a linear model of the form $\text{alt}_i \sim \text{Factor}_k$, where alt_i is an indicator variable (0: wild type, 1: altered). Nevertheless, classical R^2 for such variables are known to have a theoretical upper bound strictly smaller than 1, and in our case this maximum is often below 0.5 (black line in **Fig. R1A** below), which leads to the false interpretation that it is impossible to explain 100% of the variation in DNA alterations. We now rather display in **Fig. 1a** the pseudo R^2 of McKelvey and Zavoina, using the Veall-Zimmermann estimator (**Veall and Zimmermann, *Qual Quant*, 1992**), and we explain p.39 that this pseudo- R^2 has a similar interpretation to classical R^2 (proportion of variance in an indicator variable explained by a feature) but it is based on a logistic regression model which better models indicator variables. See also in **Fig. R1B** below an example where wild-type and altered samples are maximally separated by a MOFA latent factor (all altered samples are at one extreme of the factor), but the classical R^2 is very low (0.085) while the pseudo R^2 correctly indicates a perfect fit (1.000). We also show in a new **Fig. S3C** (see above) a comparison between pseudo R^2 and R^2 for DNA alterations, which shows that DNA alterations are actually explained almost as well as the other omic types by MOFA latent factors.

Figure R1. Comparison of R^2 and pseudo R^2 for estimating the proportion of variance in DNA alterations (SNVs, indels, and SVs) explained by MOFA factors. A) Observed R^2 (black) and pseudo R^2 (red) of each alteration (points), compared to their theoretical maximal values (solid lines), as a function of the number of samples carrying this alteration, for each factor. B) Example gene whose alteration is maximally separated by MOFA Factor 3, but R^2 is very small while pseudo R^2 is 1. Pseudo R^2 are computed using the Veall and Zimmermann (1992) estimator, which was shown to be the closest to ordinary linear regression R^2 and analogous to a proportion of variance.

2.c. How much variance could be explained by including a larger number of LFs? Are these 4 LFs statistically independent from each other?

Answer: We agree that this is important information and thus now directly show in **Fig. 1a** the total proportion of variance explained by the first four latent factors, selected for subsequent analysis: 20% of RNA variance, 27% of promoter methylation variance, 31% of body methylation variance, 46% of enhancer methylation variance, 61% of total copy number variance, 21% of minor copy number variance, and 19% of DNA alteration variance. This increases to 35% (RNA), 41% (meth. pro), 46% (meth. bod.), 67% (meth.

enh.), 85% (Major CN), 62% (Minor CN), and 47% (DNA alt), when including all 10 LFs. This information is now reported in **Table S2**.

The four LFs that have been retained for further analyses are statistically independent of one another, as shown in **Fig. S6C** (new in revision 1). This independence can also be visualized in **Fig. S3B** (new in revision 2). We expect under statistical independence of the factors, that the R^2 of a model combining all factors of the form $feature \sim Factor1 + Factor2 + Factor3 + Factor4$ (see “LF1 to LF4 predicted” in **Fig. S3B**) would be the sum of the R^2 of each independent model $feature \sim Factor1$, $feature \sim Factor2$, $feature \sim Factor3$, and $feature \sim Factor4$. In contrast, if the factors were completely colinear, the R^2 of all independent models would be equal, and the R^2 of the combined model would be equal to that of any of the independent models. In **Fig. S3B**, we observe that the R^2 of the combined model is very close to that expected under perfect statistical independence (“LF1 to LF4 expected” and “LF1 to LF4 observed”, **Fig. S3B**), corroborating the statistical independence of the factors for each omic dataset.

Of note, some of the other latent factors are not statistically independent (LF5 and LF9), none were found to be associated with survival as presented in **Fig. S7A** and their replication in other cohorts has been assessed as difficult as presented in **Fig. S2**. For these reasons, we have focussed on the first four latent factors, although we now note in p.39 that with larger cohorts, future studies might reproduce some of these other factors and have more power to detect their influence on survival.

2.d. How sensitive is MOFA to sample size (although this is the largest WGS study of MPM to date, is still small). And the Bueno cohort used for replication lacks methylation data, so it is not ideal to replicate an analysis mostly driven by methylation data. “No answer was provided. The authors need to address the issue of unbalanced MPM datasets”

Answer: With the additional work on replication that we produced for revision 1, we believe that we have also demonstrated in this manuscript that MOFA can be quite robust to changes in sample size. The latent factors identified in our MESOMICS analysis ($n=120$) were also recapitulated in two other individual cohorts of different sample sizes: Hmeljak et al, $n=73$, and Bueno et al. $n=181$, as well as a combined 3-cohort analysis of $n=374$ (**Figs. S1** and **S2**). The Bueno cohort has been used to replicate findings that are not only driven by methylation data. As the Reviewer noted, LF2, LF3, and LF4 explain a large proportion of variance within this type of omic data, but even without methylation data in the MOFA Bueno, we succeeded in replicating the findings related to LF2 and LF3. This highlights that tumor type and immune infiltration have important, detectable impacts outside of DNA methylation profile. Additionally, although the identification of LF4 was challenging, without a CIMP index to validate its existence in Bueno, we were able to replicate LF4 and CIMP findings using the TCGA cohort (see **Table S6** for further details on the replication analyses). We now also mention in the methods p.36 that our sample size is in line with the size of $n=100$ that was shown to allow to capture the main sources of variation in simulated datasets in

the original MOFA study, across a wide range of omic datasets (1 to 21), features per dataset (100 to 10,000), latent factors (5 to 60), and missing values (from 0 to 90%) (Argelaguet et al. 2018); we also mention that $n > 100$ fits general recommendations for dimensionality reduction based on matrix factorization such as PCA for stable latent factors and weights so the results from the sample can accurately be generalized to the population (Saccenti *J Proteom Res* 2016).

3. In Fig.S5, LF1 was associated with RNA-seq batch effects and LF4 (CIMP factor) was associated with sample wells (at least this is my interpretation since there is no legend). Did the authors consider these factors in their analyses? How? “No answer was provided to this comment, although the CIMP factor was discussed in response to Reviewer 1. Also, in the revised manuscript (now Figure S6d), the association is not there anymore.”

Answer: As the Reviewer noted, the legend for **Fig. S5C**, in the first submitted version of the manuscript, was missing and we apologize for this oversight. The light pink in the figure indicated a q -value of between 0.05 and 0.1, therefore a borderline significant association between LF1 and LF4 and some batch variables. During the first round of revisions we made a correction to the MOFA input matrices specifying copy number levels concerning 0.5% of segments, following this, the quality checks presented in **Figs. S6C** and **D** were repeated and the associations found to be q -value > 0.1 (now in **Fig. S7D**), therefore they weren't corrected for in any subsequent analyses.

4. In general, it would be important to validate or at least compare this LF classification using other approaches, e.g., iCluster or Cluster-of-clusters, typically used in genomic analyses of other cancer types similarly composed of mixed histologies and molecular features. “The authors responded to this question”

Answer: We thank the Reviewer for their suggestion and have performed the proposed analyses during the first revisions.

5. The strongest cancer prognostic factor is tumor stage, but the survival analyses do not appear to have been adjusted by stage. This could strongly bias the results. All survival analyses should include stage in the model. “No answer was provided”

Answer: We thank the Reviewer for raising this important point, and we note that tumor stage and histological subtype are currently the main known prognostic factors for mesothelioma. Unfortunately, we have not been able to adjust the survival analyses on tumor stage due to the lack of staging data in the database MESOBANK, from where the MESOMICS samples come from. Staging data was not included in the French MESOBANK as this information was not systematically recorded in clinical records in France at the time. During this period (year of diagnosis [1998-2017], median of 2011), there was a lack of pleural staging system accepted by the French community of clinicians, and an evidenced-based revision of the staging system for mesothelioma was not proposed until 2016 by the IASLC Mesothelioma Staging Project (**Pass et al. J Thorac Oncol. 2016**). From our discussions with pathologists involved in the MESOBANK, the main reason seems to be the limited impact staging would have in treatment-decision making given the dismal prognosis of the disease, with no therapeutic options. In addition, there was no consensus on staging systems worldwide in 1996 where new staging proposals were published for the first time (**Rusch et al. J Thorac Cardiovasc Surg. 1996**). This proposal was followed by an evidence-based revision of the staging system from and before 1995-2016 and published later in 2016 by the IASLC (**Pass et al. J Thorac Oncol. 2016**). Moreover, when staging was carried out several staging modalities were in progress, such as IMIG staging or UICC, and there were some discrepancies between them. This might also be the reason why stage was not included as a covariable when comparing the survival of different types or molecular groups in previously published mesothelioma genomic analyses (e.g., no adjustment in **Bueno et al. Nat Genet. 2016** nor **Hmeljak et al. Cancer Discov. 2018**). However, we would like to note that Hmeljak et al. show in their **Fig. S1** that T stage was not significantly associated with survival in their cohort. We are very sorry not to be able to address this Reviewer's concern but we hope they will understand that this is beyond our capabilities since the data is not available for us to integrate it, additionally we now mention this in the Methods p.15.

6. In the survival analyses in Fig.1e the strongest effect is provided by LF2, which corresponds to the known morphological classification. The other latent factors do not appear to strongly affect survival beyond this factor. In Fig.S5, in the analyses restricted to the epithelioid group, only LF3 is (marginally) associated with survival. In Figure S6, 22 Cox proportional hazards survival models were tested: how did the authors take multiple testing into consideration? In the cross-validation results, the combination of all 4 factors does not substantially improve the AUC (0.79) over the combination on any of three factors (0.78-0.80) or two factors (0.73-0.78). And this seems to be true (although with different estimates) also for the TCGA data. Also, based on the Introduction, aneuploidy and immune infiltration had been already identified as playing a role in MPM, and yet the morphological classification remained as the major classification variable. Thus, it is not clear whether the addition of the data related to the LF1, 3, and 4 can be useful in clinical settings or could be considered for treatment strategies. "No answer was provided, although the authors tangentially addressed this issue in response to Reviewer 3."

Answer: We thank the Reviewer for this comment. In this manuscript, we aim to highlight the importance of the sources of molecular variation we have identified in addition to tumor morphology. We hope that the modifications and additions we provided during these two revisions will emphasize their biological significance.

In the re-submitted version of the manuscript, following a small correction to the MOFA input matrices specifying copy number levels concerning 0.5% of segments, LF2 was no longer the axis with the largest Cox model effect size (**Fig. 1f**), therefore all four LFs are important factors in prognosis. Whilst, LF3 was the only axis reaching statistical significance in the survival analysis restricted to MME samples (**Fig. S7B**), importantly, as the Reviewer noted, we showed in the revision that the four factors are statistically independent (**Fig. S7C**), and LF1, LF3 and LF4 are not associated with the histological type (**Fig. 1b**), thus we hypothesize that a lack of statistical power prevents LF1 and LF4 from reaching statistical significance when examining MME samples only. Nevertheless, to legitimate the significance of these three additional axes of molecular variation, we successfully replicated their association with survival in additional cohorts, importantly using more clinically-suitable proxies (**Fig. S6**).

Concerning the twenty-two Cox models presented in **Table S5**, we addressed the importance of multiple testing by using cross-validation (on MESOMICS, **Fig. S8A-C**) and bootstrapping (on TCGA, **Fig. S8D-F**) approaches to provide the standard error of iAUC for each model to let the reader evaluate their robustness as predictive models. Still, we agree that the difference in iAUC between the models using MOFA factors is subtle. However, the difference is more substantial when compared with previously published prognostic factors (**Fig. S8C and F, left panel**), and the increase of iAUC when using MOFA factors together suggests the complementarity of these axes to predict patient survival. Additionally, the highest increase in AUC is observed when using this 4-factor model in MESOMICS (**Fig. S8A**) and TCGA (**Fig. S8D**) in the short-survival patients (≤ 1.42 years) when most deaths occur, and in the MESOMICS cohort, this model outperforms all others even in longer-survival time analysis (**Fig. S8A**).

As the Reviewer has pointed out, it is indeed critical that research continues to focus on improvements that could be made to clinical management, and continuing to refine histological classification is not likely to provide benefit to patients. Therefore, new approaches are needed such as those presented in the current study. The interpretation of the latent factors and how they may be used in a clinical setting is not straightforward, therefore we made several additions to the first version of this manuscript during revision to improve this aspect of the work. As described in the revised manuscript (p.4 and Methods) and in our response to Reviewer 3 comment 4, in **Fig. S6** we represent and validate simpler proxies for each factor, demonstrating the variability in ploidy, immune infiltration, and CIMP index, across the morphological types, highlighting the importance of investigating these features in addition to morphology, and how proxies of these have prognostic value in additional mesothelioma cohorts. Importantly, in **Fig. S10 and 19** we further investigated mesothelioma cell lines and their response to drug

targets based on their position along latent factors and genomic alterations, highlighting the complexity of the interplay between molecular features and drug sensitivity. Finally, we also now provided a summary figure (**Fig. 6**, see Reviewer 1 comment 1 answer) to describe with concrete examples the novelty of the classification suggested in this manuscript. This new main figure illustrates how our novel classification may explain inter-patient differences in three patients with similar clinical characteristics but vastly different molecular profiles. We believe that this work demonstrates the potential benefits of establishing a morpho-molecular classification for clinical management of mesothelioma.

7. There is a large difference in terms of purity estimated from RNA-seq, WGS, and by pathology (no purity estimates are reported based on methylation data) in Supp Table 1. How did the authors reconcile the purity differences and which purity estimates were used for the analyses? How much does tumor purity affect MOFA results? Is any LF correlated with tumor purity? In addition, purity was included as an additive covariate in the regression analyses, but others have shown that a multiplicative model would be more appropriate (see Zheng et al. 2017 Genome Biology for example). “No answer was provided.”

Answer: We thank the Reviewer for their comment and wish to clarify our analyses. We received a similar comment from Reviewers of the data note, therefore we now provide for the revision of the data note additional data, including a figure comparing the purity estimates (see **Fig. R2** below attached with this review). This figure shows that the three purity estimates were correlated with q -values < 0.01 , although they show important differences. We have also added several sentences in the data note explaining that the three purity measurements estimate different aspects of purity: the proportion of DNA material from the tumor for the WGS-estimated values, the proportion of infiltrating immune cells for the RNA-seq, and the amount of tumor tissue in the observed slide for the pathological estimate. Thus, we paid particular attention to mention what is the source of each purity estimate used throughout the Methods, and chose to use WGS-estimates whenever we wanted to quantify or control for the proportion of normal cells in the bulk, chose the RNA-seq estimates (that we refer to as immune infiltration) whenever we wanted an estimate of the amount of non-immune cells in the bulk, and used the pathological estimates for example, prior to sequencing, to determine if sufficient tumor content was available to include the sample.

Tumor purity is associated with some aspects of the MOFA. The morphology and CIMP index factors are associated with purity as estimated by RNA sequencing data (q -value=0.02 and $r=0.26$, and q -value=0.002 and $r=0.33$, respectively), indicating some influence on tumor immune infiltration on sample position along these latent factors. Importantly, samples with high purity in this estimate (i.e. low immune infiltration) are found distributed across these two latent factors and are therefore driven by aspects of molecular variation other than simply immune infiltration (**Fig. R3**). Furthermore, neither WGS-estimated purity nor purity estimated by the pathologist were associated with these latent factors. In contrast, and

as anticipated, the adaptive-response factor is associated with purity as estimated by the pathologist (q -value= 6.59×10^{-5} , $r=-0.39$), and with WGS (q -value= 9.84×10^{-9} , $r=-0.54$), and RNA sequencing (q -value= 3.14×10^{-11} , $r=-0.62$) estimates, reflecting the importance of non-tumoral cells, primarily immune infiltration, on the position of samples and their task specialization.

Regarding the multiplicative model proposed by Zheng and colleagues, we thank the Reviewer for this useful reference. We indeed used a multiplicative model to correct for purity in MOFA (see methods p.35), but originally did not when performing the differential expression analysis. We have therefore taken the suggestion on board and repeated our analysis of differentially expressed genes between whole-genome doubling positive and negative tumors, this time using a multiplicative model. We now show in a new panel **Fig. S26E** (see below and Methods p.28) that using a multiplicative model to incorporate the effect of purity on gene expression, while not changing our conclusion regarding the impact of WGD on immune response, in particular through interferon down-regulation, seems to improve our power to detect differentially expressed pathways.

Figure R2. Correlation between purity estimates from three different omic purity measurements: the proportion of DNA material from the tumor (genomic estimate of purity), the proportion of infiltrating immune cells (transcriptomic estimate of purity), and the amount of tumor tissue in the observed slide (pathological estimate of purity). (A) between transcriptomic and pathological estimates, (B) between genomic and pathological estimates, and (C) between genomic and transcriptomic estimates. In these three panels, q -values and coefficients r correspond to Pearson’s correlation tests. All measurements of purity were significantly correlated.

A
Up-regulated in WGD+ (137 genes) Down-regulated in WGD+ (4130 genes)

E
Up-regulated in WGD+ (200 genes) Down-regulated in WGD+ (3292 genes)

Figure S26. A and new panel E. Pathways up- and down-regulated in WGD+ samples, accounting for additive (top) or multiplicative (bottom) effects of purity on gene expression.

Figure R3. Association between MOFA latent factors and estimates of purity. In these three panels, q-values and coefficient r correspond to Pearson's correlation tests.

8. In the Pareto TI analysis, outlier tumors based on gene expression are known to interfere with the position of the archetypes. Did the author verify whether there were any outlier tumors? “No answer was provided.”

Answer: We indeed see that we forgot to mention in the methods that we used a bootstrapping approach (200 bootstraps each subsampling 75% of the data with replacement) to estimate archetype positions robust to outliers; we now mention it on p.42 and refer to the code on our github repository (https://github.com/IARCbioinfo/MESOMICS_data/blob/main/phenotypic_map/MESOMICS/Phenotypic_Map_MESOMICS.md).

In Fig2a, the authors described that only LF2 and LF3 create the “triangular shape delimited by 3 extremes”, not ploidy or CIMP factors. Can the results based on ploidy and CIMP be presented so one can verify the results and compare them with LF2 and LF3? “No answer was provided.”

Answer: We thank the Reviewer for this comment and now provide further precisions on which combination of latent factors the Pareto algorithm was tested in p.42. In fact, after sorting the factors with the highest percentage of variance explained from the RNA dataset, the algorithm increases the number of factors to test together with the number of archetypes to find, meaning that in our case the algorithm performed tests on LF2-LF3, LF2-LF3-LF4, and LF2-LF3-LF4-LF1 spaces and never directly LF1-LF4. For each set of factors and number of archetypes to find, the algorithm computes a p -value from the Pareto model fit and the method described by Hausser *et al. Nat Commun. 2019* indicates that if several models are significant, the users should select the model with the lowest number of archetypes. We now provide the fit results for each model tested by the Pareto algorithm in **Table S8**. Additionally, we also provide an illustration of the non-significant (p -value=0.971) fit in LF1-LF4 space in a new **Fig. S11**.

New Figure S11. Comparison of the best Pareto front fit in (A) morphological factor (LF2) and adaptive-response factor (LF3) space and (B) ploidy factor (LF1) and CIMP factor (LF4) space. The scatter plots represent sample positions along each factor and coloured vertices the position of the archetype defined by the ParetoTI algorithm in each space.

9. There is a large discrepancy between CIMP-index (32-56%) and CIMP-normal index (1.3-19%). How do the authors explain this large difference? The range of CIMP in tumors is based on beta values >0.3. This is not the typical way the CIMP phenotype is identified. CIMP is important because exhibits highly recurrent hyper-methylation of TSS genes. Can the authors replicate the CIMP analyses using unsupervised clustering methods and evaluate how the LFs vary by CIMP subtypes? “No answer was provided, although the authors discussed the CIMP factors in response to Reviewer 1.”

Answer: In order to clarify our analyses with regard to this phenotype we have provided additional details about the CIMP index, below, and in the study methods (section: *CpG island methylator phenotype index*, p.31).

Firstly, we have been deliberate in not describing this finding in MESOMICS as a CIMP+ phenotype as the method we have used to investigate CpG island methylation level, based on DNA methylation array data, differs from the classical gene panel model assessed through methylation-specific PCR (Weisenberger *et al. Nat Genet. 2006*; Hughes *et al. Cancer Res. 2013*). Instead we refer to our measurement as a CIMP *index*, with a continuous rather than categorical interpretation. The Reviewer

refers to an additional method for calculating a CIMP index, the CIMP-normal index, which we tested in the MESOMICS cohort as it was previously used in a mesothelioma series (Blum *et al. Nat Commun.* 2019). In this method, probes located within CpG islands were retained, and the mean beta value across all probes within each island was calculated for the three adjacent normal tissues available in the MESOMICS cohort. Islands whose methylation level was <30% in all three adjacent normal samples were retained ($n=15,824$), denoted as normally hypomethylated islands. The CIMP-normal index was then calculated as the proportion of these 15,824 islands with $\geq 30\%$ methylation (beta value ≥ 0.3) per sample. CIMP-normal index values ranged from 0.013 to 0.19, corresponding to 0.13% to 19% of normally hypomethylated islands to be hypermethylated per sample. These values differ from the CIMP index presented in the manuscript as they are calculated from a subset of CpG islands present on the array (those typically hypomethylated in normal tissue), in contrast to all islands on the array as are used for the CIMP index calculation. Although these values differ, there is a significant correlation between CIMP index and CIMP-normal index values ($p\text{-value}=3.27\times 10^{-66}$, $r=0.96$). The CIMP index, rather than the CIMP-normal index, was used for subsequent analyses as the method for CIMP-normal index was based on first identifying normally hypomethylated islands, therefore requiring normal pleura. The 'normal' tissues available in the MESOMICS cohort are material adjacent to mesothelioma tumors, therefore they are unlikely to be pure non-tumour pleural tissue.

A further method used to identify CIMP-high samples has been to perform unsupervised clustering over variably methylated probes across a cohort. To address the Reviewer's concerns for additional replication, we have performed unsupervised k -means consensus clustering on the 8,000 most variable probes (calculated from standard deviation across m -values) with 1,000 iterations over random subsamples of 80% of the 8,000 probe set (R package ConsensusClusterPlus) similarly to Sturm *et al. Cancer Cell.* 2012. As anticipated from additional clustering analyses performed in response to the Reviewer's concerns about our methodological approach using MOFA (presented in Fig. S34), consensus clustering of methylation array data only also resulted in unstable clusters where $K>2$ (Fig. R4). We further performed linear regression analyses to examine relationships between the methylation clusters identified, and MOFA latent factors and CIMP index. This identified that whilst consensus clustering with methylation data resulted in clusters significantly associated with sample position along latent factors ($q<0.05$), and CIMP index, it does not clearly capture the continuous nature of the mesothelioma methylation profile (Fig. R5). Therefore, we believe the most appropriate way to investigate any potential CpG island methylation phenotype is with a continuous index-based model, as presented in the manuscript.

Figure R4. Consensus matrix heatmaps for $K=2$ to $K=5$, generated from 8,000 most variable probes for $n=119$ samples.

Figure R5. Sample position along MOFA latent factor 4 (CIMP-factor) (top row), and sample CIMP index value (bottom row), coloured by methylation class identified through consensus clustering over 8,000 most variable probes for $n=119$ samples.

10. The TCGA-MOFA methylation analyses are based on the arrays with ~450K probes and appear to replicate the MESOMICS analyses based on the EPIC array with ~850K probes. Can more features in enhancer probes (enriched in the 850k EPIC array) improve the MOFA analysis? “No answer was provided.”

Answer: Yes, the Reviewer is correct that the TCGA and MESOMICS cohorts have different types of Illumina Methylation Beadchip Array data available. In MESOMICS we have used the EPIC array, which was designed to cover an increased number of regulatory regions including enhancers. In order to try and address the Reviewer’s query, we have examined which of the probes input into MOFA would also be found on the previous iteration of the Illumina array, the HM450K. Of the 5000 promoter-associated probes incorporated, 3849 (77%) of the probes are also on the HM450K array, and of the 5000 body-associated probes 3878 (78%) are on the previous array, this is in contrast to just 1520 of the 5000 enhancer-associated probes (30%) being available on the HM450K array. This indicates that many enhancer regions which display highly variable methylation patterns between mesothelioma samples are not captured by the HM450K array, and the EPIC array therefore provides us with perhaps a more complete picture of methylation patterns in this cancer. This information has now been included in the Methods section on p.35.

It is difficult to say whether the inclusion of additional probes *improves* the MOFA analysis, as we did not consider testing the difference between inputting EPIC probes vs. HM450K probes into the model given that we generated the more exhaustive EPIC array data for this study. Whilst we feel such a technical test is outside the scope of the purpose of our work, we can consider our replication analyses with the TCGA dataset (HM450K) as partially addressing this query. With the TCGA data, we were able to replicate many of our methylation-based findings including recapitulating MOFA latent factors 2-4 (those explaining much of the variation within the methylation datasets, **Fig. S2**), variable CIMP index (**Fig. S4**), and associations between latent methylation components hidden in enhancer-associated probe profiles with sarcomatoid, epithelioid, and immune cell proportions (**Tables S2 and S6**). In summary, we believe that the EPIC array provides additional information on enhancer methylation profiles in mesothelioma over and above the HM450K array, however our findings related to methylation profile are robust given their validation in the TCGA cohort.

11. Notes: lines 1077-1078 report 2 different estimates (1.3% or 0.0013?); lines 1125-1127: the number of probes used do not match: $150 + 207 + 2446 = 2803$ not 3764 probes. “No answer was provided.”

Answer: We apologize for this error in calculation, the correct values are as follows: “This resulted in 3,764 probes across all 77 genes, specifically 153 promoter probes corresponding to 17 EMT genes, 209 enhancer probes corresponding to 54 EMT genes, 2,575 body probes corresponding to 77 EMT genes, and an additional 827 probes not annotated to promoter, enhancer, or gene body regions, corresponding to 73 genes.” We have corrected this sentence in the manuscript Methods on p.33.

12. I am not clear how the tumor evolution was estimated across all samples. Do the different LFs identify tumors with different evolutionary trajectories? Fig5d only describes 6 samples. Was the timing of the lesions only conducted in these 6 samples? Four out of 6 are linked to TERT amp (likely based on tumor-only samples). And was asbestos exposure only ascertained in one sample (grey area)? It would be important to time the asbestos exposure in relation to different genomic changes to try to understand whether the asbestos exposure was involved in the initiation or progression of the tumors or had no detectable effect on tumor evolution. “No answer was provided, although the authors acknowledged the limitations of their approach in response to Reviewer 1.”

Answer: As the Reviewer noted, we now explicitly state in p.10 that we studied “timing of WGD, *TERT* amp, and copy neutral LOH in the few samples ($n=6$) with such events where a subclonal deconvolution was possible,” and we also note in the Methods p.43 that this analysis focused on T/N matched samples. We now also note in p.10 that asbestos exposure was available for 5/6 samples (2 non exposed, 3 exposed) and that 2 had reported periods of exposure, one encompassing the estimated timing of the LOH event and one more than 50 years before the estimated timing of the *TERT* amplification, although we note that such results based on a few samples need to be confirmed in much larger series. Regarding the association with the different LFs, we now mention on p.10-11 that we detect a significant enrichment of neutrally evolving tumors at the extreme of the morphology and adaptive-response factors.

13. For the gene expression analyses, did the authors take into account copy number alterations (beyond WGD), which could substantially affect gene expression levels? “No answer was provided.”

Answer: We agree with the Reviewer that copy number can strongly affect gene expression, and we indeed used copy number estimates as suggested by the Reviewer in the WGD analyses because we were then interested in finding a “WGD effect” on gene expression that would be beyond what is expected from just having an increased number of copies. We now mention in the other expression analyses (namely, the archetype expression analyses p.43) that in that particular case, because we are interested in reconstructing a genotype-to-phenotype map, we actually want to first visualize the resulting

phenotype that therefore includes gene expression modulated by copy number changes (see **Fig. 2**) and second, to identify genes which amplification or deletion directly impacts gene expression (see **Fig. 5**) and thus purposely avoided correcting for copy number changes, which would have removed the signal and thus potentially lost the interesting biological impact of copy number changes on tumor specialization.

14. How did the authors identify 3 specific cancer tasks from IGSEA analyses? Can results from other pathways be shown (Table S7 only reports the 3 archetypes)? “No answer was provided.”

Answer: We ran IGSEA for each archetype identified by using the association between gene expression and archetype proportion (see Methods for more details) resulting in a set of enriched pathways (GO terms). Inspired by **Hausser et al. Nat Commun. 2019** methods, and from this list of enriched pathways, we annotated groups of GO terms under the same biological function that corresponds to known cancer tasks. We also indeed noticed from the comments of Reviewer 1 that the **Fig. 2c** display was misleading, and changed it for revision 1, and we believe that it is now more adequate to show how cancer tasks are inferred. Indeed, we report all the significant pathways both from the hyper-pathways and from other pathways, to show that these hyper-pathways constitute a large proportion of enriched pathways. We also report in **Table S8** all pathways and mention which (if any) hyper-pathway they belong to.

15. IGSEA results are confusing: for many gene sets, there only 10-15% of genes overlap with a given pathway, but they reach very low p-values. For example, for downregulated pathways - myogenesis has 11/200 overlapping genes but has much lower p-value (3.90E-07) than the down-regulated pathways in the xenobiotic metabolism with 33/200 genes overlap (p=0.02). Can the authors clarify these findings? “No answer was provided.”

Answer: We indeed forgot to mention the exact test that we performed (hypergeometric test), and that we performed the tests separately on upregulated and downregulated genes. We now mention this in Methods p.28 and write that this explains the counter-intuitive result the Reviewer mentions regarding results in **Table S3**: “Note that because the number of significantly upregulated and downregulated genes had different orders of magnitude (137 upregulated vs. 4129), the proportion of overlap expected by chance are much higher in the downregulated GSEA while even modest overlaps (e.g., 6 genes) in the upregulated GSEA are sufficiently surprising to be statistically significant.”

Table S3 is confusing (and there are 2 “Table S3”). It’s unclear what is the mean expression between the two groups. For example, how were genes with different expression estimate signs labelled (up/down regulated)? In addition, gene names annotation should be included. Similar issues are in Table S7.

Answer: We have clarified **Table S3**, which indeed contained many columns with confusing names. We have added gene names as suggested and shifted the column comparing WGD+ vs. WGD- groups to the beginning (left) to make the difference in mean expression between the two groups more visible, and explicitly separated up- and down-regulated genes in WGD+ samples. We apologize for the confusion about the multiple tables, the Nature submission tracking system unfortunately only authorizes tabs from a same table to be uploaded separately, hence the many different tables with similar names; this issue should be resolved at publication with the different tabs having explicit names related to their content (here, “differentially expressed genes” and “GSEA results” to separate gene-level and pathway-level results). For the IGSEA performed on archetypes the down/up regulated genes have been defined by the sign of the Pearson’s correlation coefficient from the test between gene expression and each archetype proportion (the highest proportion the closest the position to the archetype is). In **Table S8** (former **Table S7**), we also annotated each enrichment with a fold change estimated between the 10% closest samples vs. the 10% furthest samples from each archetype, similarly to **Hausser *et al. Nat Commun. 2019*** methods. We now show the gene’s HUGO symbols in **Tables S3** and **S8**.

Samples with DNMT3B and EZH2 mutations - are they overly expressed in the tumors? “No answer was provided”

Answer: Within the MESOMICS cohort, three samples have mutations in *DNMT3B*, two with nonsynonymous SNVs and one with a stopgain mutation, whilst no samples harbored an *EZH2* mutation (**Table S13**). *DNMT3B* mutation was not significantly associated with gene expression ($p=0.076$, linear regression model adjusted for tumor purity estimated by PURPLE, see **Figure R6** below attached to this review), and its effect was not in the direction of overexpression.

Figure R6. Density of *DNMT3B* expression in MESOMICS cohort. Density of *DNMT3B* expression over 109 MPM (MESOMICS cohort). Blue lines designate the expression level of three samples harboring damaging *DNMT3B* alterations.

16. The authors calculated a genomic instability score using CNV, gene expression, and methylation data, but I am not clear on how these scores were combined or which score was used for which analyses. It seems that there is no great correlation between the scores calculated from different omics in Supp Table 2. “No answer was provided.”

Answer: We thank the Reviewer for their comment and wish to clarify our analyses. We investigated genomic instability within the Cell division archetype (p.6) using proxies from the three available omic data types. We performed Pearson’s correlation tests between each archetype and each score of instability. Therefore, these proxies have not been combined and we reported individual p -, q -values, and correlation coefficients for these tests in an additional supplementary table (**Table S6**, line 33) resulting from the first revision of this manuscript. Individually, and after multiple-testing correction, each of these

proxies was significantly associated with the Cell division archetype. In addition, we replicated these associations in the Bueno and TCGA series (see **Table S6** for details). We now also mention the correlation between these scores p.33-34 (see also **Fig. R7** below for an illustration). Although each of these pairwise correlations is significant (q -value= 1.53×10^{-5} ; q -value= 6.63×10^{-3} ; and q -value= 1.39×10^{-3} for genomic vs. transcriptomic, genomic vs. epigenomic, and transcriptomic vs. epigenomic scores, respectively), we observe differences that we think can be explained by the fact that each proxy captures a part of genomic instability that can be partially independent: the proportion of the genome with copy number changes, the expression of genes that are involved in genomic instability pathways, and the low global methylation level of the genome that is associated with instability.

Figure R7. Correlation between genomic instability from three different omic measurements: the proportion of CN changes (genomic estimate of genomic instability), the transcriptomic score of genomic instability gene set (transcriptomic estimate of genomic instability), and the global DNA methylation level (epigenomic estimate of genomic instability). (A) between genomic and transcriptomic estimates, (B) between genomic and epigenomic estimates, and (C) between transcriptomic and epigenomic estimates. In these three panels, q -values and coefficient r correspond to Pearson's correlation tests.

17. In Fig.3A, MPM driver gene overview, the top genes BAP1 and NF2 are both identified as having large deletion events (mostly heterozygous deletions). What is the relationship between these heterozygous deletions and SV (and SNV) events? The gene expression plot for figure 3a should separate different alteration types for these two genes. "No answer was provided."

Answer: We thank the Reviewer for this comment. Indeed in the MESOMICS samples, we found 46 samples with copy number deletions in the *BAP1* gene and 14 samples with damaging SVs. Thirteen of the 14 damaging SVs (one interchromosomal translocation and 12 deletions) occurred in conjunction with a

BAP1 copy number deletion. In addition, among the 46 deleted samples, 9 also display a damaging SNV, and one sample displays all three kinds of events. Similarly, we found 69 samples with copy number deletions in *NF2* and 20 with damaging SVs that all co-occurred with copy number deletions (one inversion, three interchromosomal translocations, two duplications and 14 deletions). In addition to these 20 cases, 8 display damaging SNVs.

We apologize that the legend of **Fig. 3a** was not precise enough on how the tests presented on the right subpanel were performed. In fact, the expression level comparison has been done with true wild-type samples, meaning we excluded cases with SVs or SNVs in these genes for this WT group of samples (see **Table S6** for details on statistical tests and replication). We now specify this in the **Fig. 3a** legend. However, the group of samples with copy number deletion include cases also with SVs and/or SNVs corresponding to 46% of the deleted cases for *BAP1* and 41% for *NF2*. Therefore we have now reproduced our analyses and excluded these cases, and found the same significant differences in gene expression with q -values ≤ 0.0001 for *NF2*, *MTAP*, and *BAP1*, and WT vs. Heterozygous, and WT vs. Homozygous for *CDKN2A* (no longer a significant difference between Heterozygous and Homozygous *CDKN2A* deletions). These new results are now reported on p.27

18. 23% of the samples have homologous recombination-deficiency (HRD) phenotype. Given the potential clinical implications of this finding, can the authors verify these results with other approaches (e.g., CHORD, HRDetect)? Are there any germline variants in HRD genes in these samples? “No answer was provided, although the authors reported previous cell lines results and clinical trial data to support their findings on HRD in response to Reviewer 1.”

Answer: We indeed did use two methods to detect HRD samples: CHORD, as the Reviewer suggested (see results in **Table S11**), and the new copy number signature recently identified as associated with HRD (**Steele et al. *bioRxiv*. 2021**). We thank the Reviewer for their suggestion of checking germline variants in HRD genes. We now mention on p.7-8 and in the Methods p.20 that we looked for pathogenic variants (from the CLINVAR database) in a list of 26 HRD genes (from **Toh and Ngeow, *The Oncologist*. 2021**) and indeed found 6 variants reported as pathogenic (1 in *BRCA1*, 3 in *BRCA2*, 1 in *ATM*, and 1 in *NBN*), all reported as being linked with a Hereditary cancer-predisposing syndrome. As expected, these variants were significantly more common in HRD tumors compared to non-HRD tumors (only a frameshift deletion in *NBN* was found in non-HRD tumors; odds ratio of 20, Fisher’s exact test p -value=0.002). In addition, we note that the *BRCA1* and *BRCA2* germline-mutated tumors indeed were correctly classified as BRCA1-type and BRCA2-type by CHORD, further validating our results, and that two out of the six patients with germline mutations in HR genes have a history of cancer (breast cancer for one BRCA2-mutated patient, and oral cavity Carcinoma and skin Carcinoma for another BRCA2-mutated patient; 3 patients have no data available and one had no reported history of cancer). We now provide these results in **Table S13**.

In Figure 3, only NF2 is reported on chr22q; however, there are multiple cancer driver genes (TSS) located on this region (such as CHEK2 and EP300). Unless there is a need to emphasize NF2 for its link to MPM, these two cancer driver genes should be considered. “No answer was provided.”

Answer: The *NF2* association with this disease is emphasized as it is one of the most well-defined driver genes of MPM. As presented in Fig. 4, *NF2* was detected as an IntOGen driver gene in the MESOMICS cohort, but also in the TCGA and Bueno series individually, and jointly (see Fig. S21C for details). For this reason, we prefer not to report other cancer genes in Fig. 3b but focus on MPM driver genes. Fig. 4 reports all MPM driver genes (IntOGen and recurrent SV-altered genes) and *CHEK2* and *EP300* were not found in this list. However, the Reviewer is right that two other genes of note, *MYH9* (IntOGen driver) and *TTC28* (recurrent SV-altered gene) do share the same chromosome arm as *NF2*, significantly deleted in MESOMICS series (Fig. 3b). We annotated the genes in Fig. 4 as belonging to significant broad or focal copy number events detected by GISTIC2, and now mentioned these two other interesting genes on chr 22q in the manuscript p.8-9, and, of note, the list of genes for each focal GISTIC2 event is also given in Table S9.

19. In Fig.4, can the specific structural variants be defined (large indels, translocations and fusion transcripts)? For example, do large indels overlap with heterozygous deletion? What is the difference between translocation and fusion transcripts? How was the “CNB” estimated? By number of segmentations? Copy number breakpoints? “No answer was provided”

Answer: We indeed see that the legend of Fig. 4 was not sufficiently clear, and now provide the definitions of all these elements in the legend. We mention that large indels are detected by SV callers and thus can overlap copy number deletions as described in Fig. 3. Translocation can refer to intra- or interchromosomal translocation detected by SV callers while fusion transcripts are detected at the transcriptomic level and can overlap SV events. CNB has indeed been calculated as the number of CN segments, as mentioned in p.23.

Are any specific alteration types in BAP1 and NF2 in the matched T/N WGS data? “No answer was provided.”

Answer: We have used Fisher's exact test to identify enrichment for alteration types in *BAP1* and *NF2* genes (CN types as presented in **Fig. 3a** and SVs and SNV types as presented in **Fig. 4**) in T-only ($n=72$) or T/N ($n=43$) groups. Using the data available in **Tables S10, S12, and S13**, we tested for *BAP1* heterozygous ($n=41$, q -value=0.18) and homozygous deletions ($n=5$, q -value=1), translocation ($n=13$, q -value=0.66), and frameshift indels ($n=11$, q -value=0.66), and for *NF2* heterozygous deletions ($n=67$, q -value=0.26), large indels ($n=7$, q -value=0.76) and translocation ($n=13$, q -value=0.76), and frameshift indels ($n=5$, q -value=0.24), and corrected for multiple testing for each gene. Therefore, we did not find any significant enrichment in alteration types in T-only or T/N samples (all q -value>0.05).

20. Mutational signatures: the authors mentioned de novo signature extraction for SNVs, but no results are reported. "No answer was provided"

Answer: We agree with the Reviewer that this is important information, in particular to support the non-existence of an asbestos SBS signature, and we have thus added the results of the *de novo* extraction to **Table S13**, in particular the TMB for each de novo signature and the mapping of de novo signatures to known COSMIC signatures. We now mention in the Methods p.23 that "Five *de novo* signatures were identified and decomposed with high fidelity (cosine similarities greater than 0.93) into 10 known COSMIC signatures."

It is interesting that APOBEC mutations have much lower TMB than other signatures, including the age signature (Fig S10). This is different than many cancer types and could be discussed in the Discussion. "No answer was provided".

Answer: We realize that the S-plot representation in **Fig. S10** (now **Fig. S13**) is not well adapted to compare TMB across signatures, because it does not show whether tumors with a low APOBEC signature TMB actually correspond to tumors with a high age-signature TMB. We have investigated the observation of the Reviewer and directly compared the joint TMB of APOBEC and age signatures across samples from the MESOMICS and PCAWG cohorts (new **Fig. S13D**, see below), and actually find that MPM is similar to other tumors with a low burden of APOBEC signatures: age signatures have a slightly larger TMB than APOBEC signatures. This is because tumors with a low SBS2 and SBS13 TMB also tend to have a low SBS1, SBS5, and SBS40 signatures. We now mention this observation on p.23.

Figure S13D. Comparison of the Relative TMB of age related signatures SBS1, 5, and 40, and APOBEC signatures SBS2 and 13 in the MESOMICS and PCAWG cohorts.

Moreover, APOBEC 2 and 13 are usually associated. In this figure, SBS13 is only presented in half of samples with SBS2. Also, 3 samples have a platinum therapy signature: weren't the samples treatment naïve? If not, this should be mentioned at the beginning, since treatment may strongly affect genomic changes. If yes, why these signatures? The authors may want to check the signature assignment. "No answer was provided."

Answer: We now comment on this in the Methods p.23 and show the results in a new panel **Fig. S13C**. Indeed, as the Reviewer mentioned, SBS13 is absent in 3 out of 6 samples where we detected SBS2: two samples where SBS2 was present at a very low level (just 4 and 26 mutations), and one sample where it was present at higher levels (169 mutations). We compared this observation with the PCAWG mutational signatures (*Alexandrov et al. Nature, 2020*) and found similar patterns in such low-SBS2 signature samples (see figure below), such as the Lymph-BNHL cohort (Lymphoid - Mature B-cell lymphoma) where 3 out of the 8 samples with SBS2 did not have any trace of SBS13.

Figure S13C. Comparison of the mutation burden of APOBEC signatures SBS2 and 13 in the MESOMICS cohort and in more than 2000 tumors from the PCAWG cohort.

Regarding the platinum agent signature, it appears we forgot to mention in the mutational signature analysis methods that for these analyses, we have added the 3 samples known to be non-chemonaive to confirm the high-level presence of a platinum therapy signature; we now mention this on p.23. Nevertheless, the non-chemonaive were excluded from all other analyses to avoid the effects the Reviewer is mentioning as we mention in the manuscript p.15 in the description of the cohort.

The authors proposed age as the etiological factor responsible for signatures 1, 5, and 40. However, at least for signature 40 the etiology is still unknown.

The authors showed 7 copy number signatures but it's unclear how the number of copy number signatures were derived/chosen and how they were assigned to be associated with certain processes. "No answer was provided."

Answer: We now mention in the legend of **Fig. S13** that although SBS40 is associated with age, its etiology is still unknown. We have added more details about how the copy number signatures were derived and what reference we used for signature attribution and associated processes p.23, and also added the *de novo* CNV signatures and assignment to COSMIC signature statistics in **Table S10**. In addition, because CNV signatures are very new and rapidly changing, we have added the exact signatures we fit (based on COSMIC 3.1) in **Table S10**.

21. **ecDNA: on Figure 3C, the ecDNA region on chr10 appears to have extremely high WGS coverage compared to the rest of the region, but the estimated copy numbers are not higher than other regions. I suspect copy numbers were wrongly estimated here. Otherwise, how can the authors explain this? "No answer was provided."**

Answer: We thank the Reviewer for their careful review of the figure that allowed us to identify a bug in the plotting of the coverage for some regions of the amplicon. Indeed in former **Fig. 3c**, a CN of 2 corresponded to a coverage of 10X, and some regions on chr10 seemed to have a coverage >100X, which should have roughly corresponded to a CN >20, but CN estimates reported in the figure were only around 5-10. We discovered that this was due to a default option of AmpliconArchitect that downsampled the reads to 10X for fast computing of an approximate coverage, but this approximation was apparently off by a factor of 2 in this region. We thus replotted the figure removing this option, forcing AmpliconArchitect to plot the exact coverage in the region using all mapped reads, and this resolved the issue (see **revised Fig. S14** below). In recomputing this plot, we also discovered that the AmpliconArchitect developers have created a new tool to visualize ecDNA structures in a simpler way (CycleViz, <https://github.com/jluebeck/CycleViz>). We have now opted for this new representation in the main figure (see below), while keeping the more complex representation showing split reads and coverage, which is perhaps more useful for bioinformaticians but less for biological interpretations, in **Fig. S14**.

Figure S14. ecDNA prediction for amplicon 1 of sample MESO_019_T.

Revised Figure 3c. Patient with oncogene amplification due to a chromothripsis event (MESO_019). Left: Chromosomes involved in chromothripsis event (outer circle: shattered regions, intermediate circle: copy number, inner circle: structural variants). Middle: reconstructed ecDNA structure. Right: gene expression in MESO_019 relative to the expression in other tumors from the cohort (quantile). Oncogenes found within the ecDNA region are represented in red.

In Figure S11, there appears to be a recurrent ecDNA identified on chr13:15-17Mbp. However, this region largely overlaps with the chr13 centromere, which should be excluded from the analyses. “No answer was provided.”

Answer: There are indeed 4 out of 105 samples where the AmpliconArchitect software detected an amplicon from a seed in the chr13 region 16.0-16.5Mbp. We kept these regions because they are outside of the list of centromeric regions excluded by the Amplicon suite pipeline during the seed search step by CNVkit (step finding “seed” highly amplified segments), which uses the UCSC cytoBands for the position for centromeres (p11.1 and q1; 16.5-18.9Mbp). In addition, although AmpliconArchitect can extend the seeds to neighboring regions and does not explicitly exclude centromeric regions, it implements several procedures to avoid spurious calls in repetitive and low-complexity regions (in particular, we now mention our use of the `amplified_intervals.py` script for filtering such regions before using the seeds), such as creating a mappability map of the human genome and filtering out reads based on mappability. Finally, the amplicons we detected were each supported by multiple split-reads after these filters (>25 breakpoints per ecDNA amplicon), which we believe is unlikely to be an artifact. In the end, we thus prefer to report these segments; we now mention this in the Methods p.24, and also mentioning that apart from MESO_019, no ecDNA included known oncogenes.

In Figure S13, in tumor 019T there is chromothripsis in the same regions where ecDNA was identified. The authors could emphasize this finding, since chromothripsis can be a primary mechanism of genomic DNA rearrangements and amplification into ecDNA. "No answer was provided."

Answer: We thank the Reviewer for noting this interesting fact, we now mention the possible link between these two observations on p.7 and highlight it in **Fig. 3c** (see above).

22. In Figure S8, the color legend is missing. "The color legend for the Pearson correlation has been added to what is now Figure S9."

Answer: Done.

Reviewer #3:

Remarks to the Author:

I appreciate the thoughtful response to the previous critiques.

Answer: Thanks!

Decision Letter, second revision:

7th Nov 2022

Dear Dr. Fernandez-Cuesta,

Thank you for submitting your revised manuscript "Whole-genome sequencing and multi-omic integrative analyses reveal novel axes of molecular variation and specialized tumor profiles in Malignant Pleural Mesothelioma" (NG-A59168R1). It has now been seen by Reviewer #2 and their comments are below. The reviewers find that the paper has improved in revision, and therefore we'll be happy in principle to publish it in Nature Genetics, pending minor revisions to satisfy our editorial and formatting guidelines.

Sincerely,

Safia Danovi
Editor
Nature Genetics

Reviewer #2 (Remarks to the Author):

I am satisfied by the authors' thoughtful and detailed responses to my questions and comments.

Final Decision Letter:

26th Jan 2023

Dear Dr. Fernandez-Cuesta,

I am delighted to say that your manuscript "Multi-omic analysis of malignant pleural mesothelioma identifies molecular axes and specialized tumor profiles driving inter-tumor heterogeneity" has been accepted for publication in an upcoming issue of Nature Genetics.

Your paper will be published online after we receive your corrections and will appear in print in the

next available issue. You can find out your date of online publication by contacting the Nature Press Office (press@nature.com) after sending your e-proof corrections. Now is the time to inform your Public Relations or Press Office about your paper, as they might be interested in promoting its publication. This will allow them time to prepare an accurate and satisfactory press release. Include your manuscript tracking number (NG-A59168R2) and the name of the journal, which they will need when they contact our Press Office.

Please note that *Nature Genetics* is a Transformative Journal (TJ). Authors may publish their research with us through the traditional subscription access route or make their paper immediately open access through payment of an article-processing charge (APC). Authors will not be required to make a final decision about access to their article until it has been accepted. [Find out more about Transformative Journals](https://www.springernature.com/gp/open-research/transformative-journals)

Authors may need to take specific actions to achieve [compliance with funder and institutional open access mandates](https://www.springernature.com/gp/open-research/funding/policy-compliance-faqs). If your research is supported by a funder that requires immediate open access (e.g. according to [Plan S principles](https://www.springernature.com/gp/open-research/plan-s-compliance)) then you should select the gold OA route, and we will direct you to the compliant route where possible. For authors selecting the subscription publication route, the journal's standard licensing terms will need to be accepted, including [self-archiving-and-license-to-publish](https://www.nature.com/nature-portfolio/editorial-policies/self-archiving-and-license-to-publish). Those licensing terms will supersede any other terms that the author or any third party may assert apply to any version of the manuscript.

Please note that Nature Portfolio offers an immediate open access option only for papers that were first submitted after 1 January, 2021.

To assist our authors in disseminating their research to the broader community, our SharedIt initiative provides you with a unique shareable link that will allow anyone (with or without a subscription) to

read the published article. Recipients of the link with a subscription will also be able to download and print the PDF.

An online order form for reprints of your paper is available at https://www.nature.com/reprints/author-reprints.html. Please let your coauthors and your institutions' public affairs office know that they are also welcome to order reprints by this method.

If you have not already done so, we invite you to upload the step-by-step protocols used in this manuscript to the Protocols Exchange, part of our on-line web resource, natureprotocols.com. If you complete the upload by the time you receive your manuscript proofs, we can insert links in your article that lead directly to the protocol details. Your protocol will be made freely available upon publication of your paper. By participating in natureprotocols.com, you are enabling researchers to more readily reproduce or adapt the methodology you use. Natureprotocols.com is fully searchable, providing your protocols and paper with increased utility and visibility. Please submit your protocol to <https://protocolexchange.researchsquare.com/>. After entering your nature.com username and password you will need to enter your manuscript number (NG-A59168R2). Further information can be found at <https://www.nature.com/nature-portfolio/editorial-policies/reporting-standards#protocols>

Sincerely,

Safia Danovi
Editor
Nature Genetics